# Fair-FedMOE: Group-Fair One-Shot Federated Learning via Prototype-Guided Experts for Medical Imaging Analysis

Lingzhao Meng [1]  Shuai Guo [1]  Weishan Zhang[† 1]  Zengxiang Li[† 2 3]  Han Yu [4]  Nan Liu [2 5]
Daniel Shu Wei Ting [2 3 5]  Yuru Liu [1]  Shudong Wang [1]  Tao Chen [1]

## Abstract

Group fairness can ensure equitable performance across different demographic subgroups for medical image analysis. However, the current fine-tuned foundation models (FMs) exhibit significant subgroup disparity. One-shot federated learning (OFL) can potentially mitigate this by leveraging cross-institutional data diversity within a single communication round. However, heterogeneous distributions across medical institutions may cause OFL local models to diverge severely, resulting in parameter conflicts that amplify disparity upon aggregation. To address these challenges, we propose Fair-FedMOE, a group-fair OFL framework for medical FMs. During local training, Fairness-aware Expert Routing leverages learnable prototypes to route samples to group-specific experts, enabling subgroup-specialized learning to capture group-specific features without inter-group interference. During model aggregation, Prototype-guided Differential Aggregation computes personalized weights based on prototype similarity and applies differentiated aggregation strategies to filter conflicting updates. We propose RES-AUC, a Rawlsian justice-inspired metric based on worst-group performance that remains stable as groups increase. Extensive experiments on retinal and chest X-ray datasets with multiple FMs demonstrate consistent fairness gains without sacrificing accuracy.

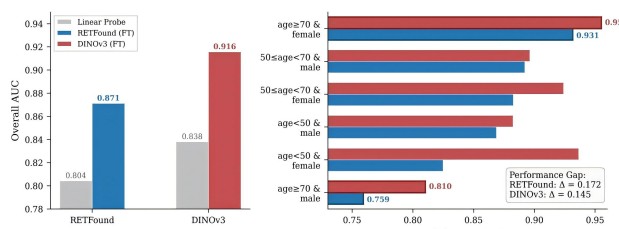

a) Subgroup Performance Disparity

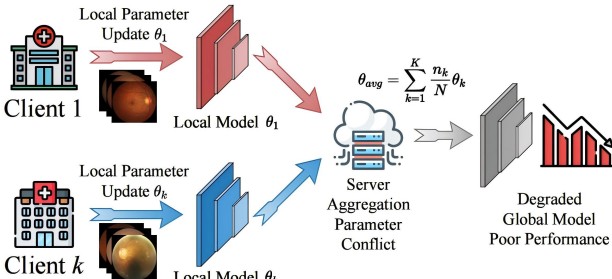

b) Model Inconsistency

*Figure 1.* Fairness challenges in OFL fine-tuning of medical FMs. (a) **Subgroup Performance Disparity**: Both retinal-specific and general-purpose FMs show significant performance gaps across subgroups. (b) **Model Inconsistency**: Data heterogeneity causes local models to diverge. Naive weighted averaging introduces parameter conflicts and degrades global model performance.

Our code is available at https://github.com/lylmlz/Fair-FedMOE. [†]Corresponding author. [1]State Key Laboratory of Chemical Safety, College of Computer Science and Technology, China University of Petroleum (East China), Qingdao, China [2]SingHealth Duke-NUS AI in Medicine Institute, Singapore [3]SingHealth AI Office, Singapore [4]Nanyang Technological University, Singapore [5]Duke-NUS Medical School, Singapore. Correspondence to: Weishan Zhang <zhangws@upc.edu.cn>, Zengxiang Li <shawn.li.zx@singhealth.com.sg>.

*Proceedings of the 43$^{rd}$ International Conference on Machine Learning*, Seoul, South Korea. PMLR 306, 2026. Copyright 2026 by the author(s).

## 1. Introduction

Group fairness requires models to perform consistently across demographic subgroups (Liu et al., 2023; 2025a). This poses a critical challenge for medical image analysis, as prediction bias can lead to misdiagnosis and underdiagnosis (Li et al., 2025b). Foundation models (FMs) have made significant progress in medical imaging (Zhou et al., 2023; Qiu et al., 2024; Oquab et al., 2024; Siméoni et al., 2025). However, as shown in Figure 1 (a), fine-tuned foundation models, whether retinal-specific like RETFound (Zhou et al., 2023) or general-purpose like DINOv3 (Siméoni et al., 2025), still exhibit substantial performance gaps across demographic subgroups. We term this **subgroup performance disparity**, raising a key question: (1) *How to achieve group fairness while maintaining overall performance?*

Demographic imbalance within individual institutions ex-

acerbates this disparity, while cross-institutional data offers greater diversity. Federated Learning (FL) can leverage this diversity to mitigate subgroup disparity without sharing raw data (McMahan et al., 2017; Liu et al., 2025b; Xie et al., 2025; Chen et al., 2025b). However, vanilla FL requires frequent communication, causing high overhead. In medical settings, strict data governance and privacy regulations further limit the feasibility of frequent model exchanges. One-shot FL (OFL) reduces this cost by completing training in a single round (Guha et al., 2019; Zhang et al., 2022; Su et al., 2023; Zeng et al., 2025), but introduces new challenges. As shown in Figure 1 (b), medical imaging data exhibits significant heterogeneity due to differences in equipment, imaging protocols, and patient demographics (Yan et al., 2023; Hu et al., 2025). Such heterogeneity causes OFL local models to diverge severely, a problem we term **model inconsistency**. Naive weighted averaging of inconsistent models causes parameter conflicts, which degrades aggregation quality and further amplifies subgroup disparity. This raises another question: (2) *How to aggregate divergent client models to preserve both performance and fairness?*

To address these challenges, we propose Fair-FedMOE, a group-fair OFL framework for medical FMs. During local training, we introduce Fairness-aware Expert Routing (FER), which decouples subgroup learning through prototype-guided routing, mitigating subgroup performance disparity. During server aggregation, we propose Prototype-guided Differential Aggregation (PDA), which computes personalized weights from prototype similarity and filters conflicting updates through module-specific masks, mitigating model inconsistency. We further introduce regularization during local training to facilitate aggregation by aligning update directions and encouraging parameter sparsity.

We also identify limitations in existing fairness metrics. Disparity-based metrics like DEOdds (Agarwal et al., 2018) and DPD (Agarwal et al., 2019) quantify inter-group performance gaps but ignore absolute performance levels. Composite metrics like ES-AUC (Luo et al., 2024b) consider both performance and disparity but become unreliable as group count increases. Inspired by the maximin principle in Rawlsian justice theory (Rawls, 2017), we propose Rawlsian Equity-Scaled AUC (RES-AUC), which evaluates fairness based on worst-group performance while considering inter-group disparity, and remains bounded and stable as group count increases. Our contributions are as follows:

- To our knowledge, we are the first to investigate group fairness in OFL and identify two key challenges, sub-group performance disparity and model inconsistency.
- We propose Fair-FedMOE, a group-fair OFL framework with Fairness-aware Expert Routing and Prototype-guided Differential Aggregation, enabling subgroup-specialized learning and conflict-aware ag-

gregation within a single communication round.
- We design RES-AUC, a fairness metric based on worst-group performance that remains bounded and stable as the number of sensitive groups increases.
- Extensive experiments on diverse retinal datasets with multiple FMs demonstrate consistent fairness improvements while maintaining competitive accuracy.

## 2. Related Work

### 2.1. Federated Learning in Medical Imaging

Federated Learning (FL) enables collaborative model training across healthcare institutions without sharing raw data, addressing privacy concerns (McMahan et al., 2017; Zhang et al., 2021). However, medical data often exhibits significant statistical heterogeneity due to variations in imaging equipment and patient demographics. This diversity hinders a single global model, motivating Personalized Federated Learning (PFL). Existing PFL methods fall into two categories. 1) Global model adaptation methods that guide local training using global information through regularization (Zhang et al., 2023; Xie et al., 2024a) and cross-domain knowledge transfer via knowledge distillation (Xie et al., 2024b) or data synthesis (Jin et al., 2025). 2) Personalized model learning methods dynamically construct client-specific models tailored to local distributions through parameter decoupling (Sun et al., 2023; Chen et al., 2025b; Gonthina et al., 2025) or Mixture-of-Experts routing (Xie et al., 2025). However, these methods overlook group fairness and typically require frequent client-server interactions, incurring significant communication overhead.

### 2.2. Group Fairness in Federated learning

Fairness in FL differs from centralized settings, where existing research primarily addresses client fairness to ensure similar accuracy across clients (Cheng et al., 2024; Pan et al., 2024). In contrast, group fairness, the focus of our work, aims to ensure equitable performance across demographic groups defined by sensitive attributes (e.g., gender or age). Prior methods can be categorized as follows. 1) Optimization-based methods such as FCFL (Cui et al., 2021), FedMinMax (Papadaki et al., 2022), mFairFL (Su et al., 2024), and FairTrade (Badar et al., 2024) formulate fairness as constrained multi-objective or minimax problems, optimizing worst-group performance to balance utility and fairness. 2) Aggregation-based methods such as FairFed (Ezzeldin et al., 2023) and FedHEAL (Chen et al., 2024) mitigate global bias via dynamically adjusting client weights through adaptive aggregation strategies. 3) Regularization-based methods such as F2PGNN (Agrawal et al., 2024) and FlexFair (Xing et al., 2025) introduce fairness constraints into local optimization objectives, while FairLoRA (Li et al., 2025a) decouples parameters into shared and group-specific

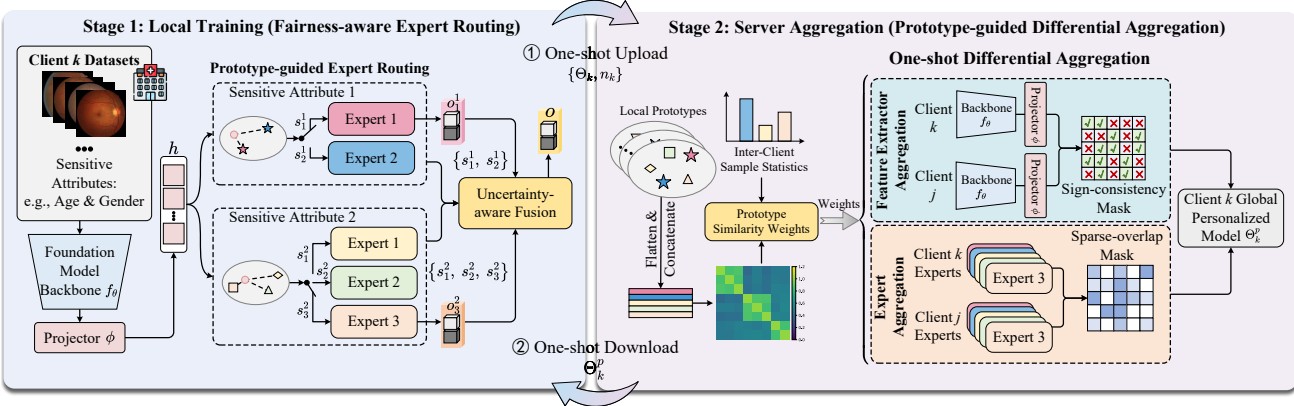

*Figure 2.* Overview of Fair-FedMOE. **Stage 1** (left): Fairness-aware Expert Routing routes samples to group-specific experts via prototype similarity and fuses predictions across attributes with uncertainty-aware weighting. **Stage 2** (right): Prototype-guided Differential Aggregation computes personalized weights based on prototype similarity and applies sign-consistency masks for feature extractors and sparse-overlap masks for experts to generate personalized models.

components via efficient fine-tuning. However, existing methods train a unified model for all groups and rely on regularization or reweighting to mitigate bias, making it difficult to capture distributional differences across subgroups. More importantly, no prior work has explored group fairness in OFL, where single-round communication poses unique challenges. Our work fills this gap.

## 2.3. MoE-based Federated Learning

Mixture-of-Experts (MoE) has been incorporated into FL to handle data heterogeneity via expert specialization. Existing methods differ in the granularity at which experts specialize. 1) Client-level methods assign one expert per client or client cluster (Zec et al., 2020; Dun et al., 2023; Xie et al., 2025). 2) Sample-level methods route each input to experts through a gating network (Jiang et al., 2025; Yi et al., 2026). 3) Task-level methods dedicate experts to individual tasks (Mei et al., 2024). However, in these methods, samples from different demographic groups are processed through shared experts without demographic awareness, inevitably biasing expert learning toward majority groups, failing to capture subgroup-level disparities and precluding group fairness learning.

## 2.4. One-shot Federated Learning

One-shot Federated Learning (OFL) trains a global model through a single client-server communication round, significantly reducing communication overhead. Existing OFL methods can be broadly classified into three categories. 1) Parameter optimization-based methods that directly aggregate model parameters (Su et al., 2023; Liu et al., 2024; Xu et al., 2026). 2) Knowledge Distillation-based methods that transfer knowledge via auxiliary data (Guha et al., 2019; Zhang et al., 2022; Kang et al., 2025; Chen et al., 2025a). 3) Selective ensemble-based methods that combine predictions from multiple client models (Zeng et al., 2024; Tang et al.,

2024; Zeng et al., 2025). However, these methods optimize a unified global model, which struggles with high statistical heterogeneity in medical settings. More importantly, previous work overlooks group fairness in OFL, which is vital in medicine as discussed in Section 2.2.

**Summary.** Existing fairness-aware FL methods rely on unified models that struggle to capture distributional differences across subgroups, while requiring multiple communication rounds. Existing OFL methods reduce communication to a single round but overlook demographic disparities that directly impact diagnostic equity in medical imaging. Our work addresses both limitations within a unified framework.

## 3. Design of Fair-FedMOE

### 3.1. Problem Definition

Consider $K$ clients, where each client $k$ holds a private dataset $\mathcal{D}_k = \{(x_i, y_i, a_i)\}_{i=1}^{n_k}$ with input $x_i$, label $y_i \in \{1, \ldots, C\}$ over $C$ classes, and $M$ sensitive attributes $a_i = (a_i^1, \ldots, a_i^M)$ (e.g., gender, age). For the $m$-th attribute, $\mathcal{G}^m = \{1, \ldots, G_m\}$ denotes the group space of $G_m$ subgroups. We denote the model as $\Theta = \{f_\theta, \phi, \mathcal{P}, \mathcal{E}\}$, where $f_\theta : x \to \mathbb{R}^d$ is the backbone, $\phi : \mathbb{R}^d \to \mathbb{R}^{d'}$ is a projector mapping features to a prototype-aligned embedding space, $\mathcal{P}^m = [p_1^m, \ldots, p_{G_m}^m] \in \mathbb{R}^{G_m \times d'}$ are learnable prototypes for the $m$-th attribute, and $\mathcal{E}^m = \{\mathcal{E}_g^m : \mathbb{R}^{d'} \to \mathbb{R}^C\}_{g=1}^{G_m}$ are group-specific expert heads. All model components, including the feature extractor consisting of backbone $f_\theta$ and projector $\phi$, prototypes $\mathcal{P}$, and experts $\mathcal{E}$, are uploaded to the server for aggregation.

Given this setup, we aim to learn personalized models $\{\Theta_k^p\}_{k=1}^K$ that maximize performance while ensuring group fairness. We consider the OFL setting with single-round communication, which reduces overhead but introduces model inconsistency due to the lack of iterative updates.

## 3.2. Overview

Figure 2 illustrates the Fair-FedMOE framework. In the *local training stage*, we propose Fairness-aware Expert Routing (FER), which maintains learnable prototypes for each group, routes samples to group-specific experts based on prototype similarity, and integrates predictions via uncertainty-aware fusion. In the *server aggregation stage*, we introduce Prototype-guided Differential Aggregation (PDA), which computes personalized weights from prototype similarity, applies sign-consistency masks for feature extractor and sparse-overlap masks for experts to filter conflicting updates, generating personalized models for each client.

## 3.3. Fairness-aware Expert Routing

**Motivation.** Medical imaging data varies significantly across demographic groups. For instance, aging often leads to media opacity and vascular sclerosis, while gender influences fundus pigmentation and structure (Poplin et al., 2018; Betzler et al., 2021). This statistical heterogeneity limits a unified model's ability to fit all groups, leading it to overfit the majority and produce biased predictions (Tan et al., 2023). This challenge mirrors the task conflict problem in multi-task learning, where MoE has shown that routing inputs to specialized experts can decouple conflicting objectives (Chen et al., 2023; Wu et al., 2025). However, standard routing strategies lack explicit demographic guidance (Hua et al., 2025), limiting their ability to capture group-specific features in fairness-critical settings. To address this, we propose Fairness-aware Expert Routing (FER). As illustrated in the left panel of Figure 2, FER routes samples to group-specific experts via learnable prototypes and integrates predictions through uncertainty-aware fusion.

**Prototype Initialization**. To establish a semantic basis for expert routing, we initialize a set of prototypes for each sensitive attribute $m$. Specifically, for each subgroup $g \in \mathcal{G}^m$, the prototype $p_g^{m,(0)}$ is initialized as the feature centroid of the corresponding local samples:

$$p_g^{m,(0)} = \frac{1}{|\{i \mid a_i^m = g\}|} \sum_{i:a_i^m=g} h_i, \qquad (1)$$

where $h_i = \phi(f_\theta(x_i))$ denotes the projected feature.

**Prototype-guided Expert Routing**. To explicitly align inputs with group-specific experts, FER uses a similarity-based routing strategy. Specifically, given the projected feature $h$, we compute its cosine similarity $s_g^m = \frac{h^\top p_g^m}{\|h\|_2 \cdot \|p_g^m\|_2}$ with respect to each prototype. We then select the subgroup index $g^*$ that maximizes this similarity and route the input to the corresponding expert:

$$o^m = \sigma\left(\mathcal{E}_{g^*}^m(h)\right), \quad \text{where } g^* = \arg\max_{g \in \mathcal{G}^m} s_g^m, \quad (2)$$

where $\sigma$ denotes the Softmax function. We apply hard routing independently within each sensitive attribute to enforce expert specialization, and aggregate predictions across different attributes for final classification. During training, the prototype constraint loss $\mathcal{L}_{pc}$ (Eq. 7) uses ground-truth attributes to supervise prototype learning. During inference, routing decisions depend solely on feature similarity, eliminating the need for sensitive attribute labels.

Since $g^*$ serves as a pure index selector and is excluded from the computational graph, gradients propagate through two independent paths. Expert parameters $E_{g^*}^m$ receive gradients from the task loss via standard backpropagation. Prototype parameters $\mathcal{P}^m$ are directly supervised by $\mathcal{L}_{pc}$ through ground-truth attribute labels $a_i^m$, independent of routing decisions. This ensures all prototypes are updated throughout training, preventing expert collapse under hard routing.

**Uncertainty-aware Fusion**. Since clinical diagnosis requires jointly considering multiple sensitive attributes, the model needs to aggregate predictions across different attributes. However, expert confidence varies, making naive averaging suboptimal. To this end, we propose an uncertainty-aware fusion strategy that considers two types of uncertainty: routing uncertainty, where lower prototype similarity indicates more ambiguous group membership, and prediction uncertainty, quantified by output entropy. The fusion weight $\alpha^m$ is computed as:

$$\alpha^m = [s_{g^*}^m]_+ \cdot \exp\left(\sum_{c=1}^{C} o_c^m \ln o_c^m\right), \qquad (3)$$

where $[\cdot]_+$ denotes the ReLU operation to ensure non-negative fusion weights. Consequently, experts exhibiting both high routing similarity and low prediction uncertainty receive higher importance. The final prediction is obtained via weighted integration $o = \sum_{m=1}^{M} \frac{\alpha^m}{\sum_{m'=1}^{M} \alpha^{m'}} o^m$.

**Remark.** Besides, prototypes $\mathcal{P}^m$ are uploaded to the server for computing personalized aggregation weights.

## 3.4. Prototype-guided Differential Aggregation

**Motivation.** In OFL, the lack of iterative communication exacerbates model inconsistency caused by cross-institutional data heterogeneity. Different modules face distinct challenges. For the feature extractor, clients fine-tune from the same initialization but converge to different local optima, resulting in parameter updates with opposite signs that cancel out under naive averaging (Li et al., 2023), as empirically analyzed in Appendix E.8. For expert networks $\mathcal{E}^m$, limited samples per subgroup make experts susceptible to local noise, necessitating selective aggregation of stably activated parameters. Inspired by the conflict rate analysis of (Xu et al., 2026), let $C = \frac{1}{DK} \sum_i^D \min(k_i, K - k_i)$ denote the overall conflict rate, where $D$ is the total number of parame-

ters, and $k_i$ is the number of clients with a positive update on the $i$-th parameter. Reducing $C$ directly increases $|\mathbb{E}[\Delta\theta^p]|$ and decreases $\mathrm{Var}(\Delta\theta^p)$, stabilizing the aggregated update direction. We extend this analysis to a prototype-guided aggregation setting with group-specific experts. Specifically, prototype-guided weighting up-weights clients with aligned local distributions, sign-consistency regularization and masking reduce $C$ for the feature extractor, and sparse-overlap mask aggregation preserves group-specific signal over local noise for expert networks.

**Prototype Space Alignment.** For cross-client prototype similarity to be meaningful, prototypes must reside in a shared semantic space. In our framework, all clients initialize $f_\theta$ and $\phi$ from identical pre-trained weights, ensuring aligned embedding spaces. $\mathcal{L}_{\mathrm{sign}}$ (Eq. 8) regularizes local update directions to prevent embedding drift. Client-specific variations caused by data heterogeneity are preserved as the semantic basis for personalized aggregation.

**Prototype Similarity Weights.** After local training, each client uploads model parameters $\{f_\theta, \phi, \mathcal{E}\}$ and prototypes $\mathcal{P}$ to the server. Prototypes encode group-specific information learned from local data, and clients with similar prototypes likely share similar data distributions. We flatten and concatenate prototypes as $q_k = \mathrm{flatten}([\mathcal{P}_k^1; \ldots; \mathcal{P}_k^M])$ and compute personalized weights:

$$W_{ij} = \beta \cdot \frac{\exp(\cos(q_i, q_j))}{\sum_{l=1}^K \exp(\cos(q_i, q_l))} + (1-\beta) \cdot \frac{n_j}{\sum_{l=1}^K n_l}, \quad (4)$$

where $\beta$ controls the trade-off between prototype similarity and sample size.

**Proposition 3.1** (Prototype-distribution correspondence). *Assume that the alignment term of $\mathcal{L}_{pc}$ drives same-group features to concentrate near their group centroid, i.e., $h_i \approx \bar{h}_{g,k}$ for $a_i^m = g$. At the first-order optimality of $\mathcal{L}_{pc}$, each $\ell_2$-normalized prototype satisfies $\bar{p}_{g,k}^m \parallel \mathbb{E}_{x \sim \mathcal{D}_{g,k}}[\phi(f_\theta(x))]$, i.e., it aligns with the direction of the group-conditional feature mean of its corresponding subgroup. Consequently, $\cos(q_i, q_j)$ in Eq. (4) measures group-level feature alignment across clients, providing a distributional basis for personalized aggregation weights. Proof in Appendix F.1.*

**One-shot Differential Aggregation.** For the feature extractor comprising $f_\theta$ and $\phi$, let $\Delta\theta_k = \theta_k - \theta^{(0)}$ denote the parameter update of client $k$ relative to initialization. To filter out conflicting updates, we define the sign-consistency mask $\mathcal{S}_{ij} = \mathbb{I}[\mathrm{sign}(\Delta\theta_j) = \mathrm{sign}(\Delta\theta_i)]$ and aggregate only sign-consistent updates:

$$\theta_i^p = \theta^{(0)} + \frac{\sum_{j=1}^K W_{ij} \cdot \mathcal{S}_{ij} \odot \Delta\theta_j}{\sum_{j=1}^K W_{ij} \cdot \mathcal{S}_{ij} + \epsilon} \quad (5)$$

where $\odot$ denotes element-wise multiplication and $\epsilon$ ensures numerical stability.

**Proposition 3.2** (Sign-consistency reduces parameter conflict). *Assume the pretrained initialization $\theta^{(0)}$ induces a dominant optimization direction shared across clients, i.e., $\mathrm{sign}(\mathbb{E}_k[\Delta\theta_{k,i}])$ is consistent for most parameters $i$. Then $\mathcal{L}_{\mathrm{sign}}$ reduces $C$ by discouraging sign crossings during fine-tuning. The mask $\mathcal{S}_{ij}$ in Eq. (5) further restricts aggregation to sign-consistent updates, suppressing residual conflicts. Proof in Appendix F.2.*

For expert networks $\mathcal{E}^m$, we impose $\ell_1$ regularization during local training (Eq. 9) to encourage sparsity. Let $\Omega_k = \mathbb{I}[|\mathcal{E}_k| > \tau_s]$ denote the binary mask of parameters with magnitude above threshold $\tau_s$. We aggregate only co-activated parameters:

$$\mathcal{E}_i^p = \frac{\sum_{j=1}^K W_{ij} \cdot (\Omega_i \odot \Omega_j) \odot \mathcal{E}_j}{\sum_{j=1}^K W_{ij} \cdot (\Omega_i \odot \Omega_j) + \epsilon} \quad (6)$$

**Remark.** $\ell_1$ regularization concentrates expert activations on high signal-to-noise parameters, so co-activated parameters $\Omega_i \odot \Omega_j$ are more likely to carry stable group-specific signal rather than local noise.

### 3.5. Overall Objective

We adopt a two-stage training strategy: each client first trains locally, then uploads parameters to the server for one-shot aggregation. During local training, the objective for client $k$ consists of four components.

**Classification Loss.** We use cross-entropy loss $\mathcal{L}_{cls}$ to supervise task prediction.

**Prototype Constraint Loss.** To encourage sample features to align with their corresponding group prototypes while promoting orthogonality among prototypes within each attribute, we define:

$$\mathcal{L}_{pc} = -\frac{1}{n_k} \sum_{i=1}^{n_k} \sum_{m=1}^M \log \frac{\exp(h_i^\top p_{a_i^m}^m / \tau)}{\sum_{g=1}^{G_m} \exp(h_i^\top p_g^m / \tau)}, \\ + \frac{1}{M} \sum_{m=1}^M \sum_{g,g' \in \mathcal{G}^m, g \neq g'} \left( \bar{p}_g^{m\top} \bar{p}_{g'}^m \right)^2 \quad (7)$$

where $p_{a_i^m}^m$ is the prototype of sample $i$'s group under attribute $m$, $\tau$ is the temperature hyperparameter, and $\bar{p}_g^m = p_g^m / \|p_g^m\|_2$ is the $\ell_2$-normalized prototype.

**Sign Consistency Loss.** To mitigate sign conflicts during local training, we regularize the feature extractor to maintain consistent update directions with initialization:

$$\mathcal{L}_{sign} = \sum_{\theta \in \{f_\theta, \phi\}} \frac{\sum_i \mathrm{ReLU}\left(-\mathrm{sign}\left(\theta_i^{(0)}\right) \cdot \theta_i\right)}{\|\mathrm{sign}\left(\theta^{(0)}\right)\|_1 + \epsilon}, \quad (8)$$

where $\theta^{(0)}$ denotes the initial parameters and $\mathrm{sign}(\cdot)$ returns

**Algorithm 1** Fair-FedMOE Training Procedure

---

**Input:** Client datasets $\{\mathcal{D}_k\}_{k=1}^K$, pre-trained model $\theta^{(0)}$, local training epochs $T$
**Output:** Personalized models $\{\Theta_k^p\}_{k=1}^K$
// Local Training Stage
**for** each client $k = 1, \ldots, K$ **in parallel do**
    Initialize prototypes $\{\mathcal{P}_k^m\}_{m=1}^M$ via Eq. (1)
    Record initial parameter signs $\text{sign}(\theta^{(0)})$
    **for** $t = 1, \ldots, T$ **do**
        Prototype-guided expert routing via Eq. (2)
        Aggregate expert outputs with uncertainty-aware fusion via Eq. (3)
        Compute $\mathcal{L}_{total}$ via Eq. (10) and update $\Theta_k$
    **end for**
    Upload $\{\Theta_k, n_k\}$ to server
**end for**

// Server Aggregation Stage
Compute personalized weight matrix $W$ via Eq. (4)
**for** each client $k = 1, \ldots, K$ **do**
    Aggregate $\{f_{\theta_k}^p, \phi_k^p\}$ with sign-consistency mask via Eq. (5)
    Aggregate $\mathcal{E}_k^p$ with sparse-overlap mask via Eq. (6)
**end for**
**return** $\{\Theta_k^p\}_{k=1}^K$

---

the element-wise sign. $\mathcal{L}_{sign}$ only affects aggregation mask selection, leaving all parameters active in the forward pass.

**Sparsity Regularization.** For expert networks, we impose $\ell_1$ regularization to encourage sparsity:

$$\mathcal{L}_{\ell_1} = \sum_{m=1}^M \sum_{g=1}^{G_m} \|\mathcal{E}_g^m\|_1 \qquad (9)$$

The total objective for client $k$ is formulated as:

$$\mathcal{L}_{total} = \mathcal{L}_{cls} + \lambda_1 \mathcal{L}_{pc} + \lambda_2 \mathcal{L}_{sign} + \lambda_3 \mathcal{L}_{\ell_1} \qquad (10)$$

where $\lambda_1, \lambda_2, \lambda_3$ are hyperparameters that balance each loss term.

Algorithm 1 summarizes the Fair-FedMOE procedure. In the local training stage, each client initializes prototypes, records initial parameter signs, and trains the model for $T$ epochs. After training, clients upload model parameters, prototypes, and sample sizes to the server. In the server aggregation stage, personalized weights are computed based on prototype similarity. The feature extractor is aggregated with sign-consistency masks to filter conflicting updates, while expert networks are aggregated with sparse-overlap masks to preserve group-specific knowledge. Finally, the server returns personalized models to each client.

*Table 1.* ES-AUC vs. RES-AUC on synthetic cases ($A_{\text{mean}} = 80$). Upper: AUC distribution patterns (Symmetric vs. Skewed). Lower: metric stability as group count increases with fixed disparity.

| | Case | Group AUCs | $A_{\min}$ | ES-AUC | RES-AUC |
|---|---|---|---|---|---|
| Pattern | Symmetric | (88, 84, 76, 72) | 72 | 64.52 | 73.68 |
| | Skewed | (92, 88, 80, 60) | 60 | 57.14 | 64.60 |
| #Groups | $G = 2$ | (90, 70) | 70 | 66.67 | 70.00 |
| | $G = 4$ | (90, 83, 77, 70) | 70 | 63.49 | 72.62 |
| | $G = 6$ | (90, 86, 82, 78, 74, 70) | 70 | 58.82 | 73.17 |

*Table 2.* Dataset statistics for DR and glaucoma detection tasks.

| Task | Dataset | Imaging Device | Positive | Negative | Total |
|---|---|---|---|---|---|
| DR Detection | BRSET | Nikon/Canon Fundus | 748 | 10,061 | 10,809 |
| | mBRSET | Smartphone Camera | 1,134 | 3,750 | 4,884 |
| | ODIR-5K | Canon/Zeiss/Kowa Fundus | 2,123 | 4,269 | 6,392 |
| Glaucoma Detection | Harvard-FairVLMed | SLO | 5,048 | 4,952 | 10,000 |
| | FairFedMed-Oph | SLO | 7,431 | 7,734 | 15,165 |

### 3.6. Rawlsian Equity-Scaled Metrics

**Motivation.** Evaluating fairness in medical imaging requires metrics that capture both overall performance and inter-group equity. Disparity-based metrics like DE-Odds (Agarwal et al., 2018) and DPD (Agarwal et al., 2019) measure inter-group performance gaps but ignore absolute performance levels. Composite metrics like ES-AUC (Luo et al., 2024b) account for both, but exhibit severe group-number sensitivity, as shown in Table 1, the metric degrades as the number of sensitive groups increases, even when actual disparity is fixed. Moreover, ES-AUC has no direct link to worst-group performance, which is critical for protecting vulnerable populations in healthcare.

To address these issues, we draw on Rawlsian justice theory (Rawls, 2017), which measures fairness by the outcome of the worst-off rather than the average. Following this principle, we propose Rawlsian Equity-Scaled AUC (RES-AUC), which bases evaluation on the worst-performing group and scales improvement toward the mean based on performance uniformity:

$$\text{RES-AUC} = A_{\min} + (A_{\text{mean}} - A_{\min}) \cdot \gamma, \qquad (11)$$

where $A_{\min}$, $A_{\max}$, and $A_{\text{mean}}$ denote the minimum, maximum, and mean AUC across all groups, and $\sigma_g$ denotes the standard deviation of group-wise AUC values. The discount factor $\gamma = 1 - \frac{2\sigma_g}{A_{\max} - A_{\min} + \epsilon} \in [0, 1]$ penalizes performance variance, so that higher uniformity yields greater credit for mean performance. By construction, RES-AUC is bounded within $[A_{\min}, A_{\text{mean}}]$, ensuring worst-group performance is always reflected while remaining stable across group counts. Proof of the bounds is provided in Appendix F.3.

## 4. Evaluations of Fair-FedMOE

### 4.1. Experimental Setting

**Datasets.** We evaluate Fair-FedMOE on two tasks: diabetic retinopathy (DR) detection and glaucoma detection.

*Table 3.* Performance comparison on ***DR detection*** using RETFound and DINOv3, evaluated across different sensitive attribute settings. **Bold** indicates the best results. Underline denotes the second-best within Traditional FL and One-shot FL. $\Delta_{TFL}$ and $\Delta_{OFL}$ show improvements over the best Traditional FL and One-shot FL methods. We report mean ± std over 3 random seeds.

| Methods | DR Detection (RETFound) | | | | | | DR Detection (DINOv3) | | | | | |
| --- | --- | --- | --- | --- | --- | --- | --- | --- | --- | --- | --- | --- |
| | Age | | Gender | | Age × Gender | | Age | | Gender | | Age × Gender | |
| | AUC | RES-AUC | AUC | RES-AUC | AUC | RES-AUC | AUC | RES-AUC | AUC | RES-AUC | AUC | RES-AUC |
| Local | 86.75±0.26 | 82.02±0.45 | 86.75±0.26 | 85.91±0.46 | 86.75±0.26 | 79.28±0.81 | 89.23±0.82 | 85.97±1.61 | 89.23±0.82 | 88.65±0.78 | 89.23±0.82 | 83.20±0.88 |
| *Traditional FL* (TFL) | | | | | | | | | | | | |
| FedAvg | 79.29±1.71 | 73.06±2.19 | 79.29±1.71 | 77.52±1.58 | 79.29±1.71 | 71.43±2.06 | 84.89±0.88 | 79.41±2.12 | 84.89±0.88 | 84.12±1.19 | 84.89±0.88 | 77.75±2.63 |
| FairFed | 82.65±0.27 | 78.02±0.69 | 82.39±0.29 | 81.56±0.33 | 82.39±0.26 | 75.18±0.43 | 86.98±0.59 | 84.46±0.77 | 86.81±0.73 | 85.69±0.77 | 86.83±0.63 | 81.35±1.39 |
| FairLoRA | 86.90±0.16 | 82.68±0.51 | 86.97±0.19 | 86.10±0.34 | 86.68±0.27 | 80.32±0.58 | 89.76±0.36 | 87.31±1.00 | 90.07±0.27 | 89.28±0.30 | 89.85±0.24 | 84.90±0.84 |
| FlexFair | 82.72±0.29 | 77.35±0.94 | 82.79±0.33 | 82.21±0.38 | 83.54±0.20 | 76.98±0.85 | 90.23±0.72 | 88.81±1.09 | 91.28±0.69 | 90.18±0.83 | 90.46±1.10 | 86.45±1.97 |
| *One-shot FL* (OFL) | | | | | | | | | | | | |
| O-FedAvg | 81.23±0.18 | 74.80±0.39 | 81.23±0.18 | 80.15±0.23 | 81.23±0.18 | 72.16±0.26 | 82.54±1.82 | 75.42±3.01 | 82.54±1.82 | 81.75±1.92 | 82.54±1.82 | 72.67±4.02 |
| FuseFL | 81.97±0.25 | 75.98±0.46 | 81.97±0.25 | 81.48±0.34 | 81.97±0.25 | 73.49±1.31 | 83.37±0.38 | 78.19±0.92 | 83.37±0.38 | 82.39±0.41 | 83.37±0.38 | 75.83±1.05 |
| FAFI | 83.40±0.44 | 78.61±0.70 | 83.40±0.44 | 81.90±0.40 | 83.40±0.44 | 74.17±1.30 | 89.44±0.29 | 87.49±1.18 | 89.44±0.29 | 88.62±0.32 | 89.44±0.29 | 86.13±1.13 |
| **Ours** | **87.64±0.32** | **84.43±1.05** | **87.57±0.32** | **86.83±0.30** | **87.39±0.29** | **82.32±0.80** | **92.35±0.23** | **91.43±0.51** | **92.56±0.24** | **91.61±0.54** | **92.45±0.35** | **89.22±1.61** |
| $\Delta_{TFL}$ | ↑0.74 | ↑1.75 | ↑0.60 | ↑0.73 | ↑0.71 | ↑2.00 | ↑2.12 | ↑2.62 | ↑1.28 | ↑1.43 | ↑1.99 | ↑2.77 |
| $\Delta_{OFL}$ | ↑4.24 | ↑5.82 | ↑4.17 | ↑4.93 | ↑3.99 | ↑8.15 | ↑2.91 | ↑3.94 | ↑3.12 | ↑2.99 | ↑3.01 | ↑3.09 |

As fairness evaluation requires demographic annotations, we use publicly available retinal datasets with sensitive attribute labels, as summarized in Table 2. DR detection involves 3 clients using BRSET (Nakayama et al., 2024a), mBRSET (Nakayama et al., 2024b), and ODIR-5K (Li et al., 2020). Glaucoma detection involves 2 clients using Harvard-FairVLMed (Luo et al., 2024a) and FairFedMed-Oph (Li et al., 2025a). Sensitive attributes include age (age<50, 50≤age<70, age≥70) and gender (female, male), forming 6 intersectional subgroups. For glaucoma, we also consider race (Asian, Black, White), yielding 18 subgroups. To evaluate generalizability beyond retinal imaging, we additionally conduct experiments on chest X-ray datasets for pleural effusion (PE) and atelectasis (AT) detection tasks. Detailed demographic distributions are provided in Appendix A.

**Model Configuration.** Our datasets contain two imaging modalities: color fundus photographs (CFP) for DR detection and scanning laser ophthalmoscopy (SLO) for glaucoma detection. We adopt RETFound-CFP (Zhou et al., 2023) as the primary backbone, as CFP images directly match this pretraining domain, and SLO images share similar visual and structural characteristics with CFP (Esmailizadeh et al., 2026). We also evaluate DINOv3 (Siméoni et al., 2025) as a general-purpose alternative. Most methods, including Fair-FedMOE, update all backbone parameters, while FairLoRA (Li et al., 2025a) retains its native low-rank adaptation strategy. Fair-FedMOE additionally incorporates a projector and group-specific experts. Detailed architecture specifications are provided in Appendix B.

**Training Setup.** Each client trains locally for 50 epochs using AdamW optimizer. The prototype dimension $d' = 512$ and the trade-off coefficient $\beta = 0.7$. Detailed hyperparameters are provided in Appendix B.

**Baselines**. We compare against: (1) *Traditional FL*: Fe-

dAvg (McMahan et al., 2017), FairFed (Ezzeldin et al., 2023), FairLoRA (Li et al., 2025a), FlexFair (Xing et al., 2025); (2) *One-shot FL*: O-FedAvg (Guha et al., 2019), FuseFL (Tang et al., 2024), FAFI (Zeng et al., 2025). Detailed baselines are provided in Appendix C.

**Metrics.** We report AUC for overall performance, and ES-AUC, RES-AUC for fairness evaluation. Results on additional metrics and full definitions are provided in Appendix D.

### 4.2. Main Results

Tables 3 and 4 present results across two clinical tasks and two FMs. Fair-FedMOE achieves the best AUC and RES-AUC under all settings with a single communication round. Compared to Local training, Fair-FedMOE improves RES-AUC by +3.04% on DR detection and +3.94% on glaucoma detection with RETFound under age × gender, showing that cross-institutional diversity benefits fairness. In contrast, FedAvg performs worse than Local, with RES-AUC dropping by 7.85% and 8.38% respectively, as frequent averaging disrupts local optimization and accumulates conflicts. O-FedAvg slightly outperforms FedAvg with only a single round, since clients have more time to train and foundation models provide strong initialization that keeps fine-tuned models in similar parameter regions. However, without conflict-aware aggregation, parameter conflicts still degrade performance. Fair-FedMOE addresses this via sign-consistency regularization during local training and differential aggregation on the server. Compared to TFL, gains are larger under intersectional attributes where disparity is more severe, reaching +2.77% RES-AUC with DINOv3. Compared to OFL, gains reach +8.15% RES-AUC. Results hold across FMs and tasks. DINOv3 yields larger fairness gains due to richer representations that better support expert rout-

*Table 4.* Performance comparison on **glaucoma detection** using RETFound and DINOv3, evaluated across different sensitive attribute settings. **Bold** indicates the best results. Underline denotes the second-best within Traditional FL and One-shot FL. $\Delta_{\text{TFL}}$ and $\Delta_{\text{OFL}}$ show improvements over the best Traditional FL and One-shot FL methods. We report mean ± std over 3 random seeds.

| Methods | Glaucoma Detection (RETFound) | | | | | | | | Glaucoma Detection (DINOv3) | | | | | | | |
|---|---|---|---|---|---|---|---|---|---|---|---|---|---|---|---|---|
| | Age | | Gender | | Age × Gender | | Age × Gender × Race | | Age | | Gender | | Age × Gender | | Age × Gender × Race | |
| | AUC | RES-AUC | AUC | RES-AUC | AUC | RES-AUC | AUC | RES-AUC | AUC | RES-AUC | AUC | RES-AUC | AUC | RES-AUC | AUC | RES-AUC |
| Local | 79.94±0.15 | 73.42±0.14 | 79.94±0.15 | 78.27±0.15 | 79.94±0.15 | 72.60±0.31 | 79.94±0.15 | 68.13±0.57 | 81.73±0.24 | 77.07±0.74 | 81.73±0.24 | 80.13±0.25 | 81.73±0.24 | 76.24±0.68 | 81.73±0.24 | 69.44±1.25 |
| *Traditional FL* (TFL) | | | | | | | | | | | | | | | | |
| FedAvg | 70.28±1.00 | 65.17±0.91 | 70.28±1.00 | 68.98±0.73 | 70.28±1.00 | 64.22±0.52 | 70.28±1.00 | 59.47±2.28 | 72.86±1.02 | 67.75±2.19 | 72.86±1.02 | 71.51±1.21 | 72.86±1.02 | 67.35±1.82 | 72.86±1.02 | 63.25±3.05 |
| FairFed | 76.40±0.15 | 69.83±0.23 | 76.40±0.15 | 75.18±0.19 | 75.54±0.11 | 67.64±0.28 | 74.72±0.06 | 62.79±0.36 | 80.62±0.06 | 74.18±0.22 | 80.62±0.06 | 78.80±0.08 | 80.62±0.06 | 72.76±0.13 | 80.62±0.06 | 68.52±0.61 |
| FairLoRA | 78.59±0.32 | 71.27±0.49 | 78.46±0.26 | 77.15±0.24 | 80.71±0.16 | 73.46±0.39 | 79.30±0.06 | 66.86±0.76 | 81.19±0.21 | 74.63±0.36 | 81.19±0.23 | 79.58±0.26 | 81.20±0.05 | 74.02±0.26 | 82.27±1.32 | 70.99±0.72 |
| FlexFair | 79.54±2.35 | 73.61±2.82 | 76.33±0.05 | 75.14±0.07 | 81.24±0.09 | 74.83±0.13 | 76.33±0.05 | 64.07±0.66 | 79.35±0.23 | 72.54±0.24 | 82.11±0.11 | 80.38±0.11 | 82.11±0.10 | 75.09±0.36 | 82.12±0.10 | 70.01±0.24 |
| *One-shot FL* (OFL) | | | | | | | | | | | | | | | | |
| O-FedAvg | 73.10±0.05 | 66.29±0.04 | 73.10±0.05 | 72.05±0.07 | 73.10±0.05 | 65.34±0.04 | 73.10±0.05 | 59.31±0.57 | 75.39±2.20 | 68.85±2.71 | 75.39±2.20 | 73.36±2.39 | 75.39±2.20 | 67.30±2.93 | 75.39±2.20 | 64.05±2.59 |
| FuseFL | 74.73±0.08 | 68.32±0.08 | 74.73±0.08 | 73.52±0.05 | 74.73±0.08 | 62.69±0.17 | 74.73±0.08 | 62.69±0.45 | 77.32±0.10 | 70.37±0.39 | 77.32±0.10 | 76.55±0.14 | 77.32±0.10 | 70.04±0.28 | 77.32±0.10 | 64.79±0.45 |
| FAFI | 77.08±0.40 | 69.37±0.21 | 77.08±0.40 | 75.71±0.32 | 77.08±0.40 | 68.10±0.30 | 77.08±0.40 | 64.95±0.77 | 79.41±0.07 | 71.98±0.24 | 79.41±0.07 | 77.98±0.06 | 79.41±0.07 | 71.82±0.30 | 79.41±0.07 | 68.09±0.70 |
| **Ours** | **81.50±0.14** | **76.58±0.13** | **81.51±0.09** | **79.86±0.16** | **82.33±0.14** | **76.54±0.12** | **81.88±0.10** | **71.03±0.62** | **82.55±0.51** | **79.09±0.25** | **82.82±0.45** | **81.27±0.51** | **82.80±0.46** | **78.14±0.69** | **83.22±0.49** | **72.76±0.76** |
| $\Delta_{\text{TFL}}$ | ↑1.96 | ↑2.97 | ↑3.05 | ↑2.71 | ↑1.09 | ↑1.71 | ↑2.58 | ↑4.17 | ↑1.36 | ↑4.46 | ↑0.71 | ↑0.89 | ↑0.69 | ↑3.05 | ↑0.95 | ↑1.77 |
| $\Delta_{\text{OFL}}$ | ↑4.42 | ↑7.21 | ↑4.43 | ↑4.15 | ↑5.25 | ↑8.44 | ↑4.80 | ↑6.08 | ↑3.14 | ↑7.11 | ↑3.41 | ↑3.29 | ↑3.39 | ↑6.32 | ↑3.81 | ↑4.67 |

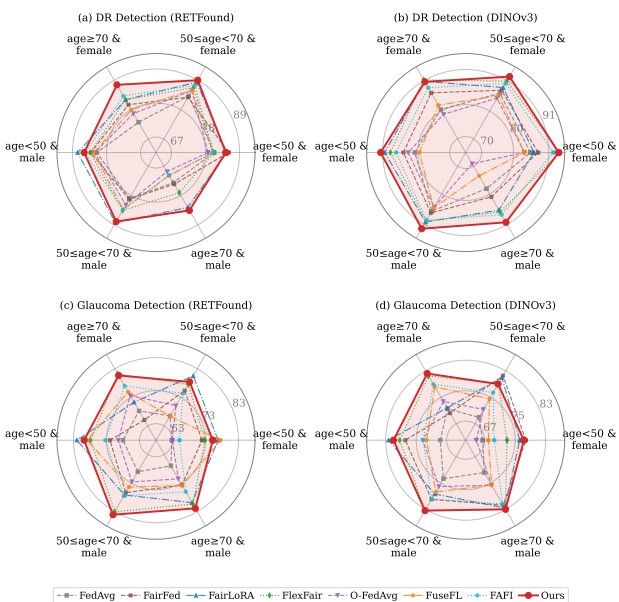

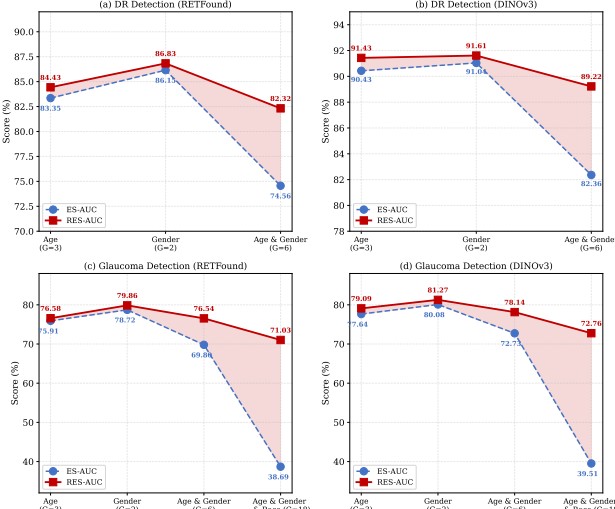

*Figure 4.* Comparison of ES-AUC and RES-AUC as group count increases. ES-AUC degrades sharply while RES-AUC remains stable. (a)-(b) DR detection. (c)-(d) Glaucoma detection.

*Figure 3.* Group-wise AUC across intersectional subgroups (age × gender). Larger and more regular polygons indicate higher overall performance and smaller inter-group disparities.

ing, as shown in Appendix E.4. When groups increase from 6 to 18 under age × gender × race, Fair-FedMOE maintains improvements while FlexFair degrades as its penalty scales with group size. This validates the scalability of our expert design. Detailed per-client results are in Appendix E.

Figure 3 shows group-wise AUC under intersectional attributes age × gender. Fair-FedMOE yields the largest and most regular polygons, indicating superior performance and balanced AUC across subgroups. In comparison, other methods show more irregular shapes, with O-FedAvg performing poorly across multiple subgroups in DR detection, and FairFed and FairLoRA degrading notably on the age≥70 & female subgroup in glaucoma detection. Fair-FedMOE addresses these issues through group-specific expert routing.

**Generalizability.** To assess generalizability beyond retinal imaging, we extend experiments to chest X-ray datasets for PE and AT detection. Fair-FedMOE attains the best AUC and RES-AUC on both tasks, consistently surpassing all TFL and OFL baselines. See Appendix E.3 for details.

**Metric Robustness.** Figure 4 compares ES-AUC and RES-AUC as groups increase from 2 to 18. ES-AUC drops sharply from 78.7% to 38.7% as its penalty accumulates across groups. In contrast, RES-AUC drops only from 79.9% to 71.0%, stable due to its bounded range within $[A_{\min}, A_{\text{mean}}]$.

### 4.3. Ablation Study

**Key Components.** Table 5 evaluates FER and PDA on DR detection with RETFound under age × gender. Local+FER trains clients with FER but without aggregation. It demonstrates FER effectiveness for group-specific representation learning, but underperforms Fair-FedMOE due to limited

*Table 5.* Ablation study on key components of Fair-FedMOE for DR detection with RETFound, evaluated across intersectional subgroups (age × gender).

| Methods | Overall | | | Subgroup Performance | | | | | |
|---|---|---|---|---|---|---|---|---|---|
| | AUC | ES-AUC | RES-AUC | age<50 & female | 50≤age<70 & female | age≥70 & female | age<50 & male | 50≤age<70 & male | age≥70 & male |
| Local+FER | 87.29±0.66 | 72.16±2.37 | 80.60±1.96 | 84.00±1.15 | **89.52±0.90** | 85.72±1.23 | 84.87±1.24 | **87.92±0.68** | 80.85±3.20 |
| O-FedAvg | 81.23±0.18 | 63.31±0.50 | 72.16±0.26 | 78.82±0.32 | 85.15±0.18 | 76.55±0.51 | 81.83±0.45 | 81.51±0.20 | 69.11±0.43 |
| w/o FER | 87.02±1.05 | 73.12±3.14 | 80.89±2.29 | 83.99±3.17 | 89.19±0.96 | 85.33±1.97 | **85.48±0.85** | 87.47±0.81 | 80.08±2.15 |
| w/o PDA | 84.48±0.41 | 70.10±1.62 | 77.92±1.20 | 82.50±1.30 | 87.39±0.36 | 83.30±0.91 | 85.16±0.71 | 84.45±0.39 | 75.88±2.03 |
| **Ours** | **87.39±0.29** | **74.56±2.24** | **82.32±0.80** | **84.56±1.14** | 89.00±0.56 | **87.27±1.52** | 85.25±0.83 | 87.75±0.86 | **83.60±1.99** |

*Table 6.* Ablation study on loss functions of Fair-FedMOE for DR detection with RETFound, evaluated across intersectional subgroups (age × gender).

| Methods | Overall | | | Subgroup Performance | | | | | |
|---|---|---|---|---|---|---|---|---|---|
| | AUC | ES-AUC | RES-AUC | age<50 & female | 50≤age<70 & female | age≥70 & female | age<50 & male | 50≤age<70 & male | age≥70 & male |
| w/o $\mathcal{L}_{pc}$ | 85.35±1.26 | 69.61±1.83 | 78.49±1.66 | 80.54±1.52 | 86.99±1.48 | 83.96±2.14 | 83.52±0.78 | 86.95±0.54 | 79.41±1.69 |
| w/o $\mathcal{L}_{sign}$ | 86.15±0.26 | 71.33±1.05 | 79.49±0.68 | 81.06±1.17 | 87.95±0.35 | 85.03±0.99 | 83.65±0.39 | 87.31±0.43 | 81.27±0.61 |
| w/o $\mathcal{L}_{\ell_1}$ | 86.07±0.18 | 71.37±0.57 | 79.59±0.60 | 81.01±0.95 | 87.91±0.30 | 84.98±0.71 | 83.63±0.49 | 87.09±0.40 | 81.31±1.00 |
| **ALL** | **87.39±0.29** | **74.56±2.24** | **82.32±0.80** | **84.56±1.14** | **89.00±0.56** | **87.27±1.52** | **85.25±0.83** | **87.75±0.86** | **83.60±1.99** |

data diversity. Removing PDA causes a larger drop than removing FER, indicating that mitigating model inconsistency is more critical than local fairness modeling. Their combination yields improvements on worst-performing subgroups, validating complementary roles.

**Loss Functions.** Table 6 isolates each auxiliary loss. Removing $\mathcal{L}_{pc}$ incurs the largest RES-AUC drop, as prototype separation is fundamental to accurate routing. $\mathcal{L}_{sign}$ and $\mathcal{L}_{\ell_1}$ show comparable contributions. The former regularizes update directions to increase sign-consistent parameters for aggregation, and the latter encourages sparse activations to separate shared patterns from client-specific noise. This validates our design, where $\mathcal{L}_{pc}$ governs local routing quality while $\mathcal{L}_{sign}$ and $\mathcal{L}_{\ell_1}$ enhance server-side aggregation.

### 4.4. Hyperparameter Sensitivity

**Trade-off Parameter $\beta$.** Figure 5 (a) shows results with $\beta \in \{0.1, 0.3, 0.5, 0.7, 0.9\}$. AUC remains stable across settings, while RES-AUC peaks at $\beta = 0.7$, indicating that prototype similarity contributes more than sample size for personalized aggregation. The drop at $\beta = 0.9$ suggests that sample size still helps regularize aggregation weights.

**Prototype Dimension $d'$.** Figure 5 (b) shows results with $d' \in \{128, 256, 512, 1024\}$. Performance remains stable across different settings. We set $d' = 512$ in all experiments to balance performance and efficiency.

## 5. Conclusion

In this paper, we propose Fair-FedMOE, the first framework for group fairness in OFL of medical FMs. Fairness-aware Expert Routing addresses subgroup performance disparity by routing samples to group-specific experts via prototype similarity. Prototype-guided Differential Aggregation mitigates model inconsistency through differentiated masking

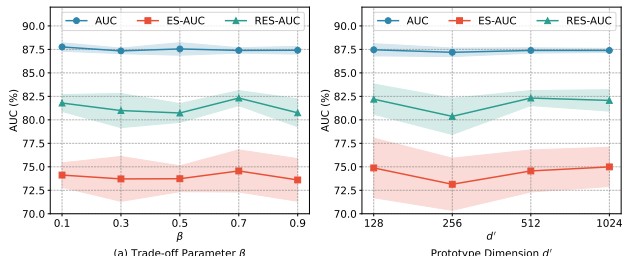

*Figure 5.* Hyperparameter sensitivity analysis on DR detection with RETFound. (a) Trade-off parameter $\beta$, (b) prototype dimension $d'$. Solid lines and shaded regions denote means and standard deviations over 3 random seeds.

strategies that filter conflicting updates. We also introduce RES-AUC, a Rawlsian-inspired metric that remains stable as the number of sensitive groups increases. Extensive experiments demonstrate state-of-the-art fairness with competitive performance in a single communication round.

## Acknowledgments

This work is supported, in part, by the National Natural Science Foundation of China under Grant Nos. 92567204 and 62072469; the Shandong Provincial Natural Science Foundation Major Basic Research Program under Grant No. ZR2024ZD20; the Singapore Ministry of Health through the National Medical Research Council Office, under SIMFONI Programme (Grant ID: RRNMR25SF201); the Ministry of Education, Singapore, under its Academic Research Fund Tier 1 (RG101/24); and the Shandong Data Open Innovative Application Laboratory.

## Impact Statement

This work aims to reduce diagnostic disparities across demographic groups in medical imaging. The proposed framework enables privacy-preserving collaboration across institutions while promoting fairness, with potential benefits for underserved populations. The use of demographic annotations during training requires appropriate data governance.

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

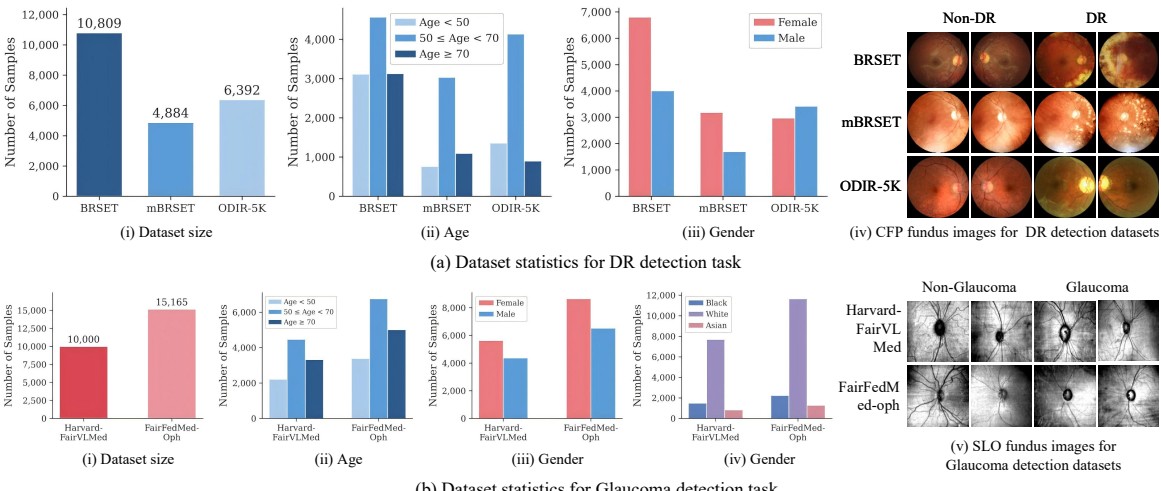

*Figure 6.* Sensitive attribute distributions across datasets for DR and glaucoma detection tasks. (a) DR detection with three clients showing dataset size, age, and gender distributions, along with sample CFP fundus images. (b) Glaucoma detection with two clients showing dataset size, age, gender, and race distributions, along with sample SLO fundus images.

## A. Dataset Details

### A.1. DR Detection (3 Clients)

**BRSET** (Nakayama et al., 2024a) (C1) originally contains 16,266 color fundus photographs from São Paulo, Brazil, acquired with Nikon and Canon fundus cameras following standardized protocols. We use 10,809 images after excluding samples with missing age annotations.

**mBRSET** (Nakayama et al., 2024b) (C2) originally contains 5,164 images from Bahia, Brazil, captured using smartphone cameras for community screening with greater quality variation. We use 4,884 images after excluding samples with missing disease labels.

**ODIR-5K** (Li et al., 2020) (C3) contains 6,392 images from multiple centers in China, collected with diverse devices including Canon, Zeiss, and Kowa, exhibiting the most imaging heterogeneity.

### A.2. Glaucoma Detection (2 Clients)

**Harvard-FairVLMed** (Luo et al., 2024a) (C1) contains 10,000 SLO images from Harvard Medical School, USA, with standardized single-center acquisition.

**FairFedMed-Oph** (Li et al., 2025a) (C2) contains 15,165 SLO images from multiple institutions with greater demographic and imaging variability.

### A.3. Chest X-ray Datasets (5 Clients)

We evaluate generalizability on CheXpert (Irvin et al., 2019), a large chest radiograph dataset collected at Stanford Hospital between October 2002 and July 2017, covering both inpatient and outpatient centers. We select two tasks with complementary characteristics. Pleural Effusion (PE) detection has clear morphological boundaries with balanced labels, where uncertain labels are handled using the U-Ignore strategy. Atelectasis (AT) detection involves subtle structural changes with prevalent label uncertainty handled by the U-Ones strategy, stress-testing FER under noisy supervision.

### A.4. Sensitive Attributes

We consider three sensitive attributes: **Age** (age<50, 50≤age<70, age≥70), **Gender** (female, male), and **Race** (Asian, Black, White). For DR detection, although BRSET provides race annotations, the samples are predominantly from Brazilian patients with limited racial diversity, and other DR datasets do not include race labels. Therefore, we use only age and gender for DR detection, forming 6 intersectional subgroups. For glaucoma detection, all three attributes are available, yielding 18 subgroups. As shown in Figure 6, demographic distributions vary substantially across institutions, and images

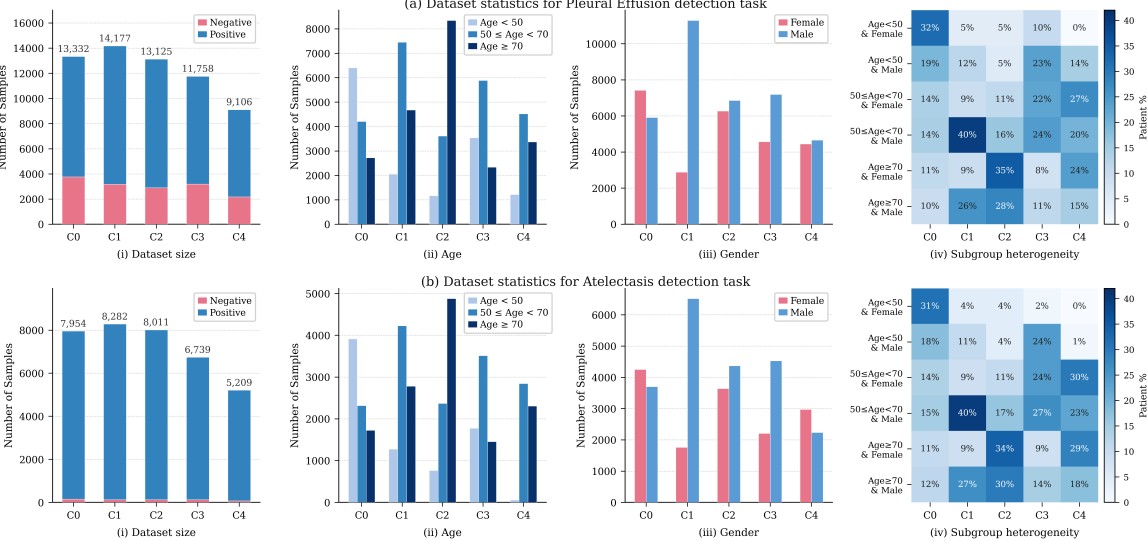

*Figure 7.* Sensitive attribute distributions across clients for PE and AT detection tasks on CheXpert. (a) PE detection with five clients showing dataset size, age, gender distributions, and cross-client subgroup heterogeneity. (b) AT detection with five clients showing dataset size, age, gender distributions, and cross-client subgroup heterogeneity.

exhibit notable visual differences due to device and protocol variations. Such heterogeneity in both population distributions and image characteristics poses significant challenges for achieving group fairness in federated settings. For chest X-ray datasets, we use age and gender as sensitive attributes, forming 6 intersectional subgroups consistent with the DR detection setting. Dataset statistics and cross-client subgroup distributions are shown in Figure 7.

### A.5. Data Splits and Preprocessing

All datasets are split into training, validation, and test sets with a 6:2:2 ratio, except Harvard-FairVLMed which uses the official 7:1:2 split. For chest X-ray datasets, splits are performed at the patient level with 5 demographically heterogeneous clients, using only frontal-view radiographs. All images are resized to 224×224 and normalized using ImageNet statistics. During training, data augmentation is performed using RandAugment and Random Erasing.

## B. Experimental Details

### B.1. Implementation Details

All experiments are conducted on NVIDIA L20 GPUs. We adopt RETFound-CFP and DINOv3-ViT-L/16 as backbone.

**Network Architecture.** The projector is a two-layer MLP with GELU activation, mapping 1024-dimensional ViT-Large features to a 512-dimensional prototype space. The projector and backbone jointly form the feature extractor, aggregated via sign-consistency masks. Each expert is a single linear layer.

**Optimization.** We use AdamW optimizer with weight decay of 0.05. The base learning rate is set to 5e-3 for RETFound and 1e-3 for DINOv3, with layer-wise learning rate decay factors of 0.65 and 0.85, respectively. We adopt cosine annealing schedule with 10 warmup epochs and minimum learning rate of 1e-6. The batch size is 64 per client.

**Hyperparameters.** The loss weights are set as $\lambda_1 = 0.1$ for prototype constraint loss $\mathcal{L}_{pc}$, $\lambda_2 = 1e\text{-}4$ for sign consistency loss $\mathcal{L}_{sign}$, and $\lambda_3 = 1e\text{-}4$ for sparsity regularization $\mathcal{L}_{\ell_1}$. The temperature $\tau$ in Eq. (7) is set to 0.1. The trade-off parameter $\beta$ in Eq. (4) is set to 0.7. The sparsity threshold $\tau_s$ is set to 1e-4. The numerical stability constant $\epsilon$ is set to 1e-8.

### B.2. Baseline Settings

For TFL methods, we set the number of communication rounds to 50 and local training epochs to 1 per round. For OFL methods, we set local training epochs to 50. All baseline methods are implemented using their official codebases with default hyperparameters. FairFed is implemented based on the codebase provided by FlexFair.

### B.3. Random Seeds

We report mean and standard deviation over 3 random seeds. For DR detection, we use seeds {42, 123, 31415}. For glaucoma detection, we use seeds {42, 2023, 31415}.

## C. Baseline Details

### C.1. Traditional Federated Learning Methods

**FedAvg** (McMahan et al., 2017) serves as the standard FL algorithm where clients train models on local data independently and the server aggregates updates by weighted averaging according to dataset sizes. It reduces communication overhead by transmitting model updates only, but suffers from performance degradation under heterogeneous data distributions and does not consider group fairness.

**FairFed** (Ezzeldin et al., 2023) dynamically adjusts client weights based on the discrepancy between local and global fairness metrics. By assigning higher weights to clients whose local fairness measures are closer to the global measure, FairFed steers the global model toward fairer outcomes through fairness-aware aggregation.

**FairLoRA** (Li et al., 2025a) introduces group fairness into FL through SVD-based low-rank adaptation. The key idea is to assign group-specific singular value matrices for preserving intra-group characteristics while sharing singular vector matrices across groups to capture inter-group relationships, enabling parameter-efficient fairness-aware medical image classification.

**FlexFair** (Xing et al., 2025) incorporates a variance regularization term to support multiple fairness criteria, including equal accuracy, demographic parity, and equal opportunity. The weighted penalty mechanism reduces performance disparities across demographic groups while preserving data privacy.

### C.2. One-Shot Federated Learning Methods

**O-FedAvg** (Guha et al., 2019) compresses multi-round FL into a single communication round, where each client trains independently for multiple epochs before uploading model parameters for one-time weighted averaging. While significantly reducing communication overhead, it suffers from model inconsistency under heterogeneous data distributions.

**FuseFL** (Tang et al., 2024) tackles model inconsistency in OFL from a causal perspective through progressive model fusion. It decomposes neural networks into multiple blocks and trains and fuses each block in a bottom-up manner, enabling local models to learn invariant features from other clients and reducing spurious correlation fitting.

**FAFI** (Zeng et al., 2025) targets both intra-model and inter-model inconsistency in OFL with a two-stage design. On the client side, self-alignment local training leverages contrastive learning and category-wise prototype learning to obtain generalizable representations. On the server side, informative feature fused inference attenuates noise-proximal features and uses aggregated prototypes for nearest-neighbor classification.

## D. Evaluation Metrics

### D.1. Performance Metrics

**Area Under the ROC Curve (AUC).** The probability that a positive sample is ranked higher than a negative sample:

$$\text{AUC} = \frac{1}{|P||N|} \sum_{i \in P} \sum_{j \in N} \mathbf{1}[f(x_i) > f(x_j)], \tag{12}$$

where $P$ and $N$ denote the positive and negative sample sets, and $f(\cdot)$ is the predicted score.

### D.2. Fairness Metrics

**Worst-Group AUC.** The minimum AUC across all demographic subgroups:

$$\text{Worst-Group AUC} = \min_{g \in \mathcal{G}} \text{AUC}_g, \tag{13}$$

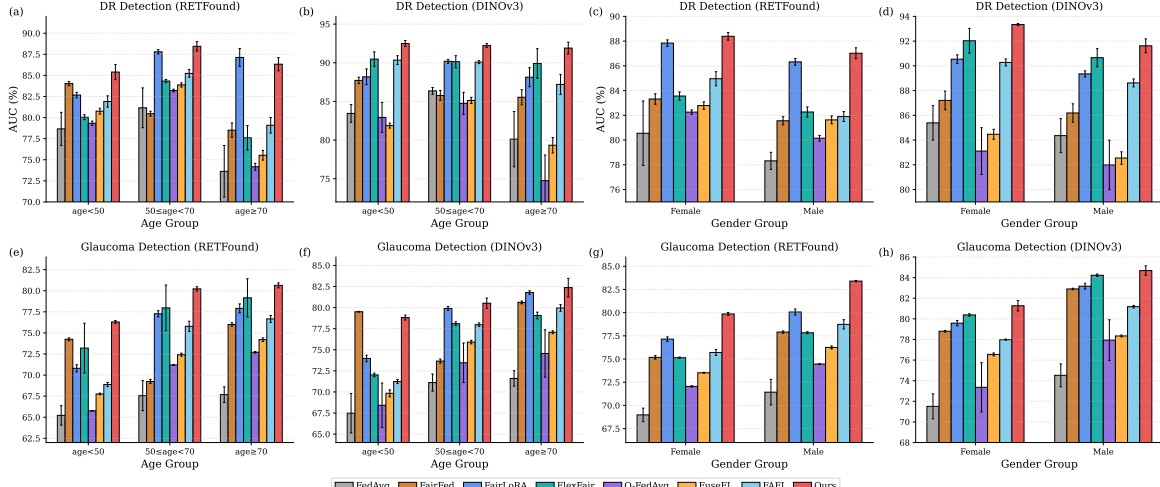

*Figure 8.* Group-wise AUC under single sensitive attribute settings, evaluated on two clinical tasks, DR detection and glaucoma detection, with two foundation models, RETFound and DINOv3. (a)–(b), (e)–(f) show results for age groups. (c)–(d), (g)–(h) show results for gender groups. Error bars denote std over 3 random seeds.

where $\mathcal{G}$ is the set of demographic groups and $\text{AUC}_g$ is the AUC for group $g$. Higher values indicate better worst-case performance across subgroups.

**Difference in Equalized Odds (DEOdds)** (Agarwal et al., 2018). The maximum disparity in prediction rates across groups conditioned on the true label:

$$\text{DEOdds} = \max_{g \in \mathcal{G}, y \in \{0,1\}} \left| P[\hat{Y} = 1 | A = g, Y = y] - P[\hat{Y} = 1 | Y = y] \right|, \tag{14}$$

where $A$ is the sensitive attribute, $Y$ is the true label, and $\hat{Y}$ is the predicted label. Lower values indicate better equalized odds.

**Equity-Scaled AUC (ES-AUC)** (Luo et al., 2024b). A metric that jointly captures performance and fairness:

$$\text{ES-AUC} = \frac{\text{Overall AUC}}{1 + \sum_{i=1}^{N} |\text{Overall AUC} - i\text{-th Group AUC}|}, \tag{15}$$

where $N$ is the number of demographic groups. Higher values indicate a better performance-fairness trade-off.

**Rawlsian Equity-Scaled AUC (RES-AUC).** Our proposed RES-AUC metric is detailed in Section 3.6.

# E. More Experimental Results

This section provides supplementary experimental results, including each client detailed performance and additional group-wise analysis.

### E.1. DR Detection Task

Tables 11 and 12 report each client results on DR detection for single and intersectional sensitive attributes. Figure 8 (a)-(d) shows the corresponding group-wise AUC under single attributes. Due to the imaging heterogeneity described in Appendix A, C3 consistently poses the greatest challenge across all methods. Under the age attribute, all methods show performance degradation on the age≥70 subgroup, but Fair-FedMOE maintains the smallest gap between age groups. Under the gender attribute, inter-group disparities are smaller across all methods, and Fair-FedMOE achieves the highest AUC for both groups. Fair-FedMOE achieves the largest improvements on C3, suggesting that prototype-guided aggregation effectively leverages complementary knowledge from higher-quality clients (C1, C2) while preserving client-specific adaptations. Under intersectional attributes, the expanded subgroup space exacerbates data scarcity for certain demographic combinations, causing baseline methods to exhibit unstable performance on minority groups. As shown in Figure 3 (a)-(b), the age≥70 & male subgroup shows the most notable performance degradation across baselines. Fair-FedMOE mitigates

*Table 7.* Performance comparison on PE and AT detection tasks on CheXpert using DINOv3, evaluated across Age × Gender intersectional attributes. **Bold** indicates best results. Underline denotes second-best within TFL and OFL. We report mean ± std over 3 random seeds.

| Methods | PE Detection | | AT Detection | |
|---|---|---|---|---|
| | AUC | RES-AUC | AUC | RES-AUC |
| Centralized | 95.56 | 94.68 | 71.37 | 65.00 |
| *Traditional FL* (TFL) | | | | |
| FedAvg | 92.39 | 91.36 | 64.04 | 61.49 |
| FairFed | 93.44 | 92.42 | 67.16 | 63.85 |
| FairLoRA | 95.02 | 94.18 | 69.79 | 64.49 |
| FlexFair | 94.79 | 93.76 | 67.71 | 63.43 |
| *One-shot FL* (OFL) | | | | |
| O-FedAvg | 90.79 | 89.43 | 61.88 | 59.25 |
| FuseFL | 92.68 | 91.48 | 66.39 | 62.14 |
| FAFI | 94.31 | 92.54 | 67.12 | 63.39 |
| **Ours** | **95.53** | **94.71** | **70.28** | **65.51** |

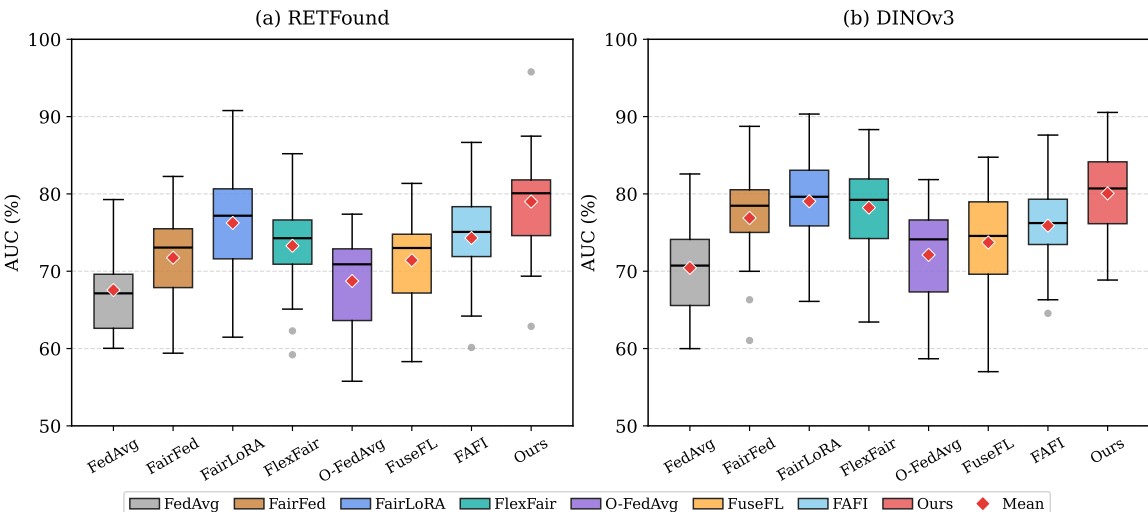

*Figure 9.* Group-wise AUC distribution under intersectional sensitive attributes, age × gender × race, on glaucoma detection with two foundation models, RETFound and DINOv3. Box plots show the distribution across 18 subgroups, with diamonds indicating the mean.

this issue through group-specific expert routing, achieving more balanced performance across all subgroups. DINOv3 shows larger fairness gains than RETFound, likely because its diverse pretraining data yields feature representations with better separability across demographic groups, facilitating more effective prototype learning and expert specialization.

### E.2. Glaucoma Detection Task

Tables 13 and 14 report each client results on glaucoma detection, with Figure 8 (e)-(h) showing group-wise AUC under single attributes and Figure 9 presenting the distribution across 18 subgroups under age × gender × race. Under the age attribute, the age<50 subgroup exhibits the lowest performance while age≥70 achieves the highest, contrasting with DR detection. Under the gender attribute, all methods show performance bias toward males, and Fair-FedMOE achieves the best results for both groups. Despite C2 aggregating multi-center data with greater imaging heterogeneity, Fair-FedMOE maintains consistent improvements on both clients. When race is introduced as third attribute, the subgroup space expands to 18 groups with severe long-tail imbalance. Baseline methods degrade under this setting, as existing fairness-aware FL methods train a unified model for all groups and rely on regularization or post-hoc corrections that scale poorly with increasing group count. In contrast, our prototype-guided expert routing exhibits linear scalability with respect to group count, as each expert specializes in a dedicated subgroup without inter-group interference. As shown in Figure 9, baselines exhibit

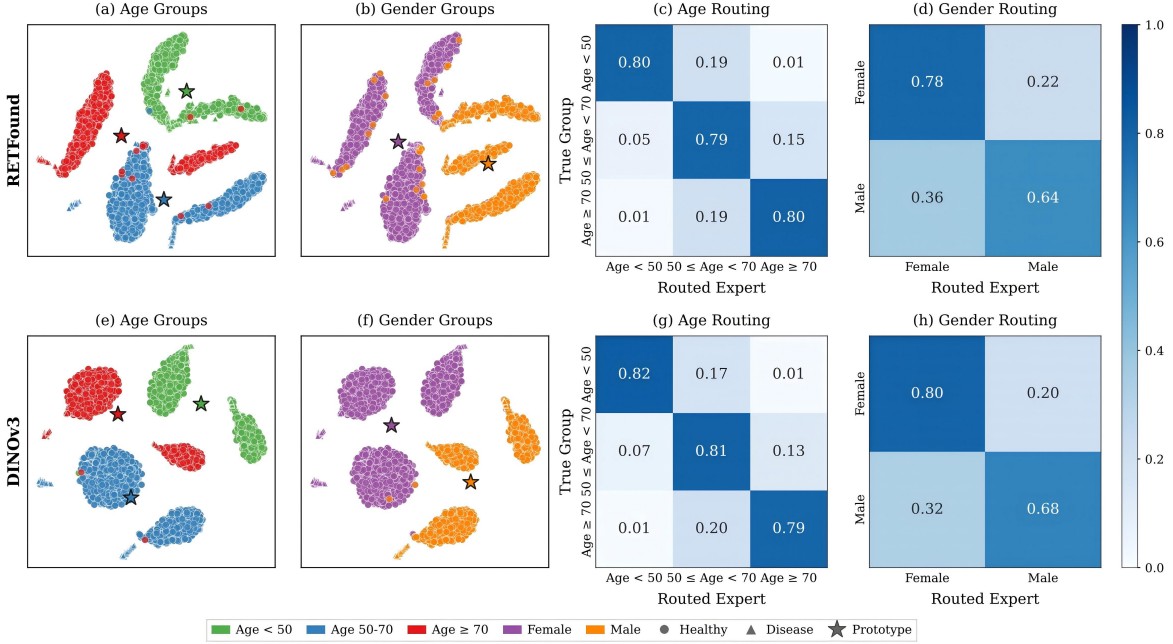

*Figure 10.* Visualization of Fairness-aware Expert Routing (FER) on BRSET client under age × gender. Top: RETFound. Bottom: DINOv3. (a)–(b), (e)–(f): t-SNE of training set features, with colors denoting age and gender groups; stars indicate learned prototypes. (c)–(d), (g)–(h): Routing accuracy on the test set.

wider interquartile ranges, whereas Fair-FedMOE achieves a more compact distribution with higher median performance. The group-wise distribution under age × gender is shown in Figure 3 (c)-(d).

### E.3. Chest X-ray Detection Task

Table 7 presents results on CheXpert for PE and AT detection using DINOv3 under Age × Gender, with 5 clients partitioned at the patient level. Fair-FedMOE achieves the best AUC and RES-AUC on both tasks, approaching the Centralized setting. On PE detection, RES-AUC improves by $+0.53\%$ over the best TFL method and $+2.17\%$ over the best OFL method. On AT detection, where the U-Ones strategy introduces label uncertainty, Fair-FedMOE remains robust, outperforming the best TFL and OFL methods by $+1.02\%$ and $+2.12\%$ in RES-AUC. The larger gains on AT suggest that prototype-guided routing focuses each expert on subgroup-specific signal rather than label noise, yielding stronger benefits under noisy supervision. These results validate the generalization of Fair-FedMOE across different modalities and label strategies.

### E.4. Expert Routing Analysis

We visualize the learned representations of FER on the BRSET client using both RETFound and DINOv3 backbones. Figure 10 (a)–(b) and (e)–(f) show t-SNE projections of training set features, where samples from different demographic groups form well-separated clusters with learned prototypes positioned near their respective centroids. DINOv3 yields more compact clusters with clearer inter-group boundaries, consistent with its stronger fairness gains in Tables 3 and 4.

Figure 10 (c)–(d) and (g)–(h) report routing accuracy on the test set. Both backbones achieve high diagonal values, confirming that samples are correctly routed to group-specific experts based solely on feature-prototype similarity, without access to sensitive attributes at inference. DINOv3 attains slightly higher routing accuracy (e.g., 0.82 vs. 0.80 for age<50), indicating that its richer representations better support prototype-guided expert routing.

### E.5. Privacy Analysis

Prototypes uploaded to the server are continuously updated via $\mathcal{L}_{pc}$, encoding richer demographic information than fixed centroids and thus more susceptible to inference attacks. To mitigate this, we $\ell_2$-normalize each prototype prior to upload, i.e., $\bar{p}_g^m = p_g^m / \|p_g^m\|_2$. For any two normalized prototypes, $\|\bar{p} - \bar{p}'\|_2 \leq 2$, bounding the global $\ell_2$-sensitivity at $\Delta_f = 2$. Each client then injects Gaussian noise before uploading as:

*Table 8.* Performance of Fair-FedMOE under varying privacy budgets $\varepsilon$ on DR detection with RETFound, evaluated across Age $\times$ Gender attributes. We report mean $\pm$ std over 3 random seeds.

| Method | AUC | RES-AUC |
|---|---|---|
| Ours ($\varepsilon = 1$) | $87.12 \pm 0.29$ | $81.88 \pm 0.99$ |
| Ours ($\varepsilon = 5$) | $87.29 \pm 0.20$ | $81.99 \pm 1.17$ |
| Ours ($\varepsilon = 10$) | $87.38 \pm 0.31$ | $82.23 \pm 1.18$ |
| Ours (Non DP) | $\mathbf{87.39 \pm 0.29}$ | $\mathbf{82.32 \pm 0.80}$ |

*Table 9.* Ablation study on routing strategy for DR detection with RETFound, evaluated across intersectional subgroups of age $\times$ gender. **Bold** indicates the best results. We report mean $\pm$ std over 3 random seeds.

| Methods | Overall | | | Subgroup Performance | | | | | |
|---|---|---|---|---|---|---|---|---|---|
| | AUC | ES-AUC | RES-AUC | age<50 & female | 50≤age<70 & female | age≥70 & female | age<50 & male | 50≤age<70 & male | age≥70 & male |
| Soft Routing | **87.58±0.20** | 73.92±0.95 | 81.45±0.34 | 83.68±1.10 | **89.00±0.31** | 87.83±0.61 | 85.69±0.41 | **88.46±0.32** | 81.39±1.05 |
| Hard Routing | 87.39±0.29 | **74.56±2.24** | **82.32±0.80** | **84.56±1.14** | 89.00±0.56 | **87.27±1.52** | 85.25±0.83 | 87.75±0.86 | **83.60±1.99** |

$$\tilde{p}_g^m = \bar{p}_g^m + \mathcal{N}(0, \sigma^2 \mathbf{I}), \quad \sigma = \frac{2\sqrt{2\ln(1.25/\delta)}}{\varepsilon} \tag{16}$$

By the Gaussian Mechanism (Dwork & Roth, 2014), Eq. (16) provides a formal $(\varepsilon, \delta)$-DP guarantee. Since PDA computes aggregation weights via cosine similarity (Eq. 4), which depends solely on vector direction, the perturbation incurs negligible utility loss. Table 8 confirms this across privacy budgets, where at $\varepsilon = 1$ AUC and RES-AUC drop by only $0.27\%$ and $0.44\%$ relative to the non-DP setting.

### E.6. Routing Strategy Analysis

Table 9 compares hard and soft routing on DR detection with RETFound under age $\times$ gender. Although soft routing achieves marginally higher AUC ($+0.19\%$), hard routing outperforms it in RES-AUC by $0.87\%$. Soft routing blends outputs from multiple experts, allowing majority-group representations to dominate and undermining subgroup specialization. Since protecting the worst-off subgroup is the core objective of Fair-FedMOE, hard routing is the more appropriate design choice.

### E.7. Scalability Analysis

Table 10 presents results under varying client counts on DR detection with RETFound under Age $\times$ Gender. Fair-FedMOE achieves the best AUC and RES-AUC across all $K$ settings, with RES-AUC declining by only $2.84\%$ as $K$ increases from 3 to 10. FedAvg and O-FedAvg drop by $10.61\%$ and $5.20\%$ respectively, as increasing client heterogeneity amplifies parameter conflicts. All other methods show consistently lower absolute RES-AUC values, demonstrating the robustness of prototype-guided aggregation under increasing client diversity.

### E.8. Prototype Similarity and Sign-Consistency Analysis

We illustrate the motivation behind the prototype similarity-based aggregation weights and sign-consistency mask in PDA through Figure 11 on DR detection.

**Prototype Similarity vs. Aggregation Quality.** Figure 11(a) shows the prototype cosine similarity and merged model performance under naive averaging across three client pairs. Client pairs with higher prototype similarity consistently yield better post-merge performance, confirming that prototype space effectively captures inter-client distributional relatedness and supporting the use of prototype similarity as the basis for aggregation weights in Eq. (4).

**Information Loss of Sign-Inconsistent Parameters.** Figure 11(b) shows the information loss of sign-inconsistent parameters under naive averaging, defined as $1 - |\Delta_i + \Delta_j|/(|\Delta_i| + |\Delta_j|)$, on models trained without $\mathcal{L}_{sign}$. All three client pairs yield information loss above $0.5$, confirming that sign-inconsistent updates carry severely degraded information after aggregation. Since OFL involves only a single communication round, this loss is unrecoverable, and the sign-consistency mask therefore excludes these corrupted updates, preserving effective model capacity.

*Table 10.* Performance comparison under varying client counts on DR detection with RETFound, evaluated across Age × Gender attributes. **Bold** indicates best results. Underline denotes second-best within TFL and OFL. We report mean ± std over 3 random seeds.

| Methods | $K=3$ | | $K=5$ | | $K=8$ | | $K=10$ | |
|---|---|---|---|---|---|---|---|---|
| | AUC | RES-AUC | AUC | RES-AUC | AUC | RES-AUC | AUC | RES-AUC |
| *Traditional FL* (TFL) | | | | | | | | |
| FedAvg | 79.29±1.71 | 71.43±2.06 | 76.57±0.15 | 66.15±0.66 | 75.38±3.29 | 63.21±5.29 | 72.44±1.67 | 60.82±2.60 |
| FairFed | 82.39±0.26 | 75.18±0.43 | 81.17±0.13 | 73.75±0.58 | 79.44±0.13 | 70.07±0.52 | 75.68±0.16 | 64.81±0.52 |
| FairLoRA | 86.68±0.27 | 80.32±0.58 | 84.99±0.22 | 79.61±1.01 | 86.17±0.29 | 81.06±0.96 | 83.84±0.27 | 77.86±0.72 |
| FlexFair | 83.54±0.20 | 76.98±0.85 | 83.52±0.15 | 77.92±0.40 | 83.00±0.26 | 75.98±0.46 | 79.44±0.36 | 71.41±0.55 |
| *One-shot FL* (OFL) | | | | | | | | |
| O-FedAvg | 81.23±0.18 | 72.16±0.26 | 79.32±0.37 | 67.37±0.91 | 79.97±2.58 | 69.16±4.37 | 78.02±0.59 | 66.96±1.00 |
| FuseFL | 81.97±0.25 | 73.49±1.31 | 80.66±0.86 | 71.46±1.61 | 80.85±0.31 | 71.29±0.74 | 79.76±0.53 | 69.52±1.85 |
| FAFI | 83.40±0.44 | 74.17±1.30 | 81.54±0.75 | 70.15±0.89 | 81.47±0.41 | 73.22±0.94 | 80.80±0.80 | 72.86±1.55 |
| **Ours** | **87.39±0.29** | **82.32±0.80** | **86.64±0.38** | **82.19±0.96** | **87.01±0.33** | **82.23±0.78** | **85.11±0.56** | **79.48±0.82** |

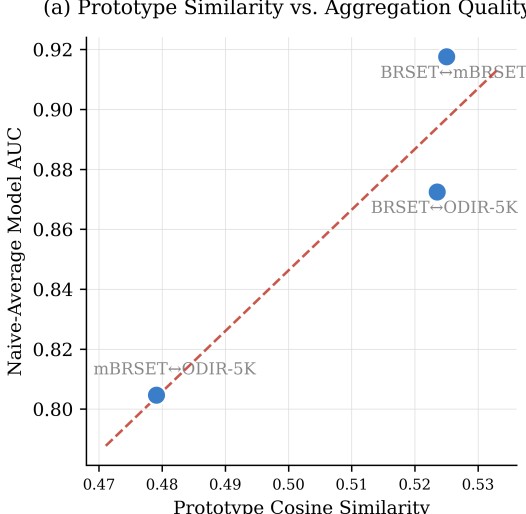
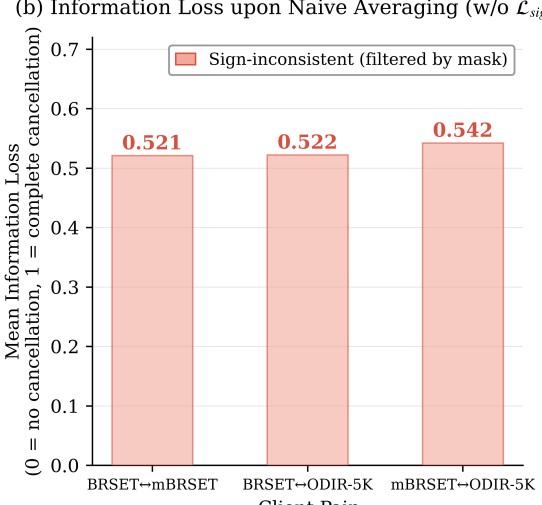

*Figure 11.* Empirical analysis of Prototype-guided Differential Aggregation (PDA) on DR detection. (a) Prototype cosine similarity vs. naive-average model AUC across three client pairs. (b) Mean information loss of sign-inconsistent parameters trained without $\mathcal{L}_{sign}$.

## F. Theoretical Analysis

### F.1. Proof of Proposition 3.1: Prototype-distribution correspondence

**Optimality of $\mathcal{L}_{pc}$ implies prototype–centroid alignment.** Differentiating $\mathcal{L}_{pc}^{\text{align}}$ with respect to $p_g^m$ and setting to zero, where $\tilde{s}_{ig}^m = \exp(h_i^\top p_g^m / \tau) / \sum_{g'} \exp(h_i^\top p_{g'}^m / \tau)$, gives $\sum_{i:a_i^m=g}(1 - \tilde{s}_{ig}^m)h_i = \sum_{i:a_i^m \neq g} \tilde{s}_{ig}^m h_i$. Adding $\sum_{i:a_i^m=g} \tilde{s}_{ig}^m h_i$ to both sides yields the exact equality $n_{g,k} \bar{h}_{g,k} = \sum_{i=1}^{n_k} \tilde{s}_{ig}^m h_i$, where $\bar{h}_{g,k} = \frac{1}{n_{g,k}} \sum_{i:a_i^m=g} h_i$. At optimality, the group centroid equals the softmax-weighted feature mean over all samples.

**Prototype orthogonality guarantees $p_g^m \propto \bar{h}_{g,k}$ is a self-consistent fixed point.** Suppose $\mathcal{L}_{pc}$ has driven same-group features to concentrate near their centroid, i.e., $h_i \approx \bar{h}_{g,k}$ for $a_i^m = g$. Under this condition, $p_g^m \propto \bar{h}_{g,k}$ is a fixed point of the gradient flow. The orthogonality term in $\mathcal{L}_{pc}$ minimizes $(\bar{p}_g^{m\top} \bar{p}_{g'}^m)^2$ for $g \neq g'$ independently, driving $\bar{p}_g^m \perp \bar{p}_{g'}^m$. Under $p_g^m \propto \bar{h}_{g,k}$ and approximate prototype orthogonality, same-group samples ($a_i^m = g$) satisfy $h_i^\top p_g^m \gg h_i^\top p_{g'}^m$, so $\tilde{s}_{ig}^m \to 1$; cross-group samples ($a_i^m \neq g$) satisfy $h_i \approx \bar{h}_{g',k} \perp p_g^m$, so $\tilde{s}_{ig}^m \to 0$. Substituting gives $\sum_i \tilde{s}_{ig}^m h_i \to \sum_{i:a_i^m=g} h_i = n_{g,k}\bar{h}_{g,k}$, confirming the fixed point. For large $n_{g,k}$, $\bar{h}_{g,k} \to \mathbb{E}_{x \sim \mathcal{D}_{g,k}}[\phi(f_\theta(x))]$. After $\ell_2$-normalization before upload:

$$\bar{p}_{g,k}^m \approx \frac{\mathbb{E}_{x \sim \mathcal{D}_{g,k}}[\phi(f_\theta(x))]}{\|\mathbb{E}_{x \sim \mathcal{D}_{g,k}}[\phi(f_\theta(x))]\|_2}. \tag{17}$$

Hence $\cos(q_i, q_j)$ measures the alignment of group-level feature distribution means across clients, providing a distributional basis for the personalized aggregation weights in Eq. (4). □

### F.2. Proof of Proposition 3.2: Sign-consistency reduces parameter conflict

$\mathcal{L}_{\text{sign}}$ **reduces** $C$. Following (Xu et al., 2026), let $\mu$ and $\sigma^2$ denote the mean and variance of the update magnitudes $|\Delta\theta_{k,i}|$. Under naive averaging:

$$\mathbb{E}[\Delta\theta^p] = (1 - 2C)\mu, \qquad \text{Var}(\Delta\theta^p) = \frac{1}{K}\left[\sigma^2 + 4C(1-C)\mu^2\right], \tag{18}$$

where $C = \frac{1}{DK}\sum_i \min(k_i, K - k_i)$ denotes the conflict rate and $k_i$ is the number of clients with a positive update on the $i$-th parameter. Since $4C(1-C)$ is monotonically decreasing on $[0, 1/2]$, and the shared pretrained initialization $\theta^{(0)}$ ensures $C < 1/2$ in practice, reducing $C$ simultaneously increases $|\mathbb{E}[\Delta\theta^p]|$ and decreases $\text{Var}(\Delta\theta^p)$. $\mathcal{L}_{sign}$ (Eq. (8)) imposes a positive penalty when $\text{sign}(\theta_{k,i}) \neq \text{sign}(\theta_i^{(0)})$, discouraging sign crossings during fine-tuning. In OFL, longer local training amplifies inter-client gradient divergence, pushing some clients' parameters past zero and creating sign conflicts with clients moving in the dominant direction. $\mathcal{L}_{sign}$ reduces the frequency of crossings and the magnitude of sign-conflicting updates. By the assumption of Proposition 3.2, $\text{sign}(\mathbb{E}_k[\Delta\theta_{k,i}])$ is consistent for most $i$. Since all clients share $\theta^{(0)}$ and $\mathcal{L}_{\text{sign}}$ keeps each $\theta_{k,i}$ on the same side of zero, update directions across clients tend to align for most parameters, driving $\min(k_i, K - k_i)$ toward zero for most $i$ and reducing $C$. As $C$ decreases, $|\mathbb{E}[\Delta\theta^p]|$ increases and $\text{Var}(\Delta\theta^p)$ decreases, stabilizing the aggregated update direction.

**Mask $S_{ij}$ suppresses residual conflicts via selective aggregation.** For client $k$, let $M_{k,i} = \{j : \text{sign}(\Delta\theta_{j,i}) = \text{sign}(\Delta\theta_{k,i})\}$. Restricting aggregation to $M_{k,i}$ via $S_{ij}$ in Eq. (5) ensures all contributing updates share the same sign:

$$\mathbb{E}[\Delta\theta_{k,i}^{p*}] = \mu_{k,i}, \qquad \text{Var}(\Delta\theta_{k,i}^{p*}) = \frac{\sigma_{k,i}^2}{|M_{k,i}|}, \tag{19}$$

where $\mu_{k,i}$ and $\sigma_{k,i}^2$ are the mean and variance of update magnitudes within $M_{k,i}$. Unlike naive averaging where sign-conflicting updates cancel out, selective aggregation preserves the dominant direction and reduces variance. The $\epsilon$ term ensures numerical stability when $|M_{k,i}|$ is small. □

### F.3. Theoretical Justification for RES-AUC

#### F.3.1. BOUNDED GUARANTEE OF THE DISCOUNT FACTOR

We show that the discount factor $\gamma$ in Eq. (11) satisfies $\gamma \in [0, 1]$.

**Lemma F.1** (Popoviciu's Inequality (Popoviciu, 1935)). *For any random variable $X$ supported on $[m, M]$, its standard deviation $\sigma$ satisfies*

$$\sigma \leq \frac{M - m}{2}, \tag{20}$$

*with equality iff $\mathbb{P}(X = m) = \mathbb{P}(X = M) = 1/2$.*

**Proposition F.2.** $\gamma = 1 - \frac{2\sigma_g}{A_{\max} - A_{\min} + \epsilon} \in [0, 1]$.

*Proof.* Group-wise AUC values lie in $[A_{\min}, A_{\max}]$. By Lemma F.1, $\sigma_g \leq (A_{\max} - A_{\min})/2$, yielding $2\sigma_g/(A_{\max} - A_{\min}) \leq 1$ and thus $\gamma \geq 0$. The bound $\gamma \leq 1$ follows from $\sigma_g \geq 0$. □

#### F.3.2. LIMITING BEHAVIOR

The boundary cases of $\gamma$ correspond to interpretable fairness regimes:

(i) $\gamma = 1$ ($\sigma_g = 0$): All groups achieve identical AUC. RES-AUC reduces to $A_{\text{mean}}$.

(ii) $\gamma = 0$ ($\sigma_g = (A_{\max} - A_{\min})/2$): Half the groups attain $A_{\max}$, the other half $A_{\min}$. RES-AUC reduces to $A_{\min}$, aligning with the Rawlsian maximin principle.

RES-AUC thus ranges between $A_{\min}$ and $A_{\text{mean}}$, balancing worst-group performance against overall equity.

## G. Discussion and Limitations

### G.1. Discussion

Fair-FedMOE addresses subgroup performance disparity and model inconsistency in one-shot federated learning for medical foundation models. To our knowledge, it is the first framework to simultaneously achieve demographic equity and communication efficiency in this setting. Prototype-guided expert routing scales linearly with the number of sensitive groups, as each expert specializes in a dedicated subgroup without inter-group interference. The learned prototypes encode group-specific characteristics that guide routing decisions. As shown in Figure 10, prototypes locate near the centroids of their corresponding clusters, and routing confusion matrices confirm that samples are correctly assigned based solely on feature-prototype similarity. This provides interpretable evidence for clinical adoption. By completing training in a single communication round, Fair-FedMOE suits clinical environments with limited bandwidth or strict data governance. The small client count in our experiments reflects inherent data-sharing barriers across medical institutions, consistent with prior medical FL works (Ezzeldin et al., 2023; Li et al., 2025a). Its core design imposes no domain-specific assumptions, and requires only that each client holds sensitive attribute annotations, with potential applications beyond medical imaging, such as financial services and recommender systems.

### G.2. Limitations

Fair-FedMOE requires sensitive attribute labels during training to supervise prototype learning via $\mathcal{L}_{\text{pc}}$. These labels are used only locally and never transmitted to the server. Inference-time routing relies solely on feature-prototype similarity. When labels are partially missing, $\mathcal{L}_{\text{pc}}$ produces no supervision signal for unlabeled samples. When annotations are systematically absent for an entire subgroup, prototype initialization becomes unreliable. In practice, all clients in our experiments cover every demographic subgroup (Figure 6). Under the extreme case where a client entirely lacks subgroup $g$, the corresponding expert receives no gradient updates and remains at random initialization. The resulting prototype deviates from the shared semantic space, yielding a lower $W_{ij}$ (Eq. 4) and thus attenuating its influence on the global model. Explicit expert-level exclusion under this extreme case remains an open problem, which we leave for future work. The current design also requires the group structure to be predefined, limiting generalization to unseen demographic categories without retraining. Prototypes optimized via $\mathcal{L}_{\text{pc}}$ encode richer demographic information than fixed centroids, making them more susceptible to inference attacks. We apply a $(\varepsilon, \delta)$-DP mechanism to prototype transmission (Appendix E.5) and Table 8 shows that utility loss remains marginal even at $\varepsilon = 1$. However, the privacy-utility trade-off under membership inference attacks remains an open challenge, which we leave for future work. We validate Fair-FedMOE on retinal and chest X-ray datasets with two foundation models. Generalization to other modalities (*e.g.*, CT, MRI) requires further investigation. We plan to explore fairness-aware learning without explicit annotations via unsupervised subgroup discovery.

*Table 11.* Comprehensive per-client performance comparison on **DR detection** using RETFound and DINOv3 for Age and Gender sensitive attributes. **Bold** indicates the best overall results in average performance. Underline denotes the best average performance within Traditional FL or One-shot FL categories when the global best is not from Ours. We report mean ± std over 3 random seeds.

| Methods | Client | Age Attribute — RETFound AUC ↑ | ES-AUC ↑ | RES-AUC ↑ | DEOdds ↓ | Worst-Group AUC ↑ | Age — DINOv3 AUC ↑ | ES-AUC ↑ | RES-AUC ↑ | DEOdds ↓ | Worst-Group AUC ↑ | Gender — RETFound AUC ↑ | ES-AUC ↑ | RES-AUC ↑ | DEOdds ↓ | Worst-Group AUC ↑ | Gender — DINOv3 AUC ↑ | ES-AUC ↑ | RES-AUC ↑ | DEOdds ↓ | Worst-Group AUC ↑ |
|---|---|---|---|---|---|---|---|---|---|---|---|---|---|---|---|---|---|---|---|---|---|
| Local | C1 | 96.13±0.19 | 89.63±0.66 | 91.72±0.58 | 32.16±5.70 | 91.25±0.58 | 97.72±0.09 | 92.79±0.42 | 94.81±0.27 | 30.71±2.84 | 94.46±0.17 | 96.13±0.19 | 95.01±0.47 | 95.70±0.30 | 10.08±3.91 | 95.70±0.30 | 97.72±0.09 | 97.13±0.45 | 97.43±0.33 | 3.53±2.50 | 97.43±0.33 |
| | C2 | 87.12±0.37 | 80.95±0.18 | 81.98±0.28 | 16.49±5.46 | 81.37±0.29 | 91.20±1.13 | 87.31±2.79 | 88.37±2.19 | 18.97±3.85 | 88.01±2.35 | 87.12±0.37 | 86.66±0.78 | 86.87±0.68 | 1.73±0.86 | 86.87±0.68 | 91.20±1.13 | 89.63±1.02 | 90.17±1.00 | 3.80±1.25 | 90.17±1.00 |
| | C3 | 77.01±0.21 | 70.98±1.46 | 72.36±0.50 | 18.18±3.53 | 71.96±0.36 | 78.78±1.23 | 73.76±2.38 | 74.72±2.36 | 13.10±2.12 | 74.26±2.54 | 77.01±0.21 | 74.08±0.66 | 75.15±0.40 | 4.21±1.64 | 75.15±0.40 | 78.78±1.23 | 78.02±0.90 | 78.35±1.01 | 1.88±0.58 | 78.35±1.01 |
| | Avg. | 86.75±0.26 | 80.52±0.77 | 82.02±0.45 | 22.28±4.89 | 81.53±0.41 | 89.23±0.82 | 84.62±1.87 | 85.97±1.61 | 20.92±2.93 | 85.58±1.68 | 86.75±0.26 | 85.25±0.64 | 85.91±0.46 | 5.34±2.14 | 85.91±0.46 | 89.23±0.82 | 88.26±0.79 | 88.65±0.78 | 3.07±1.44 | 88.65±0.78 |
| *Traditional FL* (TFL) | | | | | | | | | | | | | | | | | | | | | |
| FedAvg | C1 | 91.12±1.20 | 83.26±2.20 | 86.48±2.17 | 42.68±5.43 | 85.97±2.23 | 95.16±1.18 | 88.16±1.07 | 92.23±2.12 | 30.81±7.28 | 91.89±2.31 | 91.12±1.20 | 89.27±1.64 | 90.07±1.62 | 9.58±4.94 | 90.07±1.62 | 95.16±1.18 | 94.39±1.14 | 94.84±1.07 | 5.48±1.69 | 94.84±1.07 |
| | C2 | 77.77±1.22 | 68.09±1.24 | 69.04±2.30 | 28.80±2.91 | 68.07±2.10 | 87.37±1.19 | 80.21±0.55 | 83.48±1.14 | 15.94±1.04 | 82.94±0.90 | 77.77±1.22 | 76.96±1.94 | 77.38±1.39 | 3.66±2.39 | 77.38±1.39 | 87.37±1.19 | 86.59±1.06 | 86.76±1.12 | 4.09±1.65 | 86.76±1.12 |
| | C3 | 68.96±2.70 | 62.53±2.71 | 63.67±2.11 | 10.34±0.90 | 63.19±2.30 | 72.14±0.27 | 64.15±2.10 | 62.51±3.09 | 15.86±5.42 | 61.77±3.25 | 68.96±2.70 | 63.80±1.33 | 65.10±1.72 | 6.06±1.21 | 65.10±1.72 | 72.14±0.27 | 70.26±1.37 | 70.76±1.40 | 5.05±1.74 | 70.76±1.40 |
| | Avg. | 79.29±1.71 | 71.29±2.05 | 73.06±2.19 | 27.27±3.08 | 72.41±2.21 | 84.89±0.88 | 77.51±1.91 | 79.41±1.12 | 20.87±4.58 | 78.87±2.15 | 79.29±1.71 | 76.68±1.63 | 77.52±1.58 | 6.43±2.85 | 77.52±1.58 | 84.89±0.88 | 83.75±1.36 | 84.12±1.19 | 4.88±1.69 | 84.12±1.19 |
| FairFed | C1 | 94.77±0.18 | 86.62±1.02 | 89.14±0.92 | 18.83±2.11 | 88.58±0.97 | 96.11±0.35 | 90.47±1.12 | 93.99±0.61 | 36.47±2.88 | 93.80±0.62 | 94.83±0.18 | 93.56±0.16 | 94.39±0.08 | 4.68±0.54 | 94.39±0.08 | 96.13±0.34 | 94.71±0.41 | 95.27±0.38 | 3.75±2.12 | 95.27±0.38 |
| | C2 | 81.01±0.47 | 74.06±1.24 | 77.00±0.86 | 27.47±1.73 | 76.70±0.87 | 88.89±0.60 | 85.44±0.43 | 87.00±0.58 | 32.01±4.35 | 86.65±0.54 | 80.12±0.56 | 78.74±0.93 | 79.09±0.89 | 4.31±0.97 | 79.09±0.89 | 88.73±0.81 | 87.09±1.01 | 87.51±0.98 | 3.80±1.98 | 87.51±0.98 |
| | C3 | 72.18±0.14 | 66.55±0.24 | 67.93±0.29 | 11.69±2.14 | 67.46±0.34 | 75.94±0.82 | 72.76±0.71 | 72.39±1.12 | 22.13±1.34 | 72.10±1.21 | 72.23±0.13 | 70.69±0.06 | 71.21±0.00 | 1.97±0.67 | 71.21±0.00 | 75.58±1.03 | 73.61±0.91 | 74.28±0.95 | 4.70±0.54 | 74.28±0.95 |
| | Avg. | 82.65±0.27 | 75.74±0.83 | 78.02±0.69 | 19.33±1.70 | 77.58±0.73 | 86.98±0.59 | 82.89±0.94 | 84.46±0.77 | 30.20±2.86 | 84.18±0.79 | 82.39±0.29 | 81.00±0.38 | 81.56±0.33 | **3.65±0.73** | 81.56±0.33 | 86.81±0.75 | 85.14±0.78 | 85.69±0.77 | 4.08±2.05 | 85.69±0.77 |
| FairLoRA | C1 | 96.68±0.21 | 94.17±0.57 | 95.25±0.51 | 30.54±1.36 | 95.03±0.52 | 97.74±1.18 | 94.32±0.70 | 95.67±0.43 | 29.18±6.69 | 95.34±0.49 | 96.68±0.11 | 96.00±0.38 | 96.44±0.21 | 7.54±0.28 | 96.44±0.21 | 97.73±0.21 | 97.65±0.25 | 97.73±0.21 | 3.97±2.78 | 97.73±0.21 |
| | C2 | 87.39±0.12 | 79.67±0.76 | 81.41±0.64 | 36.02±1.29 | 80.74±0.64 | 90.40±0.59 | 88.57±1.57 | 89.35±1.50 | 18.95±4.18 | 89.22±1.59 | 87.37±0.28 | 86.70±0.60 | 87.10±0.39 | 2.92±0.62 | 87.10±0.39 | 90.77±0.34 | 90.39±0.35 | 89.08±0.18 | 2.64±1.09 | 89.08±0.18 |
| | C3 | 76.63±0.16 | 70.11±0.66 | 71.39±0.37 | 16.30±2.90 | 70.57±0.26 | 81.15±0.32 | 75.98±1.10 | 76.91±1.07 | 18.80±1.72 | 76.46±1.20 | 76.85±0.19 | 73.62±0.51 | 74.75±0.43 | 5.02±0.96 | 74.75±0.43 | 81.57±0.34 | 80.65±0.63 | 81.04±0.51 | 2.10±1.75 | 81.04±0.51 |
| | Avg. | 86.90±0.16 | 81.32±0.66 | 82.68±0.51 | 27.62±2.52 | 82.11±0.47 | 89.76±0.36 | 86.29±1.12 | 87.31±1.00 | 22.31±4.20 | 87.01±1.09 | 86.97±0.19 | 85.44±0.50 | 86.10±0.34 | 5.16±0.62 | 86.10±0.34 | 90.07±0.27 | 89.56±0.41 | 90.28±0.30 | **2.90±1.88** | 89.28±0.50 |
| FlexFair | C1 | 95.37±0.19 | 87.43±0.92 | 89.86±0.89 | 25.16±3.81 | 89.31±0.89 | 97.79±1.18 | 95.06±0.41 | 96.50±0.28 | 28.97±7.66 | 96.27±0.33 | 95.21±1.34 | 94.26±0.33 | 95.03±0.24 | 10.44±1.23 | 95.03±0.24 | 98.45±0.49 | 97.89±0.79 | 98.13±0.71 | 3.45±2.93 | 98.13±0.71 |
| | C2 | 81.00±0.58 | 72.82±1.69 | 75.80±1.67 | 7.59±1.15 | 75.30±1.93 | 90.24±1.24 | 86.95±1.70 | 88.27±2.33 | 30.80±2.26 | 88.03±2.33 | 81.05±0.60 | 80.33±0.59 | 80.58±0.61 | 1.07±0.57 | 80.58±0.61 | 91.98±0.30 | 90.75±1.39 | 91.41±0.47 | 2.73±0.94 | 91.41±0.47 |
| | C3 | 71.78±0.11 | 64.17±0.62 | 66.40±0.25 | 7.07±0.99 | 66.00±0.20 | 82.65±0.75 | 79.23±0.35 | 81.66±0.97 | 5.56±1.19 | 81.38±0.65 | 72.11±0.23 | 70.49±0.30 | 71.03±0.28 | 5.49±0.29 | 71.03±0.28 | 83.41±1.27 | 79.40±1.40 | 80.99±1.30 | 4.66±1.75 | 80.99±1.30 |
| | Avg. | 82.72±0.29 | 74.81±1.08 | 77.35±0.94 | **13.27±2.12** | 76.87±1.01 | 90.23±0.72 | 87.08±0.82 | 88.81±1.09 | 21.77±3.70 | 88.56±1.10 | 82.79±0.33 | 81.69±0.41 | 82.21±0.38 | 5.67±0.70 | 82.21±0.38 | 91.28±0.69 | 89.35±0.98 | 90.18±0.83 | 3.61±1.69 | 90.18±0.83 |
| *One-shot FL* (OFL) | | | | | | | | | | | | | | | | | | | | | |
| O-FedAvg | C1 | 94.74±0.05 | 85.43±0.21 | 88.16±0.26 | 24.69±1.27 | 87.64±0.23 | 94.16±1.19 | 85.46±2.21 | 89.87±1.64 | 40.93±5.05 | 89.14±1.76 | 94.74±0.05 | 93.97±0.02 | 94.58±0.03 | 5.97±1.43 | 94.58±0.03 | 94.16±1.19 | 93.04±1.31 | 93.74±1.17 | 4.07±2.11 | 93.74±1.17 |
| | C2 | 78.39±0.33 | 68.16±0.24 | 71.52±0.42 | 2.74±0.30 | 70.83±0.50 | 87.06±1.22 | 81.26±1.89 | 83.81±1.62 | 25.49±6.04 | 83.52±1.72 | 78.39±0.33 | 76.84±0.45 | 77.05±0.45 | 3.01±0.33 | 77.05±0.45 | 87.06±1.22 | 85.90±1.69 | 86.18±1.61 | 1.39±0.80 | 86.18±1.61 |
| | C3 | 70.56±0.16 | 63.38±0.56 | 64.71±0.48 | 5.96±0.57 | 64.03±0.51 | 66.40±3.05 | 54.51±4.49 | 57.58±5.78 | 7.00±2.20 | 50.94±6.05 | 70.56±0.16 | 68.17±0.20 | 68.81±0.21 | 5.09±0.82 | 68.81±0.21 | 66.40±3.05 | 64.97±2.61 | 65.34±2.96 | 8.94±2.42 | 65.34±2.96 |
| | Avg. | 81.23±0.18 | 72.32±0.34 | 74.80±0.39 | **11.13±0.71** | 74.17±0.41 | 82.54±1.82 | 73.74±2.86 | 75.42±3.01 | 24.47±4.43 | 74.53±3.18 | 81.23±0.18 | 79.66±0.22 | 80.15±0.23 | 4.69±0.86 | 80.15±0.23 | 82.54±1.82 | 81.30±1.94 | 81.75±1.92 | 4.80±1.78 | 81.75±1.92 |
| FuseFL | C1 | 94.70±0.17 | 84.84±0.50 | 87.79±0.34 | 22.57±1.90 | 87.33±0.41 | 92.71±0.17 | 83.24±0.53 | 87.03±0.55 | 36.34±4.23 | 86.05±0.55 | 94.70±0.17 | 93.76±0.14 | 94.77±0.08 | 7.84±1.92 | 94.77±0.08 | 92.71±0.17 | 91.86±0.24 | 92.45±0.20 | 1.25±0.50 | 92.45±0.20 |
| | C2 | 80.88±0.22 | 72.04±0.27 | 75.76±0.06 | 12.95±2.21 | 75.47±0.16 | 85.81±0.27 | 79.80±0.67 | 80.80±0.58 | 28.83±2.63 | 80.29±0.57 | 80.88±0.22 | 80.40±0.49 | 80.79±0.40 | 0.42±0.19 | 80.79±0.40 | 85.81±0.27 | 85.27±0.24 | 85.54±0.26 | 0.58±0.42 | 85.54±0.26 |
| | C3 | 70.32±0.36 | 64.33±0.63 | 64.40±0.97 | 4.58±0.55 | 63.64±1.02 | 71.60±0.71 | 67.53±1.58 | 66.75±1.44 | 21.32±0.00 | 66.37±1.74 | 70.32±0.36 | 68.24±0.60 | 68.87±0.55 | 5.64±0.45 | 68.87±0.55 | 71.60±0.71 | 68.10±0.79 | 69.20±0.77 | 6.46±0.62 | 69.20±0.77 |
| | Avg. | 81.97±0.25 | 73.74±0.47 | 75.98±0.46 | 13.37±1.22 | 75.48±0.53 | 83.37±0.38 | 76.86±0.92 | 78.19±0.92 | 28.83±2.29 | 77.57±0.96 | 81.97±0.25 | 80.80±0.41 | 81.48±0.34 | 4.63±0.86 | 81.48±0.34 | 83.37±0.38 | 81.75±0.43 | 82.39±0.41 | 4.71±0.61 | 82.39±0.41 |
| FAFI | C1 | 95.91±0.25 | 90.33±0.63 | 92.56±0.58 | 37.33±4.89 | 92.01±0.62 | 96.83±0.32 | 93.86±0.41 | 95.72±0.42 | 30.60±4.53 | 95.50±0.44 | 95.91±0.25 | 95.54±0.23 | 95.89±0.23 | 5.54±1.70 | 95.89±0.23 | 96.83±0.32 | 95.80±0.25 | 96.51±0.26 | 2.35±1.22 | 96.51±0.26 |
| | C2 | 85.21±0.66 | 79.74±0.91 | 81.61±0.72 | 18.38±6.15 | 81.11±0.76 | 92.55±0.08 | 90.71±0.89 | 91.11±0.85 | 23.32±1.70 | 90.98±0.93 | 85.21±0.66 | 83.50±0.42 | 83.95±0.51 | 3.66±1.05 | 83.95±0.51 | 92.55±0.08 | 90.49±0.24 | 91.12±0.17 | 2.96±1.34 | 91.12±0.17 |
| | C3 | 69.00±0.42 | 62.38±0.21 | 61.67±0.99 | 8.04±0.85 | 60.92±0.75 | 78.93±0.46 | 73.53±1.79 | 75.25±2.48 | 15.02±2.59 | 75.03±2.48 | 69.09±0.42 | 64.70±0.47 | 65.86±0.47 | 5.32±2.52 | 65.86±0.47 | 78.93±0.46 | 77.66±0.59 | 78.22±0.59 | 6.80±0.78 | 78.22±0.59 |
| | Avg. | 83.40±0.44 | 77.48±0.58 | 78.61±0.70 | 21.25±3.96 | 78.01±0.71 | 89.44±0.29 | 86.03±1.17 | 87.49±1.18 | 24.05±2.94 | 87.17±1.28 | 83.40±0.41 | 81.25±0.37 | 81.90±0.40 | 4.84±1.76 | 81.90±0.40 | 89.44±0.29 | 87.98±0.36 | 88.62±0.32 | 4.04±1.11 | 88.62±0.32 |
| **Ours** | C1 | 97.38±0.29 | 93.31±1.19 | 94.83±0.90 | 35.10±2.66 | 94.59±0.88 | 99.09±0.04 | 97.13±0.68 | 97.91±0.49 | 27.96±0.71 | 97.80±0.53 | 97.09±0.19 | 96.53±0.36 | 97.14±0.25 | 3.59±0.37 | 97.14±0.25 | 99.19±0.04 | 98.99±0.09 | 99.11±0.03 | 4.65±3.26 | 99.11±0.03 |
| | C2 | 87.79±0.08 | 84.55±0.27 | 85.43±0.62 | 21.62±2.37 | 85.18±1.40 | 93.35±0.05 | 91.43±0.95 | 92.64±0.53 | 18.76±2.33 | 92.52±0.50 | 87.69±0.17 | 87.03±0.25 | 87.33±0.19 | 3.98±1.96 | 87.33±0.19 | 93.48±0.28 | 91.64±0.90 | 91.86±0.69 | 1.71±0.67 | 91.86±0.69 |
| | C3 | 77.73±0.63 | 72.19±0.65 | 73.01±0.63 | 14.75±5.83 | 72.63±0.65 | 84.62±0.59 | 82.54±0.30 | 83.74±0.51 | 15.67±6.22 | 83.66±0.49 | 77.93±0.62 | 74.89±0.36 | 76.01±0.46 | 3.94±0.88 | 76.01±0.46 | 85.02±0.39 | 82.96±1.15 | 83.85±0.90 | 3.41±0.85 | 83.85±0.90 |
| | Avg. | **87.64±0.32** | **83.35±1.30** | **84.43±1.05** | 23.82±2.95 | **84.13±1.11** | **92.35±0.23** | **90.43±0.64** | **91.43±0.51** | 20.80±6.42 | **91.32±0.54** | **87.57±0.32** | **86.15±0.30** | **86.83±0.30** | 3.84±1.07 | **86.83±0.30** | **92.56±0.24** | **91.04±0.70** | **91.61±0.54** | **3.26±1.59** | **91.61±0.54** |

*Table 12.* Comprehensive per-client performance comparison on **DR detection** using RETFound and DINOv3 for Age & Gender combined sensitive attributes. **Bold** indicates the best overall results in average performance. Underline denotes the best average performance within Traditional FL or One-shot FL categories when the global best is not from Ours. We report mean ± std over 3 random seeds.

| Methods | Client | Age & Gender — RETFound AUC ↑ | ES-AUC ↑ | RES-AUC ↑ | DEOdds ↓ | Worst-Group AUC ↑ | DINOv3 AUC ↑ | ES-AUC ↑ | RES-AUC ↑ | DEOdds ↓ | Worst-Group AUC ↑ |
|---|---|---|---|---|---|---|---|---|---|---|---|
| Local | C1 | 96.13±0.19 | 82.89±1.10 | 90.47±0.70 | 42.59±5.45 | 88.17±1.00 | 97.72±0.09 | 85.82±1.51 | 93.10±0.09 | 37.86±6.06 | 91.73±0.45 |
| | C2 | 87.12±0.37 | 66.15±1.25 | 75.77±1.17 | 38.84±3.43 | 72.00±0.91 | 91.20±1.13 | 79.94±2.34 | 83.07±0.52 | 23.38±4.89 | 80.88±0.56 |
| | C3 | 77.01±0.21 | 65.26±1.84 | 71.60±0.55 | 28.93±1.13 | 69.96±1.07 | 78.78±1.23 | 69.61±3.71 | 73.43±2.02 | 20.75±3.20 | 72.06±2.18 |
| | Avg. | 86.75±0.26 | 71.44±1.40 | 79.28±0.81 | 36.79±3.47 | 76.71±1.33 | 89.23±0.82 | 76.79±2.52 | 83.20±0.88 | 27.33±4.72 | 81.56±1.06 |
| *Traditional FL* (TFL) | | | | | | | | | | | |
| FedAvg | C1 | 91.12±1.20 | 76.08±2.57 | 86.38±2.07 | 48.15±8.00 | 84.35±2.33 | 95.16±1.18 | 80.19±4.10 | 90.63±1.38 | 37.21±5.33 | 89.54±1.21 |
| | C2 | 77.77±1.22 | 53.97±3.12 | 66.74±2.97 | 41.26±9.01 | 61.59±4.41 | 87.37±1.19 | 71.18±1.38 | 80.03±3.83 | 24.77±4.99 | 77.59±4.63 |
| | C3 | 68.96±2.70 | 50.79±1.89 | 61.16±1.14 | 17.04±0.15 | 57.67±1.58 | 72.14±0.27 | 54.22±3.51 | 62.60±2.68 | 28.09±3.52 | 59.18±2.93 |
| | Avg. | 79.29±1.71 | 60.28±2.53 | 71.43±2.06 | 35.48±5.72 | 67.87±2.77 | 84.89±0.88 | 68.53±3.00 | 77.75±2.63 | **30.02±4.61** | 75.44±2.92 |
| FairFed | C1 | 94.91±0.20 | 79.14±1.19 | 89.90±0.78 | 32.64±5.10 | 88.65±0.84 | 96.09±0.38 | 82.47±2.50 | 91.13±1.80 | 42.20±2.90 | 89.15±2.56 |
| | C2 | 80.50±0.42 | 59.89±0.78 | 68.12±0.47 | 53.88±1.90 | 63.01±0.46 | 88.94±0.58 | 75.12±0.64 | 83.61±0.48 | 52.60±4.69 | 81.20±0.79 |
| | C3 | 71.75±0.10 | 60.54±0.21 | 67.51±0.04 | 16.42±2.36 | 66.41±0.18 | 75.46±0.94 | 58.16±1.51 | 69.32±2.01 | 46.00±2.22 | 66.58±2.82 |
| | Avg. | 82.39±0.26 | 66.52±0.73 | 75.18±0.43 | 34.31±3.12 | 72.69±0.49 | 86.83±0.63 | 71.92±1.55 | 81.35±1.39 | 46.93±3.27 | 78.98±2.06 |
| FairLoRA | C1 | 96.53±0.20 | 88.91±0.62 | 94.20±0.31 | 39.51±1.74 | 93.07±0.62 | 97.62±0.01 | 87.95±1.33 | 94.81±0.51 | 38.84±0.75 | 94.08±0.51 |
| | C2 | 87.24±0.36 | 66.16±0.68 | 75.15±0.76 | 47.44±1.86 | 71.57±0.99 | 90.77±0.29 | 78.25±0.27 | 84.46±1.37 | 39.90±7.12 | 81.80±1.75 |
| | C3 | 76.28±0.26 | 61.17±2.10 | 71.60±0.66 | 24.50±0.06 | 69.91±0.88 | 81.16±0.36 | 68.38±0.20 | 75.44±0.64 | 31.83±6.43 | 73.16±0.90 |
| | Avg. | 86.68±0.27 | 72.08±1.13 | 80.32±0.58 | 37.15±4.22 | 78.18±0.76 | 89.85±0.24 | 78.19±0.60 | 84.90±0.84 | 36.86±4.17 | 83.02±1.05 |
| FlexFair | C1 | 95.80±0.03 | 80.67±0.41 | 90.70±0.34 | 29.63±3.32 | 89.47±0.59 | 98.48±0.11 | 93.93±1.02 | 97.12±0.42 | 34.81±8.57 | 96.53±0.68 |
| | C2 | 81.40±0.42 | 64.04±1.90 | 71.84±1.88 | 45.16±7.36 | 68.44±2.42 | 91.37±1.31 | 78.22±3.47 | 84.79±3.19 | 50.86±11.82 | 81.92±4.24 |
| | C3 | 73.43±0.16 | 63.84±0.38 | 68.40±0.32 | 22.17±4.43 | 67.26±0.28 | 83.54±1.88 | 65.24±4.39 | 77.43±2.30 | 16.95±1.44 | 74.71±2.17 |
| | Avg. | 83.54±0.20 | 69.52±0.90 | 76.98±0.85 | 32.32±5.04 | 75.06±1.10 | 90.46±1.10 | 79.13±2.96 | 86.45±1.97 | 34.21±7.27 | 84.39±2.36 |
| *One-shot FL* (OFL) | | | | | | | | | | | |
| O-FedAvg | C1 | 94.74±0.05 | 76.79±0.20 | 87.99±0.07 | 34.02±2.95 | 85.83±0.20 | 94.16±1.19 | 76.56±3.12 | 88.92±1.87 | 49.99±8.56 | 87.72±2.07 |
| | C2 | 78.39±0.33 | 56.07±0.66 | 64.10±0.20 | 4.71±0.51 | 59.24±0.48 | 87.06±1.22 | 73.98±3.92 | 80.82±2.87 | 35.77±3.80 | 78.55±3.64 |
| | C3 | 70.56±0.16 | 57.05±0.64 | 64.39±0.49 | 10.19±1.43 | 62.25±0.61 | 66.40±3.05 | 45.33±4.95 | 48.26±7.33 | 28.60±2.37 | 39.70±9.13 |
| | Avg. | 81.23±0.18 | 63.31±0.50 | 72.16±0.26 | **16.31±1.63** | 69.11±0.43 | 82.54±1.82 | 65.29±4.00 | 72.67±4.02 | 38.12±4.91 | 68.65±4.95 |
| FuseFL | C1 | 94.70±0.17 | 76.29±0.61 | 87.80±0.15 | 39.46±2.26 | 85.19±0.07 | 92.71±0.17 | 74.14±0.61 | 84.58±0.39 | 50.23±11.74 | 81.33±0.46 |
| | C2 | 80.88±0.22 | 61.54±0.96 | 68.56±2.21 | 17.65±0.00 | 64.67±3.15 | 85.81±0.27 | 66.91±1.61 | 77.29±1.11 | 55.05±1.43 | 75.67±1.19 |
| | C3 | 70.32±0.36 | 58.13±1.38 | 64.12±1.58 | 15.59±0.99 | 61.96±2.39 | 71.60±0.71 | 58.11±2.19 | 65.60±1.66 | 43.94±1.07 | 63.04±2.35 |
| | Avg. | 81.97±0.25 | 65.32±0.98 | 73.49±1.31 | 24.23±1.08 | 70.61±1.87 | 83.37±0.38 | 66.39±1.47 | 75.83±1.05 | 49.74±4.75 | 73.35±1.33 |
| FAFI | C1 | 95.91±0.25 | 81.93±0.82 | 91.28±0.33 | 43.48±5.00 | 89.54±0.45 | 96.83±0.32 | 89.49±2.52 | 94.94±0.98 | 32.52±4.70 | 93.87±1.13 |
| | C2 | 85.21±0.66 | 61.26±1.28 | 72.20±1.57 | 29.47±6.00 | 67.02±1.69 | 92.55±0.08 | 83.09±0.62 | 88.36±0.46 | 36.97±5.05 | 86.71±0.73 |
| | C3 | 69.09±0.42 | 53.72±1.37 | 59.04±1.99 | 15.48±1.23 | 54.55±2.72 | 78.93±0.46 | 63.07±2.96 | 75.09±1.97 | 32.52±2.63 | 72.91±2.71 |
| | Avg. | 83.40±0.44 | 65.64±1.16 | 74.17±1.30 | 29.47±4.08 | 70.37±1.62 | 89.44±0.29 | 78.55±2.03 | 86.13±1.13 | 34.00±4.12 | 84.50±1.52 |
| **Ours** | C1 | 97.28±0.12 | 89.43±0.53 | 95.13±0.19 | 30.33±6.06 | 94.28±0.19 | 97.62±0.79 | 89.98±0.24 | 97.26±0.79 | 38.34±4.41 | 97.12±1.05 |
| | C2 | 87.23±0.12 | 68.93±1.58 | 79.59±1.24 | 45.34±1.78 | 77.04±1.65 | 93.68±0.62 | 83.00±5.85 | 89.97±3.02 | 22.77±4.03 | 88.40±1.99 |
| | C3 | 77.65±0.64 | 65.33±4.62 | 72.24±0.98 | 16.09±3.18 | 70.56±1.23 | 84.73±0.18 | 69.69±2.71 | 80.07±1.03 | 31.39±4.29 | 78.38±0.91 |
| | Avg. | **87.39±0.29** | **74.56±2.24** | **82.32±0.80** | 30.59±3.67 | **80.63±1.03** | **92.45±0.35** | **82.36±3.37** | **89.22±1.61** | 30.83±4.24 | **87.97±1.98** |

*Table 13.* Comprehensive per-client performance comparison on *Glaucoma detection* using RETFound and DINOv3 for Age and Gender sensitive attributes. **Bold** indicates the best overall results in average performance. Underline denotes the best average performance within Traditional FL or One-shot FL categories when the global best is not from Ours. We report mean ± std over 3 random seeds.

| Methods | Client | Age Attribute — RETFound AUC ↑ | ES-AUC ↑ | RES-AUC ↑ | DEOdds ↓ | Worst-Group AUC ↑ | Age — DINOv3 AUC ↑ | ES-AUC ↑ | RES-AUC ↑ | DEOdds ↓ | Worst-Group AUC ↑ | Gender — RETFound AUC ↑ | ES-AUC ↑ | RES-AUC ↑ | DEOdds ↓ | Worst-Group AUC ↑ | Gender — DINOv3 AUC ↑ | ES-AUC ↑ | RES-AUC ↑ | DEOdds ↓ | Worst-Group AUC ↑ |
|---|---|---|---|---|---|---|---|---|---|---|---|---|---|---|---|---|---|---|---|---|---|
| Local | C1 | 80.20±0.22 | 72.61±0.06 | 70.90±0.10 | 44.89±1.15 | 70.36±0.11 | 82.58±0.25 | 77.01±0.08 | 77.96±0.42 | 37.70±3.42 | 77.48±0.50 | 80.20±0.22 | 76.29±0.20 | 77.81±0.22 | 4.95±0.53 | 77.81±0.22 | 82.58±0.25 | 79.15±0.11 | 80.55±0.08 | 5.20±0.47 | 80.55±0.08 |
| | C2 | 79.67±0.08 | 74.50±0.09 | 75.94±0.19 | 39.41±0.79 | 75.63±0.22 | 80.89±0.23 | 75.37±0.78 | 76.18±1.07 | 35.44±1.00 | 75.90±1.05 | 79.67±0.08 | 78.01±0.11 | 78.73±0.07 | 4.09±1.10 | 78.73±0.07 | 80.89±0.23 | 78.79±0.64 | 79.71±0.41 | 3.92±0.33 | 79.71±0.41 |
| | Avg. | 79.94±0.15 | 73.55±0.08 | 73.42±0.14 | 42.15±0.97 | 73.00±0.17 | 81.73±0.24 | 76.19±0.43 | 77.07±0.74 | 36.57±2.21 | 76.69±0.77 | 79.94±0.15 | 77.15±0.15 | 78.27±0.15 | 4.52±0.82 | 78.27±0.15 | 81.73±0.24 | 78.97±0.38 | 80.13±0.25 | 4.56±0.40 | 80.13±0.25 |
| *Traditional FL (TFL)* | | | | | | | | | | | | | | | | | | | | | |
| FedAvg | C1 | 71.07±1.00 | 64.32±0.83 | 65.50±1.09 | 27.49±4.93 | 65.32±1.09 | 73.96±1.58 | 67.72±1.92 | 67.51±3.48 | 33.71±3.58 | 67.24±3.68 | 71.07±1.00 | 68.05±0.53 | 68.99±0.56 | 8.90±0.63 | 68.99±0.56 | 73.96±1.58 | 71.06±1.37 | 72.10±1.51 | 6.96±1.59 | 72.10±1.51 |
| | C2 | 69.48±0.99 | 63.42±1.09 | 64.84±0.72 | 27.48±0.43 | 64.57±0.69 | 71.76±0.47 | 66.73±0.52 | 67.99±0.90 | 26.52±3.15 | 67.71±1.00 | 69.48±0.99 | 68.70±0.91 | 68.96±0.89 | 2.30±0.49 | 68.96±0.89 | 71.76±0.47 | 70.38±1.22 | 70.93±0.92 | 3.72±1.80 | 70.93±0.92 |
| | Avg. | 70.28±1.00 | 63.87±0.96 | 65.17±0.91 | 27.48±2.68 | 64.95±0.89 | 72.86±1.02 | 67.23±1.22 | 67.75±2.19 | 30.12±3.37 | 67.48±2.34 | 70.28±1.00 | 68.38±0.72 | 68.98±0.73 | 5.60±0.56 | 68.98±0.73 | 72.86±1.02 | 70.72±1.29 | 71.51±1.21 | 5.34±1.70 | 71.51±1.21 |
| FairFed | C1 | 76.68±0.15 | 68.81±0.20 | 68.41±0.26 | 50.15±0.94 | 67.56±0.26 | 81.78±0.06 | 74.42±0.28 | 74.95±0.22 | 39.87±0.60 | 74.18±0.26 | 76.68±0.15 | 73.41±0.22 | 74.69±0.20 | 8.93±0.39 | 74.69±0.20 | 81.78±0.06 | 77.50±0.04 | 79.28±0.05 | 5.26±0.34 | 79.28±0.05 |
| | C2 | 76.11±0.15 | 69.64±0.19 | 71.25±0.21 | 36.27±0.61 | 70.91±0.22 | 79.46±0.06 | 73.33±0.18 | 73.40±0.21 | 36.18±1.36 | 73.15±0.23 | 76.11±0.15 | 75.34±0.18 | 75.68±0.17 | 2.70±0.25 | 75.68±0.17 | 79.46±0.06 | 77.40±0.16 | 78.33±0.12 | 5.89±0.08 | 78.33±0.12 |
| | Avg. | 76.40±0.15 | 69.23±0.20 | 69.83±0.23 | 43.21±0.77 | 69.24±0.20 | 80.62±0.06 | 73.88±0.23 | 74.18±0.22 | 38.03±0.98 | 73.66±0.24 | 76.40±0.15 | 74.38±0.20 | 75.18±0.19 | 5.82±0.32 | 75.18±0.19 | 80.62±0.06 | 77.45±0.10 | 78.80±0.08 | 5.58±0.21 | 78.80±0.08 |
| FairLoRA | C1 | 79.10±0.36 | 70.85±0.22 | 69.48±0.43 | 47.57±1.10 | 68.77±0.34 | 81.88±0.16 | 73.69±0.14 | 74.42±0.61 | 42.06±1.17 | 73.54±0.61 | 78.88±0.29 | 75.16±0.29 | 76.64±0.30 | 6.28±0.39 | 76.64±0.30 | 81.99±0.25 | 78.30±0.27 | 79.81±0.25 | 5.84±1.06 | 79.81±0.25 |
| | C2 | 78.09±0.28 | 71.73±0.43 | 73.06±0.55 | 42.92±0.82 | 72.83±0.50 | 78.50±0.07 | 73.10±0.25 | 72.76±0.01 | 35.63±2.28 | 74.39±0.15 | 78.04±0.19 | 77.36±0.18 | 77.66±0.18 | 1.98±0.76 | 77.66±0.18 | 80.40±0.22 | 78.45±0.34 | 79.34±0.26 | 4.67±0.66 | 79.34±0.26 |
| | Avg. | 78.59±0.32 | 71.29±0.33 | 71.27±0.49 | 45.24±0.96 | 70.80±0.42 | 81.19±0.12 | 74.26±0.10 | 74.63±0.36 | 38.85±1.72 | 73.97±0.38 | 78.46±0.26 | 76.26±0.24 | 77.15±0.24 | 4.13±0.58 | 77.15±0.24 | 81.19±0.23 | 78.38±0.31 | 79.58±0.26 | 5.26±0.76 | 79.58±0.26 |
| FlexFair | C1 | 80.24±2.63 | 73.21±3.10 | 72.32±3.01 | 41.59±4.55 | 71.83±3.23 | 80.49±0.26 | 73.07±0.24 | 73.25±0.20 | 43.77±0.91 | 72.49±0.16 | 76.63±0.05 | 73.35±0.08 | 74.64±0.04 | 7.34±1.32 | 74.64±0.04 | 83.11±0.08 | 79.23±0.16 | 80.84±0.11 | 5.50±0.48 | 80.84±0.11 |
| | C2 | 78.85±2.06 | 73.52±2.67 | 74.90±2.64 | 34.59±1.32 | 74.57±2.68 | 78.20±0.20 | 71.73±0.18 | 71.82±0.29 | 37.04±0.63 | 71.59±0.17 | 76.02±0.04 | 75.33±0.13 | 75.64±0.09 | 3.47±0.67 | 75.64±0.09 | 81.11±0.14 | 78.90±0.11 | 79.92±0.11 | 7.01±1.00 | 79.92±0.11 |
| | Avg. | 79.54±2.35 | 73.37±2.88 | 73.61±2.82 | 38.09±2.94 | 73.20±2.95 | 79.35±0.23 | 72.40±0.21 | 72.54±0.24 | 40.41±0.87 | 72.02±0.21 | 76.33±0.05 | 74.34±0.11 | 75.14±0.07 | 5.40±1.00 | 75.14±0.07 | 82.11±0.11 | 79.06±0.13 | 80.38±0.11 | 6.26±0.74 | 80.38±0.11 |
| *One-shot FL (OFL)* | | | | | | | | | | | | | | | | | | | | | |
| O-FedAvg | C1 | 73.15±0.02 | 65.08±0.10 | 63.34±0.04 | 50.71±0.54 | 62.50±0.02 | 76.03±2.50 | 68.43±2.80 | 67.94±3.39 | 43.94±0.48 | 67.36±3.34 | 73.15±0.02 | 70.31±0.04 | 71.33±0.04 | 4.41±0.93 | 71.33±0.04 | 76.03±2.50 | 71.30±2.83 | 73.01±2.81 | 7.25±1.07 | 73.01±2.81 |
| | C2 | 73.06±0.07 | 67.37±0.05 | 69.24±0.05 | 31.06±0.38 | 69.02±0.05 | 74.76±1.90 | 68.75±2.04 | 69.75±2.03 | 34.28±3.48 | 69.49±1.94 | 73.06±0.07 | 72.50±0.11 | 72.77±0.10 | 4.89±0.52 | 72.77±0.10 | 74.76±1.90 | 72.95±1.96 | 73.71±1.97 | 6.70±1.20 | 73.71±1.97 |
| | Avg. | 73.10±0.05 | 66.23±0.07 | 66.29±0.04 | 40.88±0.46 | 65.76±0.03 | 75.39±2.20 | 68.59±2.42 | 68.85±2.71 | 39.11±1.48 | 68.42±2.64 | 73.10±0.05 | 71.40±0.08 | 72.05±0.07 | 4.65±0.72 | 72.05±0.07 | 75.39±2.20 | 72.12±2.40 | 73.36±2.39 | 6.98±1.13 | 73.36±2.39 |
| FuseFL | C1 | 74.98±0.15 | 67.05±0.16 | 66.32±0.06 | 47.55±2.73 | 65.47±0.09 | 78.12±0.18 | 70.76±0.20 | 69.32±0.30 | 44.41±0.95 | 68.70±0.29 | 74.98±0.15 | 72.10±0.06 | 73.19±0.04 | 6.69±1.20 | 73.19±0.04 | 78.12±0.18 | 76.49±0.21 | 77.18±0.21 | 1.64±0.26 | 77.18±0.21 |
| | C2 | 74.48±0.02 | 68.44±0.14 | 70.31±0.10 | 33.31±0.68 | 70.04±0.11 | 76.52±0.03 | 70.72±0.22 | 71.42±0.49 | 33.68±0.94 | 70.98±0.52 | 74.48±0.02 | 73.40±0.08 | 73.84±0.07 | 3.88±0.38 | 73.84±0.07 | 76.52±0.03 | 75.43±0.09 | 75.92±0.06 | 3.35±0.34 | 75.92±0.06 |
| | Avg. | 74.73±0.08 | 67.74±0.15 | 68.32±0.08 | 40.43±1.71 | 67.76±0.10 | 77.32±0.10 | 70.74±0.21 | 70.37±0.39 | 39.04±0.95 | 69.84±0.41 | 74.73±0.08 | 72.75±0.07 | 73.52±0.05 | 5.28±0.79 | 73.52±0.05 | 77.32±0.10 | 75.96±0.17 | 76.55±0.14 | 2.50±0.30 | 76.55±0.14 |
| FAFI | C1 | 77.95±0.41 | 70.24±0.43 | 68.19±0.14 | 48.26±0.15 | 67.61±0.20 | 79.70±0.10 | 70.15±0.47 | 71.27±0.34 | 46.10±2.34 | 70.17±0.40 | 77.95±0.41 | 74.46±0.26 | 75.80±0.28 | 3.99±0.43 | 75.80±0.28 | 79.70±0.10 | 76.45±0.04 | 77.75±0.02 | 6.26±1.10 | 77.75±0.02 |
| | C2 | 76.22±0.39 | 69.64±0.41 | 70.55±0.28 | 42.53±2.25 | 70.13±0.30 | 79.12±0.05 | 72.60±0.14 | 72.69±0.13 | 40.62±0.65 | 72.30±0.09 | 76.22±0.39 | 75.18±0.35 | 75.62±0.36 | 3.97±1.36 | 75.62±0.36 | 79.12±0.05 | 77.43±0.15 | 78.20±0.11 | 3.94±1.30 | 78.20±0.11 |
| | Avg. | 77.08±0.40 | 69.94±0.42 | 69.37±0.23 | 45.40±1.20 | 68.87±0.25 | 79.41±0.07 | 71.38±0.31 | 71.98±0.24 | 43.36±1.50 | 71.23±0.20 | 77.08±0.40 | 74.82±0.30 | 75.71±0.32 | 3.98±1.10 | 75.71±0.32 | 79.41±0.07 | 76.94±0.09 | 77.98±0.06 | 5.10±1.20 | 77.98±0.06 |
| **Ours** | C1 | 82.26±0.01 | 75.63±0.04 | 74.89±0.02 | 40.96±0.72 | 74.41±0.03 | 83.58±0.34 | 78.40±0.18 | 80.18±0.15 | 34.47±4.13 | 79.92±0.07 | 82.17±0.12 | 78.48±0.36 | 79.91±0.27 | 5.30±0.15 | 79.91±0.27 | 83.65±0.25 | 80.60±0.30 | 81.85±0.22 | 3.40±0.06 | 81.85±0.22 |
| | C2 | 80.73±0.26 | 76.19±0.20 | 78.27±0.24 | 38.62±0.45 | 78.19±0.29 | 81.52±0.66 | 76.88±0.34 | 77.99±0.34 | 27.30±2.03 | 77.72±0.50 | 80.85±0.05 | 78.96±0.06 | 79.81±0.05 | 4.21±0.92 | 79.81±0.05 | 81.99±0.66 | 79.57±0.92 | 80.68±0.81 | 4.21±1.49 | 80.68±0.81 |
| | Avg. | **81.50±0.14** | **75.91±0.12** | **76.58±0.13** | **39.79±0.59** | **76.30±0.16** | **82.55±0.51** | **77.64±0.26** | **79.09±0.25** | **30.89±3.08** | **78.82±0.28** | **81.51±0.09** | **78.72±0.21** | **79.86±0.16** | 4.76±0.53 | **79.86±0.16** | **82.82±0.45** | **80.08±0.61** | **81.27±0.51** | 3.81±0.77 | **81.27±0.51** |

*Table 14.* Comprehensive per-client performance comparison on *Glaucoma detection* using RETFound and DINOv3 for intersectional sensitive attributes. **Bold** indicates the best overall results in average performance. Underline denotes the best average performance within Traditional FL or One-shot FL categories when the global best is not from Ours. We report mean ± std over 3 random seeds.

| Methods | Client | Age & Gender — RETFound AUC ↑ | ES-AUC ↑ | RES-AUC ↑ | DEOdds ↓ | Worst-Group AUC ↑ | AG — DINOv3 AUC ↑ | ES-AUC ↑ | RES-AUC ↑ | DEOdds ↓ | Worst-Group AUC ↑ | Age & Gender & Race — RETFound AUC ↑ | ES-AUC ↑ | RES-AUC ↑ | DEOdds ↓ | Worst-Group AUC ↑ | AGR — DINOv3 AUC ↑ | ES-AUC ↑ | RES-AUC ↑ | DEOdds ↓ | Worst-Group AUC ↑ |
|---|---|---|---|---|---|---|---|---|---|---|---|---|---|---|---|---|---|---|---|---|---|
| Local | C1 | 80.20±0.22 | 64.04±0.25 | 69.75±0.37 | 49.31±0.57 | 65.94±0.56 | 82.58±0.25 | 70.60±0.19 | 76.41±0.45 | 42.05±4.07 | 74.34±0.70 | 80.20±0.22 | 35.36±0.68 | 68.04±0.70 | 70.65±1.84 | 61.21±2.02 | 82.58±0.25 | 35.16±1.77 | 70.50±0.86 | 74.80±3.35 | 62.35±0.66 |
| | C2 | 79.67±0.08 | 70.07±0.24 | 75.44±0.26 | 42.13±1.21 | 74.81±0.25 | 80.89±0.23 | 75.37±0.44 | 76.07±0.92 | 37.72±4.42 | 74.91±1.19 | 79.67±0.08 | 38.14±0.54 | 68.21±0.43 | 72.71±4.42 | 60.02±1.52 | 80.89±0.23 | 38.39±0.41 | 68.39±1.64 | 68.91±0.08 | 59.09±4.36 |
| | Avg. | 79.94±0.15 | 67.06±0.24 | 72.60±0.31 | 45.72±0.89 | 70.37±0.41 | 81.73±0.24 | 69.97±0.21 | 76.24±0.68 | 39.89±3.25 | 74.62±0.95 | 79.94±0.15 | 36.75±0.61 | 68.13±0.57 | 71.68±3.13 | 60.62±1.77 | 81.73±0.24 | 36.78±1.09 | 69.44±1.25 | 71.85±6.22 | 60.72±2.51 |
| *Traditional FL (TFL)* | | | | | | | | | | | | | | | | | | | | | |
| FedAvg | C1 | 71.07±1.00 | 58.06±1.21 | 64.14±0.19 | 36.26±4.59 | 62.26±0.06 | 73.96±1.58 | 60.59±3.59 | 67.37±2.63 | 39.43±5.61 | 65.87±2.93 | 71.07±1.00 | 29.16±1.42 | 58.13±2.80 | 71.10±13.61 | 49.85±5.72 | 73.96±1.58 | 32.44±0.40 | 63.61±2.06 | 70.18±6.03 | 53.31±2.54 |
| | C2 | 69.48±0.99 | 58.39±1.76 | 64.30±0.84 | 29.13±0.51 | 63.15±0.61 | 71.76±0.47 | 62.35±1.09 | 67.33±1.00 | 30.16±2.56 | 66.71±1.25 | 69.48±0.99 | 33.60±2.10 | 60.80±1.75 | 57.01±4.96 | 54.08±3.82 | 71.76±0.47 | 36.87±3.72 | 62.89±4.03 | 56.46±7.13 | 55.56±7.38 |
| | Avg. | 70.28±1.00 | 58.22±1.48 | 64.22±0.52 | 32.69±2.55 | 62.71±0.64 | 72.86±1.02 | 61.47±2.30 | 67.35±1.82 | **34.79±4.08** | 66.29±2.09 | 70.28±1.00 | 31.38±1.76 | 59.47±2.28 | 64.05±9.29 | 51.97±4.77 | 72.86±1.02 | 34.66±2.06 | 63.25±3.05 | 63.32±6.58 | 54.44±4.96 |
| FairFed | C1 | 75.72±0.08 | 58.88±0.19 | 65.06±0.27 | 53.61±1.09 | 60.80±0.34 | 81.78±0.06 | 67.87±0.27 | 72.26±0.05 | 46.81±2.22 | 68.98±0.15 | 74.80±0.03 | 33.80±0.51 | 62.84±0.38 | 100.00±0.00 | 57.32±0.57 | 81.78±0.06 | 34.68±0.12 | 68.23±0.72 | 74.16±1.43 | 57.77±1.30 |
| | C2 | 75.36±0.15 | 63.99±0.30 | 70.21±0.26 | 36.93±0.85 | 69.35±0.31 | 79.46±0.06 | 67.23±0.55 | 73.26±0.20 | 40.13±0.54 | 71.45±0.38 | 74.64±0.09 | 32.95±0.10 | 62.73±0.43 | 63.25±2.62 | 52.27±0.61 | 79.46±0.06 | 37.22±0.42 | 68.81±0.51 | 57.08±0.80 | 63.23±0.79 |
| | Avg. | 75.54±0.11 | 61.44±0.25 | 67.64±0.25 | 45.27±0.97 | 65.07±0.32 | 80.62±0.06 | 67.55±0.41 | 72.76±0.13 | 43.47±1.38 | 70.22±0.26 | 74.72±0.06 | 33.37±0.30 | 62.79±0.36 | 81.63±1.31 | 54.80±0.39 | 80.62±0.06 | 35.95±0.27 | 68.52±0.61 | 65.62±1.11 | 60.50±1.04 |
| FairLoRA | C1 | 81.43±0.12 | 65.36±0.44 | 70.88±0.35 | 48.54±1.86 | 67.09±0.53 | 81.96±0.05 | 66.83±0.49 | 73.27±0.35 | 47.65±1.18 | 70.20±0.81 | 79.87±0.10 | 37.07±1.11 | 67.10±0.67 | 69.58±2.04 | 58.22±0.85 | 82.95±1.30 | 38.27±1.76 | 71.32±0.69 | 77.70±7.55 | 63.01±1.01 |
| | C2 | 79.98±0.21 | 69.71±0.51 | 76.03±0.43 | 42.93±2.49 | 75.44±0.53 | 80.44±0.06 | 68.33±0.71 | 74.76±0.17 | 41.15±1.30 | 75.38±0.47 | 78.72±0.00 | 34.54±0.43 | 66.63±0.46 | 79.64±2.30 | 58.83±2.20 | 81.59±1.35 | 37.82±2.46 | 70.66±0.76 | 55.94±1.62 | 63.69±0.24 |
| | Avg. | 80.71±0.16 | 67.54±0.49 | 73.46±0.39 | 45.74±2.18 | 71.27±0.53 | 81.20±0.05 | 67.58±0.60 | 74.02±0.26 | 44.40±1.74 | 71.79±0.49 | 79.30±0.06 | 35.81±0.79 | 66.86±0.76 | 74.61±2.33 | 58.52±1.53 | 82.27±1.32 | 38.04±1.76 | 70.99±0.72 | 66.82±4.59 | 63.35±0.63 |
| FlexFair | C1 | 82.05±0.04 | 66.11±0.96 | 73.06±0.09 | 41.78±1.80 | 69.63±0.18 | 83.14±0.07 | 70.13±0.68 | 74.51±0.48 | 43.57±1.59 | 71.56±0.48 | 76.64±0.05 | 33.84±0.40 | 64.17±0.70 | 92.80±5.37 | 57.25±2.15 | 83.14±0.07 | 35.77±1.03 | 69.77±0.34 | 68.58±5.04 | 60.94±1.29 |
| | C2 | 80.42±0.14 | 70.84±0.67 | 76.60±0.17 | 38.09±1.22 | 75.99±0.18 | 81.08±0.14 | 68.99±0.45 | 75.67±0.23 | 37.37±1.25 | 74.71±0.39 | 76.03±0.06 | 34.28±1.05 | 63.98±0.62 | 70.66±5.44 | 53.79±0.74 | 81.10±0.13 | 39.57±0.76 | 70.24±1.19 | 56.93±1.50 | 64.37±0.46 |
| | Avg. | 81.24±0.09 | 68.47±0.82 | 74.83±0.13 | 39.93±1.51 | 72.81±0.18 | 82.11±0.10 | 69.56±0.55 | 75.09±0.36 | 40.47±1.42 | 72.86±0.33 | 76.33±0.05 | 34.06±0.72 | 64.07±0.66 | 81.73±5.41 | 55.52±1.45 | 82.12±0.10 | 37.67±0.89 | 70.01±0.24 | **62.55±2.94** | 62.65±0.88 |
| *One-shot FL (OFL)* | | | | | | | | | | | | | | | | | | | | | |
| O-FedAvg | C1 | 73.15±0.02 | 57.48±0.04 | 61.79±0.05 | 52.98±0.74 | 57.41±0.05 | 76.03±2.50 | 60.82±3.14 | 66.17±3.21 | 52.90±0.60 | 62.46±3.34 | 73.15±0.02 | 27.87±0.35 | 57.11±0.60 | 100.00±0.00 | 47.22±1.57 | 76.03±2.50 | 35.29±2.79 | 65.22±3.46 | 69.48±2.25 | 58.57±3.99 |
| | C2 | 73.06±0.07 | 62.92±0.16 | 68.89±0.04 | 34.06±0.43 | 68.43±0.06 | 74.76±1.90 | 63.89±1.50 | 68.43±2.66 | 39.55±1.04 | 67.15±3.27 | 73.06±0.07 | 31.29±0.26 | 61.52±0.47 | 64.17±2.31 | 53.71±0.72 | 74.76±1.90 | 33.60±1.45 | 62.87±1.72 | 63.10±2.79 | 52.08±1.75 |
| | Avg. | 73.10±0.05 | 60.20±0.07 | 65.34±0.04 | 43.52±0.59 | 62.92±0.06 | 75.39±2.20 | 62.36±2.17 | 67.30±2.93 | 46.22±1.42 | 64.80±3.30 | 73.10±0.05 | 29.58±0.31 | 59.31±0.57 | 82.08±1.15 | 50.46±1.15 | 75.39±2.20 | 34.44±2.12 | 64.05±2.59 | 66.29±2.52 | 55.33±2.87 |
| FuseFL | C1 | 74.98±0.15 | 52.43±0.27 | 62.04±0.22 | 92.93±0.95 | 58.89±0.79 | 78.12±0.18 | 62.83±0.61 | 68.75±0.25 | 49.02±1.36 | 65.38±0.31 | 74.98±0.15 | 32.09±0.15 | 62.04±0.22 | 92.93±9.99 | 55.43±0.49 | 78.12±0.18 | 36.22±0.41 | 65.14±0.19 | 69.28±1.32 | 57.92±0.00 |
| | C2 | 74.48±0.02 | 50.52±1.18 | 63.34±0.11 | 70.66±2.62 | 64.88±1.90 | 76.52±0.03 | 65.52±0.71 | 71.32±0.32 | 37.64±1.24 | 70.54±0.29 | 74.48±0.02 | 34.47±0.66 | 63.34±0.11 | 70.66±2.62 | 52.44±0.66 | 76.52±0.03 | 33.41±0.78 | 64.45±0.76 | 79.63±2.62 | 53.41±1.91 |
| | Avg. | 74.73±0.08 | 51.48±0.52 | 62.69±0.17 | 81.80±6.31 | 61.88±1.34 | 77.32±0.10 | 64.17±0.66 | 70.04±0.28 | 43.33±1.30 | 67.96±0.30 | 74.73±0.08 | 33.28±0.40 | 62.69±0.17 | 81.80±6.31 | 53.94±0.57 | 77.32±0.10 | 34.82±0.66 | 64.79±0.45 | 74.46±1.97 | 55.67±0.95 |
| FAFI | C1 | 77.95±0.41 | 61.46±0.36 | 66.50±0.21 | 50.59±0.74 | 62.34±0.18 | 79.70±0.10 | 62.67±0.73 | 70.52±0.46 | 50.58±1.98 | 67.07±0.59 | 77.95±0.41 | 33.53±1.16 | 64.00±1.04 | 86.91±6.74 | 56.26±1.19 | 79.70±0.10 | 32.73±0.54 | 67.17±0.95 | 89.85±2.05 | 61.50±1.16 |
| | C2 | 76.22±0.39 | 64.59±0.42 | 69.69±0.38 | 46.56±3.23 | 67.88±0.60 | 79.12±0.05 | 65.85±0.27 | 73.12±0.15 | 45.06±0.71 | 71.93±0.20 | 76.22±0.39 | 34.50±0.43 | 65.90±0.50 | 69.80±2.22 | 60.33±0.73 | 79.12±0.05 | 38.58±0.39 | 69.00±0.46 | 59.51±1.51 | 62.70±0.37 |
| | Avg. | 77.08±0.40 | 63.02±0.31 | 68.10±0.30 | 48.57±1.99 | 65.11±0.39 | 79.41±0.07 | 64.26±0.50 | 71.82±0.30 | 47.82±1.35 | 69.50±0.39 | 77.08±0.40 | 34.02±0.79 | 64.95±0.77 | 78.35±4.48 | 58.30±0.96 | 79.41±0.07 | 35.66±0.42 | 68.09±0.70 | 74.68±1.78 | 62.10±0.77 |
| **Ours** | C1 | 82.85±0.16 | 67.92±0.16 | 74.84±0.05 | 44.35±0.64 | 71.89±0.13 | 83.57±0.49 | 72.46±0.80 | 78.03±0.10 | 42.22±5.49 | 76.13±1.15 | 82.24±0.16 | 39.25±0.93 | 72.59±0.50 | 70.00±2.36 | 64.71±1.22 | 83.67±0.63 | 38.52±1.01 | 73.81±0.52 | 67.61±6.34 | 66.65±1.40 |
| | C2 | 81.80±0.13 | 71.67±0.48 | 78.24±0.19 | 38.30±1.09 | 77.26±0.28 | 82.03±0.42 | 73.00±1.86 | 78.25±0.30 | 36.14±4.83 | 77.36±0.24 | 81.52±0.04 | 38.13±0.86 | 69.47±0.74 | 75.92±5.24 | 60.45±1.53 | 82.76±0.34 | 40.50±0.41 | 71.71±1.00 | 69.46±6.69 | 62.96±1.38 |
| | Avg. | **82.33±0.14** | **69.80±0.33** | **76.54±0.12** | 41.32±0.87 | **74.58±0.21** | **82.80±0.46** | **72.73±1.33** | **78.14±0.60** | 39.18±5.16 | **76.74±0.70** | **81.88±0.10** | **38.69±0.90** | **71.03±0.62** | 72.96±3.80 | **62.58±1.38** | **83.22±0.49** | **39.51±0.71** | **72.76±0.76** | 68.54±6.52 | **64.80±1.39** |

