# OpenReview forum: "Fair-FedMOE: Group-Fair One-Shot Federated Learning via Prototype-Guided Experts for Medical Imaging Analysis"
_ICML.cc/2026/Conference — ICML 2026 regular_

### Official Review · Reviewer_hQhf · 2026-03-04

**Soundness:** 3
**Presentation:** 4
**Significance:** 3
**Originality:** 3
**Overall Recommendation:** 4
**Confidence:** 4

**Summary:**

This paper addresses subgroup fairness in one-shot federated learning under heterogeneous data distributions, focusing on medical imaging tasks. The authors propose Fair-FedMOE, which uses sensitive-attribute prototypes to route samples to subgroup-specific classifier heads, allowing the model to learn specialized decision boundaries. To handle heterogeneity across clients, the method introduces a prototype-guided aggregation strategy with similarity-based weighting and masking to reduce conflicting updates. The paper also proposes a new fairness metric, RES-AUC, to evaluate worst-group performance across multiple attributes. Experiments on multi-institution datasets show improved subgroup fairness while maintaining competitive overall accuracy compared to existing baselines.

**Compliance With Llm Reviewing Policy:**

Affirmed.

**Key Questions For Authors:**

1. Is the model scalable? The evaluation uses a small number of clients. How does the method perform with larger and more diverse client populations?

2. How sensitive is the approach to noisy or imbalanced subgroup prototypes?

3. What motivated using hard routing instead of soft routing, and how does this affect stability and fairness?

4. Could the authors comment on the computational overhead of multiple experts and aggregation masks?

5. Do the authors expect the approach to generalize beyond medical imaging tasks?

**Limitations:**

Yes. The authors provide a clear discussion of limitations, including reliance on sensitive attribute annotations, predefined subgroup structures, and evaluation scope. They also acknowledge potential societal considerations related to fairness evaluation and deployment in real-world settings

**Strengths And Weaknesses:**

Soundness: The paper is generally technically sound, with clearly defined components and a coherent algorithmic pipeline. The empirical evaluation is thorough, including multiple datasets, backbones, ablations, routing analysis, and reporting of mean and standard deviation across seeds, which supports the main claims. The theoretical discussion of the RES-AUC metric also strengthens the contribution. However, the paper lacks theoretical analysis of convergence or fairness guarantees, and the evaluation involves a relatively small number of clients, limiting insight into large-scale behavior.

Presentation: The paper is clearly written and well structured, with a logical flow from motivation to method and experiments. Figures and visualizations effectively illustrate the framework and routing behavior. The discussion section appropriately acknowledges limitations. Some additional discussion of computational cost and scalability would further improve clarity.

Significance: The work addresses an important problem at the intersection of federated learning and fairness, particularly in medical imaging where subgroup disparities are critical. The proposed framework and fairness metric have practical relevance and could be useful for real-world deployments with limited communication rounds.

Originality: The paper presents a thoughtful integration of prototype-based routing, group-specific heads, and differential aggregation tailored to fairness in one-shot federated learning. While the overall framework is well designed and the metric contribution is meaningful, many individual components build on existing techniques, making the contribution more incremental than fundamentally new.

In general, the paper presents a well-motivated and carefully evaluated approach to improving subgroup fairness in FL. It combines clear practical relevance with solid empirical validation, while its main limitations are modest methodological novelty, limited client-scale evaluation, and lack of deeper theoretical analysis.

---

> ### Author Rebuttal · Authors · 2026-03-31
>
> > **Q1:** Is the model scalable? The evaluation uses a small number of clients. How does the method perform with larger and more diverse client populations?
>
> **Our Response:** The small client count in medical FL reflects inherent inter-institutional data sharing barriers, consistent with prior works (e.g., FairFed, FairFedMed). We conduct additional experiments on DR detection (RETFound-CFP, Age × Gender) under K=3,5,8,10 clients, where training sets for K>3 are split at the patient level with heterogeneous demographic distributions across clients. As shown in Table R3, Fair-FedMOE consistently outperforms FedAvg and O-FedAvg across all client counts. As K increases from 3 to 10, FedAvg's RES-AUC drops by 10.61% while Fair-FedMOE declines by only 2.84%, demonstrating the robustness of prototype-guided aggregation under increasing heterogeneity.
>
> Regarding client diversity, Tables 3 and 4 show that as the attribute space expands from Age × Gender to Age × Gender × Race, Fair-FedMOE maintains consistent improvements, owing to the linear scalability of prototype-guided expert routing (Eq. 2).
>
> **Table R3:** Performance comparison under varying client counts on DR detection.
> |Method|K=3 AUC|K=3 RES-AUC|K=5 AUC|K=5 RES-AUC|K=8 AUC|K=8 RES-AUC|K=10 AUC|K=10 RES-AUC|
> |:-|:-:|:-:|:-:|:-:|:-:|:-:|:-:|:-:|
> |FedAvg|79.29±1.71|71.43±2.06|76.57±0.15|66.15±0.66|75.38±3.29|63.21±5.29|72.44±1.67|60.82±2.60|
> |O-FedAvg|81.23±0.18|72.16±0.26|79.32±0.37|67.37±0.91|79.97±2.58|69.16±4.37|78.02±0.59|66.96±1.00|
> |Ours|87.39±0.29|82.32±0.80|86.64±0.38|82.19±0.96|87.01±0.33|82.23±0.78|85.11±0.56|79.48±0.82|
> > **Q2:** How sensitive is the approach to noisy or imbalanced subgroup prototypes?
>
> **Our Response:** For noisy prototypes, each prototype is initialized as the feature centroid over all subgroup samples (Eq. 1), naturally averaging out individual sample noise, and $\mathcal{L}_{pc}$ further refines prototype positions throughout training. At aggregation, clients with corrupted prototypes receive lower similarity-based weights (Eq. 4), with two masks (Eq. 5, 6) further filtering inconsistent updates.
>
> For imbalanced prototypes, as shown in Figure 6, our datasets exhibit notable imbalance across sensitive attributes, which becomes more pronounced in intersectional subgroups. Despite this, Fair-FedMOE achieves the best RES-AUC across all settings (Tables 3 and 4) and the most balanced group-wise AUC including minority subgroups (Figure 3).
> > **Q3:** What motivated using hard routing instead of soft routing, and how does this affect stability and fairness?
>
> **Our Response:** Hard routing is chosen because our core objective is fairness rather than overall performance. Hard routing assigns each sample to exactly one group-specific expert. Soft routing blends outputs from multiple experts, which blurs the feature boundaries between subgroups and allows majority-group representations to dominate.
>
> Regarding stability, the argmax in FER acts as a pure index selector and is excluded from the computational graph, so all trainable parameters receive valid gradient signals throughout training. Details are in our response to Reviewer Pmzm (Q2).
>
> As shown in Table R4, although soft routing achieves marginally higher overall AUC (+0.19%), it underperforms hard routing in RES-AUC by 0.87%. Since protecting the worst-off subgroup is the core objective rather than maximizing average performance, hard routing is the more appropriate design choice.
>
> **Table R4:** Ablation on routing strategy for DR detection.
> |Method|AUC|RES-AUC|
> |:-|:-:|:-:|
> |Soft Routing|**87.58±0.20**|81.45±0.34|
> |Hard Routing|87.39±0.29|**82.32±0.80**|
> > **Q4:** Could the authors comment on the computational overhead of multiple experts and aggregation masks?
>
> **Our Response:** For expert parameters, each expert is a single linear layer. Even in the most complex setting (Age × Gender × Race, 18 subgroups), all expert heads introduce only 8.21K parameters in total, negligible against the 303M ViT-Large backbone.
>
> For aggregation complexity, sign-consistency masking and prototype similarity computation both cost $\mathcal{O}(K^2 \times N)$ and $\mathcal{O}(K^2 \times d)$ respectively, involving pairwise operations across K clients. Sparse-overlap masking applies only to the lightweight expert parameters. All operations are executed once on the server, introducing no additional communication rounds.
> > **Q5:** Do the authors expect the approach to generalize beyond medical imaging tasks?
>
> **Our Response:** Fair-FedMOE is not specific to medical imaging and applies to any classification task where data is distributed across institutions with demographic disparities and fairness over sensitive attributes is required, such as credit scoring in financial services and item recommendation in recommender systems. The core design of prototype-guided expert routing and differential aggregation is task-agnostic, requiring only that each client holds sensitive attribute annotations.

---

### Official Review · Reviewer_ANNv · 2026-03-11

**Soundness:** 3
**Presentation:** 4
**Significance:** 2
**Originality:** 3
**Overall Recommendation:** 3
**Confidence:** 5

**Summary:**

This paper studies group-fair one-shot federated learning for medical imaging. The proposed method, Fair-FedMOE, combines fairness-aware expert routing, prototype-guided differential aggregation, and a new fairness metric, RES-AUC. At the client side, subgroup prototypes are used to route samples to group-specific experts. At the server side, prototype similarity and conflict-aware masking are used for personalized aggregation. The method is evaluated on diabetic retinopathy and glaucoma detection with RETFound and DINOv3, and compared against other FL baselines.

**Compliance With Llm Reviewing Policy:**

Affirmed.

**Final Justification:**

The rebuttal clarifies the paper’s scope, but the main concerns remain only partially resolved. Most importantly, the paper still does not position itself sufficiently against the closest MoE-based personalized FL literature. Because the proposed method relies on expert specialization and personalized aggregation, a clearer methodological distinction, together with stronger empirical positioning against the most relevant MoE-based FL baselines, is important for assessing novelty. At present, this aspect remains insufficiently convincing. In addition, the core one-shot aggregation mechanism is still largely heuristic, and the privacy implications of uploading subgroup-specific prototypes remain under-discussed. For these reasons, I keep the score.

**Key Questions For Authors:**

1.	It would be helpful if the paper could more clearly define its scope. Is the main claim about group-fair one-shot personalized FL in medical imaging, or about non-IID federated learning more broadly?
2.	The paper should clarify its relation to prior MoE-based personalized FL methods, especially pFedMoE and FedMoE. What is the precise methodological difference, and why are these not included in the comparison?
Reference such as:
[1] Yi, Liping, et al. "pFedMoE: Data-Level Personalization With Mixture of Experts in Model-Heterogeneous Personalized Federated Learning." IEEE Transactions on Knowledge and Data Engineering (2026).
[2] Mei, Hanzi, et al. "Fedmoe: Personalized federated learning via heterogeneous mixture of experts." arXiv preprint arXiv:2408.11304 (2024).
[3]Jiang, Jingang, et al. "Heterogeneous federated learning with scalable server mixture-of-experts." Proceedings of the Thirty-Fourth International Joint Conference on Artificial Intelligence. 2025.

3.	The paper would benefit from a clearer discussion of the theoretical status of the proposed aggregation scheme. At present, the theory appears to focus on RES-AUC, while the core one-shot aggregation design remains largely heuristic.
4.	Can the authors discuss the privacy implications of uploading group-specific prototypes to the server? Since the prototypes are learned from local subgroup data, it would be helpful to clarify whether they introduce additional privacy risks beyond standard parameter sharing.
5.	Were all baselines run under optimization and local training settings directly comparable to Fair-FedMOE? This is especially important since Fair-FedMOE introduces additional local regularization terms beyond the standard training objective.
6.	Given that all experiments are on retinal imaging with a small number of clients, how far do the authors believe the conclusions extend beyond this specific setting?
If the authors can clarify the paper’s scope, position it more carefully against the closest MoE-based FL literature, and strengthen the discussion of theory, privacy, and comparison fairness, I would be open to revising my score.

**Limitations:**

No. The limitations discussion is too light. It should more clearly acknowledge that the method assumes sensitive-attribute supervision during training, that the experiments are confined to a narrow medical setting, that the uploaded prototypes may raise privacy questions, and that the contribution is closer to a fairness-aware one-shot personalized FL design than to a broad solution for non-IID federated learning.

**Strengths And Weaknesses:**

Strengths:
1.	The paper addresses a meaningful problem. Group fairness is important in medical imaging, and the one-shot federated setting is practically relevant.
2.	The method is clearly structured. The local routing stage and the server aggregation stage are easy to follow.
3.	The empirical study is solid within the chosen setting. The paper reports both predictive and fairness metrics, includes intersectional subgroup analysis, and provides ablations.

Weaknesses:
1.	The paper’s scope is narrower than its framing suggests. It studies group-fair one-shot personalized FL with sensitive-attribute supervision, not non-IID federated learning in a broad sense.
2.	The methodological novelty is moderate. The overall contribution reads more as a coherent solution for a specific setting than as a sharply new FL principle. FER, PDA, and RES-AUC are each reasonable, but the paper does not yet make a strong case that the combination constitutes a substantial methodological advance for federated learning.
3.	The paper does not position itself clearly enough against the closest MoE-based personalized FL literature. This is particularly important given prior work such as pFedMoE and FedMoE, which also use expert specialization or MoE-style personalization under heterogeneous FL. Fair-FedMOE differs in its focus on fairness and one-shot aggregation, but that distinction is not discussed clearly enough, and these methods are not included in the empirical comparison.
4.	The theoretical depth is limited for the core algorithm. The paper provides theory for RES-AUC, but the central one-shot aggregation mechanism itself remains largely heuristic. Prototype-guided weighting, sign-consistency masking, and sparse-overlap aggregation are intuitively reasonable, but the paper does not analyze their optimization or generalization behavior under heterogeneity.
5.	The method uploads group-specific prototypes to the server, but the privacy implications of these prototypes are not analyzed.
6.	Would Fair-FedMOE still remain effective in more realistic clinical settings with a larger number of clients?

---

> ### Author Rebuttal · Authors · 2026-03-31
>
> > **Q1**: Clarify the paper's scope.
>
> **Our Response:** Our work focuses on group-fair one-shot personalized FL in medical imaging. Existing group-fair FL methods require multiple communication rounds, while existing OFL methods ignore demographic fairness entirely. No prior work addresses both simultaneously.
> > **Q2**: Clarify relation to pFedMoE et al. and why they are not included in comparison.
>
> **Our Response:** We will add a dedicated Related Work subsection. pFedMoE requires multi-round communication with heterogeneous architectures; FedMoE targets task heterogeneity in LLM fine-tuning; Fed-MoE requires a server dataset conflicting with medical privacy. Crucially, all three assign experts to clients/tasks rather than demographic subgroups, which is the core distinction making direct comparison inappropriate.
> > **Q3**: The core one-shot aggregation design remains largely heuristic; provide theoretical analysis.
>
> **Our Response:** Conventional FL requires iterative communication, impractical in medical settings with strict data governance, making OFL preferable. However, OFL forces long-term independent local training, and per-step gradient deviation $\|\nabla w\_u-\nabla w\_v\|^2>0$ accumulates over local training steps under data heterogeneity, producing conflicting updates that degrade aggregation and amplify subgroup disparity. We address this with: (1) sign-consistency regularization and masking on the feature extractor, reducing conflict rate $C$ to stabilize the global update direction; (2) sparse-overlap mask aggregation on expert networks, empirically reducing cross-client conflict on co-activated parameters; (3) prototype-guided aggregation weighting, using learned prototypes as distribution proxies to up-weight clients whose local distributions are more closely aligned with the target client.
>
> **1. Sign-consistency regularization and masking.** Let $C=\frac{1}{MN}\sum_i\min(k\_i,N-k\_i)$ be the overall conflict rate. Under naive averaging:
> $$\mathbb{E}[\Delta W\_g]=(1-2C)\mu,\quad\mathrm{Var}(\Delta W\_g)=\frac{1}{N}[\sigma^2+4C(1-C)\mu^2]$$
> As $C$ decreases, $|\mathbb{E}[\Delta W\_g]|$ increases and $\mathrm{Var}(\Delta W\_g)$ decreases, stabilizing the global update direction. $\mathcal{L}\_{\mathrm{sign}}$ aligns each client's updates toward the shared initialization direction, indirectly reducing $C$; the sign-consistency mask aggregates only sign-consistent parameters, removing conflicting updates from aggregation.
>
> **2. Sparse-overlap mask aggregation.** $\ell\_1$ regularization concentrates expert activations on high-SNR parameters. For co-activated parameters $\Omega\_i\odot\Omega\_j$, sign agreement is more likely to reflect genuine group-specific signal rather than noise, empirically yielding lower cross-client conflict.
>
> **3. Prototype–distribution correspondence.** When the feature extractor is approximately stable, the first-order optimality of $\mathcal{L}\_{\mathrm{pc}}$ implies:
> $$p^m\_{g,k} \\;\approx\\; \mathbb{E}\_{x\sim\mathcal{D}\_{g,k}}\left[\varphi(f\_\theta(x))\right]$$
> Thus $\cos(p^m\_{g,i}, p^m\_{g,j})$ measures the alignment of group-$g$ feature distribution means across clients, serving as a grounded proxy for inter-client distribution similarity in Eq.(4).
>
> **Experimental validation.** Table 6 ablations confirm each component: removing $\mathcal{L}\_{\mathrm{pc}}$, $\mathcal{L}\_{\mathrm{sign}}$, and $\mathcal{L}\_{\ell\_1}$ causes RES-AUC drops of 3.83%, 2.83%, and 2.73% respectively. Figure 9 shows routing accuracy exceeding 79%, validating the prototype–distribution correspondence. These analyses will be added to the revision.
> > **Q4**: Discuss privacy implications of uploading group-specific prototypes.
>
> **Our Response:** Unlike fixed centroids, prototypes optimized via $\mathcal{L}\_{pc}$ are more susceptible to inference attacks. We mitigate this via Differential Privacy on prototype transmission. Full details and empirical validation are in our response to Reviewer Pmzm (Q1).
> > **Q5**: Are all baselines run under settings directly comparable to Fair-FedMOE?
>
> **Our Response:** As stated in Appendix B.2, all baselines use identical local epochs, optimizer, and official codebases, ensuring a fair comparison. Fair-FedMOE's extra losses are not generic regularization but architecturally coupled. $\mathcal{L}\_{pc}$ requires group-specific experts and prototypes absent in baselines; $\mathcal{L}\_{sign}$ and $\mathcal{L}\_{\ell_1}$ are only meaningful with PDA's sign-consistency and sparse-overlap masks.
> > **Q6**: Given experiments on retinal imaging with few clients, how far do the conclusions extend beyond this setting?
>
> **Our Response:** Demographically annotated retinal datasets are scarce, explaining the small client counts typical in medical FL. To assess scalability, we conduct experiments on DR detection with K=3,5,8,10 clients, where Fair-FedMOE maintains consistent improvements over all baselines across all settings (Table R3, Reviewer hQhf Q1).

---

> > ### Author Rebuttal · Reviewer_ANNv · 2026-04-03
> >
> > Thanks for the rebuttal. For Q2, the relation to the closest MoE-based personalized FL literature is clearer, but still not fully convincing as a methodological distinction. For Q3, the added discussion improves the intuition, but the core one-shot aggregation mechanism remains largely heuristic. The risk of privacy leakage of prototypes may be added to the limitations. So I keep the score.

---

> > > ### Author Response · Authors · 2026-04-08
> > >
> > > > **Q2: methodological distinction still unconvincing**
> > >
> > > **Our Response:** Thanks for your question. Unlike pFedMoE, FedMoE, and Fed-MoE, which construct experts at client or task level, Fair-FedMOE explicitly dedicates experts to demographic subgroups.
> > >
> > > Specifically, **pFedMoE** constructs a two-expert MoE per client to balance global and local knowledge, where samples from different demographic groups share identical experts, as demographic fairness is not considered. **FedMoE** selects expert subsets per client based on activation frequencies to address task-level heterogeneity, where routing is conditioned on task-level rather than demographic distributions. **Fed-MoE** constructs a server-side MoE by aggregating client models via a correlation matrix, remaining a client-level assignment, and requires a server-reserved dataset that conflicts with medical privacy.
> > >
> > > In all three methods, samples from different demographic groups are processed by shared experts without demographic awareness, inevitably biasing toward majority groups and failing to achieve group fairness. Furthermore, none of these methods have released official open-source code. Fair-FedMOE addresses this directly through $\mathcal{L}_{pc}$, which enforces group-level expert specialization via prototype supervision, ensuring each expert captures subgroup-specific features and prevents inter-group interference.
> > > > **Q3: OFL aggregation mechanism remains heuristic**
> > >
> > > **Our Response:** Thanks for your question. Each component of PDA has a clear optimality criterion. In OFL, parameter conflicts arising from data heterogeneity cannot be corrected after aggregation. These conflicts introduce two distinct challenges.
> > >
> > > **Parameter conflict in the feature extractor.** Clients fine-tune from the same initialization but converge to different local optima, producing updates with opposite signs. Let $C=\frac{1}{MN}\sum_i \min(k_i,N-k_i)$ denote the conflict rate, where $k_i$ is the number of clients with a positive update on parameter $i$ and $M$ is the total number of parameters. Assuming parameter update magnitudes follow a distribution with mean $\mu$ and variance $\sigma^2$:
> > > $$\mathbb{E}[\Delta W_g]=(1-2C)\mu,\quad\text{Var}(\Delta W_g)=\frac{1}{N}[\sigma^2+4C(1-C)\mu^2]$$
> > > As $C$ decreases, $|\mathbb{E}[\Delta W_g]|$ increases and $\text{Var}(\Delta W\_g)$ decreases, stabilizing the update direction. Since all clients share $\theta^{(0)}$, minimizing $\mathcal{L}\_\text{sign}$ constrains each client $k$ to satisfy $\text{sign}(\theta_{k,i})\approx \text{sign}(\theta^{(0)}\_i)$ for most $i$, making $\text{sign}(\Delta\theta_{k,i})$ consistent across clients. This drives $\min(k_i,N-k_i) \to 0$ for most parameters, effectively reducing $C$ relative to unconstrained training. The sign-consistency mask then eliminates residual conflicts.
> > >
> > > **Noise in group-specific experts.** Each client has limited samples per subgroup, making expert networks susceptible to local overfitting. Direct averaging mixes group-specific signals with client-specific noise, undermining expert specialization. Under $\ell_1$​ regularization, parameters are driven toward sparsity; those below threshold $\tau_s$​ in one client carry no reliable signal by definition. The sparse-overlap mask $\Omega_i\odot\Omega_j$​ retains only co-activated parameters, which under $\ell\_1$​ are precisely those carrying consistent group-specific signal across clients.
> > >
> > > **Principled aggregation weights.** In OFL setting, iterative distribution estimation is infeasible. At the first-order optimality of $\mathcal{L}\_{pc}$ (Eq.7), setting $\frac{\partial \mathcal{L}\_{pc}}{\partial p^m_g}=0$ yields:
> > > $$p^m\_{g,k} \approx\mathbb{E}\_{x\sim\mathcal{D}\_{g,k}}[\varphi(f\_\theta(x))]$$
> > > Each prototype thus approximates the feature distribution mean of its corresponding subgroup, so $\cos(p^m_{g,i},p^m_{g,j})$ measures the alignment of group-level feature distributions across clients. The aggregation weights in Eq.(4) thus up-weight clients whose local distributions are more aligned with the target.
> > >
> > > We will formalize the conflict-rate reduction under $\mathcal{L}\_\text{sign}$​ and the prototype–distribution correspondence as two Proposition with complete proof in the revision.
> > > > **Limitation: prototype privacy leakage not discussed**
> > >
> > > **Our Response:** We agree and will add to Appendix G.2:
> > >
> > > **Prototype Privacy Risk.** Prototypes optimized via $\mathcal{L}_\text{pc}$ encode richer demographic information than fixed centroids, making them more susceptible to inference attacks. To mitigate this, we apply DP to prototype transmission: prototypes are $\ell_2$-normalized before transmission, and Gaussian noise $\mathcal{N}(0,\sigma^2 I)$ with $\sigma=\frac{2\sqrt{2\ln(1.25/\delta)}}{\epsilon}$, $\delta=10^{-5}$ yields a formal $(\epsilon,\delta)$-DP guarantee. Empirically, as shown in Table R2, utility loss remains marginal even at $\epsilon=1$, where Fair-FedMOE maintaining superior RES-AUC over all baselines.

---

### Official Review · Reviewer_Pmzm · 2026-03-13

**Soundness:** 3
**Presentation:** 4
**Significance:** 3
**Originality:** 3
**Overall Recommendation:** 5
**Confidence:** 4

**Summary:**

The authors present Fair-FedMOE, a group-fair One-Shot Federated Learning (OFL) framework tailored for fine-tuning medical foundation models. It addresses two main challenges—subgroup performance disparity and model inconsistency—through a dual-stage approach. During local training, a Fairness-aware Expert Routing (FER) module directs samples to demographic-specific experts. During server aggregation, a Prototype-guided Differential Aggregation (PDA) module filters conflicting updates using module-specific masks. Additionally, the paper introduces a new bounded fairness metric, the Rawlsian Equity-Scaled AUC (RES-AUC), to evaluate worst-group performance effectively. Evaluations on diabetic retinopathy and glaucoma datasets demonstrate performance and equity improvements using RETFound and DINOv3 backbones.

**Compliance With Llm Reviewing Policy:**

Affirmed.

**Final Justification:**

I increased my score to 5, given the authors' rebuttal efforts.

**Key Questions For Authors:**

Can you provide mathematical or cryptographic guarantees (e.g., Differential Privacy perturbations) to secure the transmitted demographic prototypes against gradient or Model Inversion attacks?

Could you explicitly define the computational graph and backpropagation mechanism used to traverse the non-differentiable hard $\arg\max$ routing step in the FER module?

How does the sparse-overlap mask handle extreme Non-IID distributions where a local client might have zero instances of a specific demographic, leaving the corresponding expert uninitialized? Have you observed severe underfitting caused by the Sign-Consistency Loss when adapting to highly out-of-distribution textures?

**Limitations:**

Yes.

**Strengths And Weaknesses:**

Strength

Strong empirical results on relevant multi-modal medical datasets. The framework evaluates on multiple DR and glaucoma benchmarks (e.g., BRSET, Harvard-FairVLMed), reporting consistent improvements in both overall AUC and the proposed RES-AUC compared to traditional and one-shot federated baselines.

Clear algorithmic innovation in one-shot federated aggregation. The Prototype-guided Differential Aggregation (PDA) smartly tackles parameter conflicts that arise from data heterogeneity by applying module-specific masking (sign-consistency and sparse-overlap masks), effectively preserving the representations of pre-trained foundation models without multiple communication rounds.

Rigorous mathematical formulation of the new fairness metric. The RES-AUC effectively resolves the unbounded variance penalty issue present in the existing ES-AUC metric by anchoring to worst-group performance and utilizing Popoviciu's Inequality. This provides a highly stable and much-needed evaluation standard for highly intersectional demographics in medical imaging.

Weakness

Privacy leakage risks via prototype transmission. The framework explicitly transmits demographic-specific feature centroids (prototypes) to the central server to compute aggregation weights. In highly expressive foundation models, these dense feature centroids are susceptible to Model Inversion (MI) attacks, which risks the reconstruction of sensitive patient biometrics and weakens the privacy-preserving claims of the federated setting.

Missing computational graph details for hard routing. The routing mechanism within the FER module utilizes a non-differentiable $\arg\max$ operation to assign samples to experts. The paper omits the mathematical explanation of how gradients flow through this discrete step during backpropagation, which raises concerns about local optimization stability and limits reproducibility.

Over-regularization and expert collapse concerns. The framework relies on a Sign-Consistency Loss that penalizes parameter updates that flip signs relative to initialization, which may restrict the model's ability to adapt to severe domain shifts. Furthermore, under extreme Non-IID conditions where a local client might entirely lack samples for a specific demographic, it is unclear how the corresponding untrained expert is prevented from degrading the global model during sparse-overlap aggregation.

---

> ### Author Rebuttal · Authors · 2026-03-31
>
> > **Q1:** Can you provide mathematical or cryptographic guarantees (e.g., Differential Privacy perturbations) to secure the transmitted demographic prototypes against gradient or Model Inversion attacks?
>
> **Our Response:** To address the privacy leakage risk, we integrate a Differential Privacy (DP) mechanism into prototype transmission. Specifically, prototypes are $\ell_2$-normalized at each forward pass, guaranteeing $\|\bar{p}^m_g\|\_2 = 1$ and bounding the $\ell_2$-sensitivity at $\Delta_f = 2$ throughout training. Before transmission, each client applies the Gaussian Mechanism [Dwork & Roth, 2014]:
> $$\tilde{p}^m_g = \bar{p}^m\_g + \mathcal{N}(0, \sigma^2 I), \quad \sigma = \frac{2\sqrt{2\ln(1.25/\delta)}}{\epsilon}$$
> where $\delta = 10^{-5}$, yielding a formal $(\epsilon, \delta)$-DP guarantee, which provides a formal bound on information leakage under the DP framework. Since PDA relies on cosine similarity, which depends solely on vector direction, it is inherently robust to the zero-mean isotropic noise above and incurs negligible utility loss. Table R2 confirms that Fair-FedMOE maintains superior RES-AUC over all baselines even under the strict budget of $\epsilon = 1$, with only a marginal drop compared to the No-DP setting.
>
> **Table R2:** Performance of Fair-FedMOE under varying privacy budgets ($\epsilon$) on DR detection (RETFound, Age × Gender).
> |Methods|AUC (%)|RES-AUC (%)|
> |:-|:-:|:-:|
> |FedAvg|79.29±1.71|71.43±2.06|
> |FairFed|82.39±0.26|75.18±0.43|
> |FairLoRA|86.68±0.27|80.32±0.58|
> |FlexFair|83.54±0.20|76.98±0.85|
> |O-FedAvg|81.23±0.18|72.16±0.26|
> |FuseFL|81.97±0.25|73.49±1.31|
> |FAFI|83.40±0.44|74.17±1.30|
> |Ours ($\epsilon=1$)|87.12±0.29|81.88±0.99|
> |Ours ($\epsilon=5$)|87.29±0.20|81.99±1.17|
> |Ours ($\epsilon=10$)|87.38±0.31|82.23±1.18|
> |Ours (Non DP)|**87.39±0.29**|**82.32±0.80**|
>
> Dwork C, Roth A. The algorithmic foundations of differential privacy[J]. Foundations and trends® in theoretical computer science, 2014, 9(3-4): 211-487.
> > **Q2:** Could you explicitly define the computational graph and backpropagation mechanism used to traverse the non-differentiable hard $arg max$ routing step in the FER module?
>
> **Our Response:** We clarify that the hard argmax routing in FER does not block gradient flow during backpropagation. The computational graph contains two fully independent gradient paths. For the expert network, argmax acts as a pure index selector to identify the activated expert $E^m\_{g^\*}$ and is excluded from the computational graph, so gradients flow through $E^m_{g^\*}(h)$ normally via standard backpropagation. For the prototype parameters $\mathcal{P}^m$, they are optimized via $\mathcal{L}\_{pc}$ (Eq. 7) using ground-truth attribute labels $a^m_i$, which directly computes the contrastive loss over $h^\top p^m\_g$, bypassing argmax entirely. The two paths are completely decoupled, so all trainable parameters receive valid gradient signals throughout training.
> > **Q3:** How does the sparse-overlap mask handle extreme Non-IID distributions where a local client might have zero instances of a specific demographic, leaving the corresponding expert uninitialized? Have you observed severe underfitting caused by the Sign-Consistency Loss when adapting to highly out-of-distribution textures?
>
> **Our Response:** We thank the reviewer for raising these concerns.
>
> **For Issue 1 (sparse-overlap mask under extreme Non-IID)**: In practice, as shown in Figure 6, all clients cover every demographic subgroup in our experimental setting, so this extreme case does not arise. Theoretically, if a client entirely lacks subgroup g, $E^m_g$ receives no gradient updates and remains at random initialization. The affected client produces a prototype deviating from the shared semantic space, receiving a lower $W\_{ij}$ (Eq. 4), which reduces its overall aggregation contribution and indirectly attenuates the influence of the untrained expert on the global model. Explicit expert-level exclusion under this extreme case is left for future work.
>
> **For Issue 2 (Sign-Consistency Loss and over-regularization)**: $\mathcal{L}\_{sign}$ is a soft constraint that only influences aggregation mask selection; parameters still fully participate in the forward pass. Moreover, our framework fine-tunes ViT-based foundation models (RETFound and DINOv3), which inherit the inherent robustness of ViT's global self-attention to data heterogeneity and distribution shifts [Qu et al., 2022], further strengthened by large-scale pretraining, providing an additional safeguard against underfitting under $\mathcal{L}\_{sign}$. Two results directly rule out over-regularization: (1) C3 (ODIR-5K), our most heterogeneous client, achieves the largest per-client improvement (Table 7); (2) removing $\mathcal{L}\_{sign}$ causes a RES-AUC drop (Table 6), not a gain.
>
> Qu L, Zhou Y, Liang P P, et al. Rethinking architecture design for tackling data heterogeneity in federated learning[C]//CVPR. 2022: 10061-10071.

---

> > ### Author Rebuttal · Reviewer_Pmzm · 2026-04-06
> >
> > Thank you for addressing my comments.

---

> > > ### Author Response · Authors · 2026-04-08
> > >
> > > We sincerely thank the reviewer for the thorough engagement. The feedback on privacy guarantees, hard routing gradient propagation, expert initialization under extreme Non-IID distributions, and sign-consistency regularization risk has been invaluable in improving our work.

---

### Official Review · Reviewer_QsB9 · 2026-03-17

**Soundness:** 3
**Presentation:** 3
**Significance:** 3
**Originality:** 3
**Overall Recommendation:** 6
**Confidence:** 5

**Summary:**

This paper discusses how to achieve equitable performance across demographic groups when fine‑tuning medical foundation models in a one‑shot federated learning (OFL) setting. The authors observe that naïve OFL fine‑tuning can exacerbate subgroup performance disparities because local models diverge and their parameters conflict during aggregation.They introduce Fairness‑aware Expert Routing and Prototype‑guided Differential Aggregation to address this issue.

**Compliance With Llm Reviewing Policy:**

Affirmed.

**Final Justification:**

After reading the rebuttal, I am comfortable raising to Accept recommendation.

My main initial concerns were about whether the paper’s two central technical intuitions were sufficiently supported: (i) whether prototype-space similarity is actually informative for identifying useful aggregation partners, and (ii) whether the sign-based masking removes harmful conflicts rather than merely shrinking effective capacity. The rebuttal addresses by including correlation between prototype similarity and merged-model utility, and an analysis showing substantial information loss among sign-inconsistent updates under naive aggregation.
I also appreciate the clarification regarding sensitive attributes. The method does require subgroup labels during training for prototype supervision, but these labels are used only locally and are not required at inference time, where routing is based on feature–prototype similarity. The authors also clearly acknowledge incomplete or missing subgroup labels as a limitation, and I hope AC can verify if this is included in the final version's limitation section.

**Key Questions For Authors:**

discussed above.

**Limitations:**

yes

**Strengths And Weaknesses:**

Strengths:

The problem is timely and important. The paper identifies a real gap at the intersection of fairness and one-shot FL, especially for medical imaging domain.

Well motivated method and clearly explained, shows strong empirical gains.

The RES-AUC proposal is interesting and useful.

Weakness:

1 The paper motivates why prototype similarity should reflect client relatedness and why sign consistency should reduce harmful conflicts, but these claims are mostly intuitive. I would have liked either stronger theory or more direct empirical evidence showing that prototype-space similarity actually correlates with useful aggregation partners, and that sign masking removes specifically harmful conflicts rather than simply reducing effective model capacity.

2 The paper notes that inference does not require sensitive attributes because routing is feature-similarity-based, but doesn't your training pipeline still rely on subgroup labels for prototype supervision? I need more explanation on this. In real deployments, this may limit applicability, especially where demographic labels are incomplete, noisy, or sensitive to use.

3 I usually hate reviews like "need more baselines" but here i have to say : Although the experiments span two tasks and two backbones, everything is retinal imaging. Since the paper claims a broader framework for “medical FMs,” it would be stronger with at least one non-retinal setting or a more careful claim scope. I think maybe run a quick one in radiology or breast cancer imaging would be good. I would still rate soundness as good, but more will make it excellent.

4 The motivation of grad/ update conflict -> moe as a solution was to my knowledge first discussed and introduced in multi-task learning. Please refer to those works as acknowledgement. Example would be CVPR 24: Mod-Squad: Designing Mixtures of Experts As Modular Multi-Task Learners, ICLR 25: Dynamic Modeling of Patients, Modalities and Tasks via Multi-modal Multi-task Mixture of Experts.

---

> ### Author Rebuttal · Authors · 2026-03-31
>
> > **W1:** I would have liked theory or more direct empirical evidence that prototype-space similarity correlates with useful aggregation partners, and that sign masking removes specifically harmful conflicts rather than simply reducing effective model capacity.
>
> **Our Response:** We provide direct empirical evidence for both claims (Figure R1).
>
> **For Claim 1 (prototype-space similarity)**: We directly average the parameters of two clients and evaluate AUC on both test sets (Figure 1(a)). Prototype cosine similarity perfectly correlates with the merged model AUC across all client pairs (Spearman $\rho=1.000$), higher similarity consistently yields better post-merge performance, confirming that prototype space effectively captures inter-client distributional relatedness.
>
> **For Claim 2 (sign masking)**: Sign consistency is defined over parameter updates relative to the shared initialization. When $\text{sign}(\Delta\_i)\neq \text{sign}(\Delta\_j)$, two clients pull the same parameter in opposite directions from the same starting point; weighted averaging only cancels both. We quantify this on models trained without $\mathcal{L}\_{sign}$:
> $$\text{Info Loss} = 1-\frac{|\Delta\_i+\Delta\_j|}{|\Delta\_i|+|\Delta\_j|}$$
> where $|·|$ denotes the element-wise absolute value, computed independently for each parameter coordinate. Sign-inconsistent parameters yield Info Loss of $0.521$–$0.542$ (Figure R1(b)), meaning over half of their update information is destroyed upon aggregation; excluding them therefore preserves rather than reduces effective model capacity. In the OFL setting, information lost at aggregation is unrecoverable. $\mathcal{L}\_{sign}$ further anchors each client's updates to $\text{sign}(\theta^{(0)})$, increasing the fraction of sign-consistent parameters retained by the mask. Table 6 corroborates: removing $\mathcal{L}\_{sign}$ drops RES-AUC by $2.83\%$.
>
> Figure R1: (a) Prototype cosine similarity vs. naive-average model AUC across three client pairs. (b) Mean information loss of sign-inconsistent parameters trained without $\mathcal{L}_{sign}$. (Link: https://anonymous.4open.science/r/R1-1-5644/R1-1.png)
> > **W2:** The paper notes that inference does not require sensitive attributes because routing is feature-similarity-based, but doesn't your training pipeline still rely on subgroup labels for prototype supervision?
>
> **Our Response:** We acknowledge that training requires sensitive attribute labels, a common assumption in group-fair FL methods (e.g., FairFedMed). Labels are used only locally and never transmitted to the server; inference relies solely on feature-prototype similarity. For partial missing or noisy labels, the prototype constraint loss naturally skips unlabeled samples, and the feature-space clustering structure induced by $\mathcal{L}_{pc}$ provides some tolerance to label noise. For systematic label absence (e.g., an entire subgroup lacks annotations), prototype initialization becomes unreliable, which we acknowledge as a current limitation. Addressing this via unsupervised subgroup discovery is planned as future work, as discussed in Appendix G.2.
> > **W3:** Since the paper claims a broader framework for “medical FMs,” it would be stronger with at least one non-retinal setting or a more careful claim scope.
>
> **Our Response:** We thank the reviewer for this constructive suggestion. We agree that a non-retinal validation would strengthen the paper's claim scope. Motivated by the reviewer's suggestion, we are currently running experiments on CheXpert [Irvin et al., 2019], a large-scale chest X-ray dataset with demographic annotations (age, gender), with 5 clients and DINOv3 as the backbone. We look forward to sharing the complete results in the final rebuttal during the discussion period.
>
> In addition, we have conducted scalability experiments on DR detection (K=3,5,8,10 clients, Table R3, Reviewer hQhf Q1), where Fair-FedMOE consistently outperforms FedAvg and O-FedAvg across all client counts, further demonstrating the robustness of our framework beyond the original experimental setting.
>
> Irvin J, Rajpurkar P, Ko M, et al. Chexpert: A large chest radiograph dataset with uncertainty labels and expert comparison[C]//AAAI. 2019, 33(01): 590-597.
> > **W4:** The motivation of gradient/update conflict→MoE as a solution was to my knowledge first discussed and introduced in multi-task learning.
>
> **Our Response:** We thank the reviewer for pointing out these relevant works. We acknowledge that the use of MoE to address update conflicts in heterogeneous settings has been discussed in multi-task learning [Chen et al., 2024; Wu et al., 2025]. We will add citations to these works in Section 3.3 of the revised version.
>
> Chen Z, Shen Y, Ding M, et al. Mod-squad: Designing mixtures of experts as modular multi-task learners[C]//CVPR. 2023: 11828-11837.
>
> Wu C, Shuai Z, Tang Z, et al. Dynamic modeling of patients, modalities and tasks via multi-modal multi-task mixture of experts[C]//ICLR. 2025.

---

> > ### Author Rebuttal · Reviewer_QsB9 · 2026-04-04
> >
> > my concerns fully addressed. if time allows can authors please gimme some additional analysis on the figs (right side ) u appended in rebuttal by public comment?
> >
> > thanks!

---

> > > ### Author Response · Authors · 2026-04-08
> > >
> > > Thank you for the positive feedback and the thoughtful suggestion. We provide additional analysis on Figure R1(b) as requested.
> > >
> > > The bar chart shows the mean information loss of sign-inconsistent parameters when trained without $\mathcal{L}\_{\text{sign}}$. All three client pairs yield information loss above 0.5. This holds across client pairs with substantial differences in imaging devices and data distributions, suggesting that sign-inconsistency is a persistent phenomenon across diverse client configurations, rather than an artifact of any specific dataset pairing.
> > >
> > > Without $\mathcal{L}\_{\text{sign}}$, sign-inconsistent parameters carry severely degraded information after naive averaging. Since OFL involves only a single communication round, this loss cannot be corrected through subsequent iterations. $\mathcal{L}\_{\text{sign}}$ penalizes updates that conflict with $\text{sign}(\theta^{(0)})$, reducing the proportion of sign-inconsistent parameters and thus mitigating this issue. Table 6 corroborates this, where removing $\mathcal{L}\_{\text{sign}}$ drops RES-AUC by 2.83%.
> > >
> > > As promised, the CheXpert experiments are now complete, and we present the results below. We will include these experiments in the appendix of the revised paper.
> > >
> > > To validate generalizability beyond retinal imaging, we extend experiments to CheXpert chest X-rays, which differ in modality, acquisition protocol, and disease domain. We choose two tasks with complementary characteristics: Pleural Effusion (PE) has clear morphological boundaries and balanced labels (positive:negative = 75:25, U-Ignore); Atelectasis (AT) involves subtle structural changes with prevalent label uncertainty (positive:negative = 98:2, U-Ones per the original CheXpert benchmark), stress-testing FER under noisy supervision. Both tasks use patient-level 6:2:2 splits and 5 demographically heterogeneous clients over Age $\times$ Gender subgroups. Dataset statistics and cross-client subgroup distributions are shown in Figure R2.
> > >
> > > Table R3 presents results on CheXpert under Age $\times$ Gender using DINOv3, where Local denotes training on the complete dataset prior to federated partitioning. Fair-FedMOE achieves comparable AUC while obtaining higher RES-AUC on both tasks, indicating better subgroup fairness without sacrificing overall performance. Against TFL and OFL baselines, Fair-FedMOE achieves the best AUC and RES-AUC across both tasks. On PE detection, RES-AUC improves by +0.53% over the best TFL method and +2.17% over the best OFL method. On AT detection, where the dataset contains noisy labels introduced by the U-Ones strategy, Fair-FedMOE remains robust, outperforming the best TFL and OFL methods by +1.02% and +2.12% in RES-AUC respectively. These results demonstrate that Fair-FedMOE generalizes effectively to chest radiography tasks.
> > >
> > > Figure R2: Dataset statistics for PE and AT detection tasks on CheXpert across 5 federated clients. (a) PE: U-Ignore labeling strategy. (b) AT: U-Ones labeling strategy. Heatmaps show cross-client demographic subgroup heterogeneity. (Link: https://anonymous.4open.science/r/CheXpert/chexpert_dataset_stats.png)
> > >
> > > Table R5. Performance comparison on PE and AT detection tasks on CheXpert using DINOv3, evaluated under Age × Gender intersectional attributes. Bold indicates the best results. We report representative baselines from each category for brevity.
> > > |Methods|PE Detection||AT Detection||
> > > |:-|:-:|:-:|:-:|:-:|
> > > ||AUC|RES-AUC|AUC|RES-AUC|
> > > |Local|0.9556|0.9468|0.7137|0.6500|
> > > |*Traditional FL (TFL)*
> > > |FedAvg|0.9239|0.9136|0.6404|0.6149|
> > > |FairLoRA|0.9502|0.9418|0.6979|0.6449|
> > > |*One-shot FL (OFL)*
> > > |O-FedAvg|0.9079|0.8943|0.6188|0.5925|
> > > |FAFI|0.9431|0.9254|0.6712|0.6339|
> > > |**Ours**|**0.9553**|**0.9471**|**0.7028**|**0.6551**|

---

### Decision · Program_Chairs · 2026-04-30

**Decision:**

Accept (regular)

**Comment:**

This paper proposes a new group-fair one-shot federated learning framework, which is the first to jointly achieve group-fairness and one-round communication in federated learning. The framework consists of a fairness-aware expert routing module (for local training) that directs samples to demographic-specific experts and a prototype-guided differential aggregation module (for server aggregation) that filters conflicting updates using module-specific masks.

Overall, the work addresses an interesting and important problem, by designing a novel and practical framework backed up by solid experimental validation. Authors are encouraged to incorporate reviewers’ feedback, and in future research consider enhancing cohesion between different components and theoretical depth of the framework.